# UniFy: Efficient Modeling of Non-Stationary Periodicity for Time Series Forecasting

## Abstract

Periodic structures dominate long-term temporal dependencies in real-world signals, forming the cornerstone of long-term time series forecasting (LTSF). Existing methods typically aim to capture globally stable periodicity while overlooking the fact that real-world systems often exhibit substantial waveform variations across different periodic intervals. Representing such non-stationary sequences with a fixed period can lead to underfitting or overfitting of periodic components, thereby degrading forecasting accuracy. We further identify that the fundamental reason for this phenomenon is *frequency competition*, where multiple frequencies interfere with each other and distort the learning of periodic structures. We address this with a purely linear model: **Uni**fying Competing **F**requency (UniFy). It employs a multi-round Adaptive Frequency Selector (AFS) to progressively extract frequency components into multiple subspaces, mitigating frequency competition. Each subspace is then modeled by an Independent Linear Modeler (ILM) to extract its principal component, and the predictions from all subspaces are fused through Multi-subspace Calibration (MSC) to generate the final output. UniFy enables accurate and efficient modeling of non-stationary periodicity. Extensive experiments on 12 real-world datasets demonstrate the superiority of UniFy, delivering an average 16.0% MSE improvement on both long-term and short-term forecasting tasks, as well as an average 15.5% improvement in few-shot and zero-shot scenarios. Furthermore, its purely linear architecture ensures excellent computational efficiency and scalability. The code for our experiments is anonymously available at: `https://anonymous.4open.science/r/UniFy-22F2/`.

## 1 Introduction

Multivariate time series forecasting (MTSF) plays a critical role in various domains Jin et al. (2024) Liang et al. (2024), including transportation Li et al. (2024), finance Ni et al. (2024), and energy Chen et al. (2022), serving as a fundamental technique for intelligent scheduling, risk management, and resource optimization Mohammadi Foumani et al. (2024). With the rapid advancement of deep learning, forecasting models have achieved significant improvements on multiple public benchmarks. Both Transformer-based approaches (e.g., SDFormer Zhou et al. (2024), Autoformer Wu et al. (2021)) and lightweight linear models (e.g., CycleNet Lin et al. (2024), TimeMixer Wang et al. (2024)) have demonstrated strong predictive performance across diverse datasets.

In multi-step forecasting tasks, the ability of a model to accurately capture dominant dependency patterns is crucial for reducing error accumulation over longer horizons Dudek (2023). Extensive studies and practical evidence show that real-world time series often exhibit pronounced periodic structures, such as daily fluctuations in traffic flow or seasonal variations in energy consumption Shi & Li (2018). These periodic components encode a large portion of long-term dependencies and play a key role in improving forecasting stability and accuracy. Prior study has further revealed that **periodic structures typically represent the most deterministic part of a time series**, while other patterns, e.g., trends and event-driven fluctuations, are more uncertain and harder to predict Li et al. (2025).

Importantly, these periodic patterns are rarely stationary in practice: their amplitudes, phases, or even cycle lengths may drift over time, leading to what we refer to as non-stationary periodicity. From this perspective, non-stationary forecasting encompasses a broad spectrum of challenges, and

the non-stationary evolution of periodic components is among the most prominent and structured ones. Prioritizing accurate and efficient modeling of such non-stationary periodicity thus provides a strong inductive bias for time series forecasting, and also establishes a solid foundation for tackling the more general problem of non-stationary forecasting.

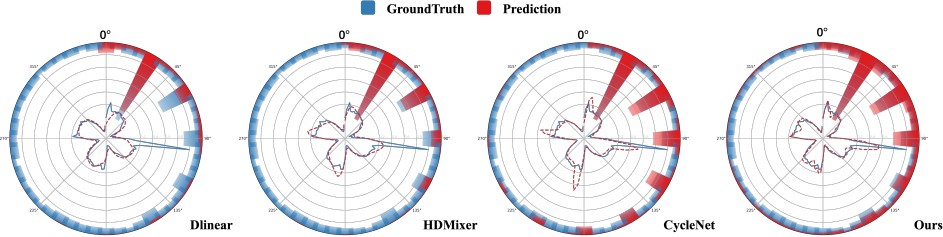

Figure 1: Polar-coordinate visualization comparing predictions and ground truth on the ETTh1 dataset. Inner curves depict time-domain prediction versus ground truth, and outer bars indicate prediction performance across frequency components. The polar axis is aligned clockwise with the Cartesian x-axis, starting at 0°.

To ground our discussion, Fig. 1 compares the predictions of Seasonal–Trend Decomposition (STD)–based models (DLinear) and state-of-the-art (SOTA) methods (HDMixer, CycleNet) in both time and frequency domains. In the time domain, we observe the following phenomenon:

Traditional periodic modeling methods are largely inspired by STD, focusing on capturing the dominant periodic patterns in a sequence and their variations. Models such as TimeMixer Wang et al. (2024), DLinear Zeng et al. (2023), and Autoformer follow this principle by using linear layers or Transformer blocks to extract the strongest cycles. While effective for stable and globally consistent periodicity, their results tend to collapse to a single periodic template, which lacks flexibility when periodicity evolves over time.

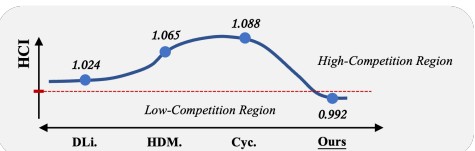

Figure 2: High-Competition Impact (HCI) comparison across datasets. A lower HCI indicates that the model is less affected by frequency competition. Refer to Appendix B for more details.

To improve adaptability, recent methods have introduced alternative designs. For example, HD-Mixer Huang et al. (2024) enhances local patch representations to retain richer periodic details, while CycleNet Lin et al. (2024) transfers periodic patterns from history to the future to reduce modeling bias. Despite this added flexibility, their predictions still approximate a single periodic template.

The frequency–domain view further clarifies the difference in failure modes: STD–based models concentrate almost exclusively on the dominant frequency and largely ignore local abrupt changes (e.g., spikes), leading to *underfitting* of periodic details; in contrast, HDMixer and CycleNet capture more frequency components and detect some sharp variations, but also introduce spurious spikes in inherently smooth regions, leading to *overfitting* of periodic details.

Taken together, although their spectral emphases differ, both classes of methods manifest similar template–like behavior in the time domain. This reveals a key gap: **existing approaches lack a principled mechanism to balance accurate modeling of the dominant component against interference from secondary components**. As a consequence, models tend to simplify periodic modeling into a fixed template and incur systematic bias. This bottleneck is particularly prominent in handling non-stationary periodicity.

We refer to this balancing dilemma as *frequency competition* (more details are available in Section A). As shown in Fig. 2, quantitative experiments under identical experimental conditions confirm that frequency competition in the data indeed has a negative impact on model prediction results.

In summary, in the presence of unavoidable frequency competition in time series, the dominant frequency suppresses weaker but informative components, leading to the loss of non-stationary details.

The chaotic secondary frequencies, in turn, affect the learning of the dominant frequency, causing spurious spikes in the base patterns. When multiple frequencies compete without clear separation, the model may misinterpret certain frequencies and generate spurious periodic patterns. This competitive mechanism fundamentally explains the *underfitting* and *overfitting* issues observed in Fig. 1.

Inspired by this insight, we propose **Uni**fying Competing **F**requency (UniFy), an exceptionally simple periodic modeling framework. We design an Adaptive Frequency Selector (AFS) that performs multi-round weighted selection of frequency components, constructing multiple independent frequency subspaces—highlighting the dominant frequencies in earlier rounds and progressively extracting finer components thereafter—thus alleviating frequency competition in the shared space. Each subspace is then processed by an Independent Linear Modeler (ILM) to perform dedicated modeling without interference from other frequencies. Finally, a Multi-subspace Calibration (MSC) module fuses the predictions from all subspaces and calibrates the bias between components. Thanks to its nearly entirely linear architecture, UniFy achieves a perfect balance between prediction accuracy, computational efficiency, and implementation simplicity.

Our main contributions are summarized as follows:

- We attribute the performance degradation in non-stationary periodic modeling to frequency competition, where dominant frequencies suppress secondary but informative components. Based on this insight, we propose UniFy that enhances periodic inductive bias and improves the stability and accuracy of long-horizon forecasts.

- We introduce a systematic architecture to explicitly mitigate frequency competition. The Adaptive Frequency Selector (AFS) progressively partitions the spectrum into multiple subspaces, ensuring that dominant and secondary components are decoupled rather than competing in a shared space. Each subspace is then handled by an Independent Linear Modeler (ILM). Finally, a Multi-subspace Calibration (MSC) module reconciles the subspace outputs in the temporal domain, correcting bias and scale misalignment accumulated during aggregation.

- We evaluate UniFy on 12 real-world MTSF datasets, achieving 16.0% average MSE gains on long-term and short-term forecasting tasks and up to 15.5% improvements in few-shot and zero-shot settings, while maintaining low latency, minimal memory usage, and strong computational efficiency, making it well-suited for large-scale applications.

## 2 RELATED WORK

Transformer architectures have achieved remarkable success in modeling long-term dependencies in time series, which has attracted considerable attention from both academia and industry. For instance, SDFormer Zhou et al. (2024) introduces a dynamic directional attention mechanism to strengthen the modeling of multivariate correlations.

In parallel, linear models have also gained increasing popularity in recent years. DLinear Zeng et al. (2023) demonstrates that simple linear structures excel at capturing periodic patterns. FreTS Yi et al. (2023) extends linear models into the frequency domain and leverages MLPs to enhance expressiveness from a global perspective. RLinear Li et al. (2023) further explores efficient recurrent structures for linear modeling. HDMixer Huang et al. (2024) employs a scalable patching mechanism that preserves distinct temporal segments and uses MLPs to model semantic interactions across patches, effectively improving local information capture. TimeMixer Wang et al. (2024) extracts multi-scale temporal features through downsampling and applies multiple MLPs at different granularities, followed by feature fusion for forecasting. CycleNet Lin et al. (2024) assumes that periodic components remain relatively stable in long-term forecasting and directly propagates historical periodic patterns into the prediction horizon.

Compared with these approaches, our method places stronger emphasis on the learning conflicts caused by frequency competition, and addresses this challenge within a lightweight linear framework. This design not only alleviates the redundancy across frequency bands but also provides an efficient and interpretable solution for time series forecasting.

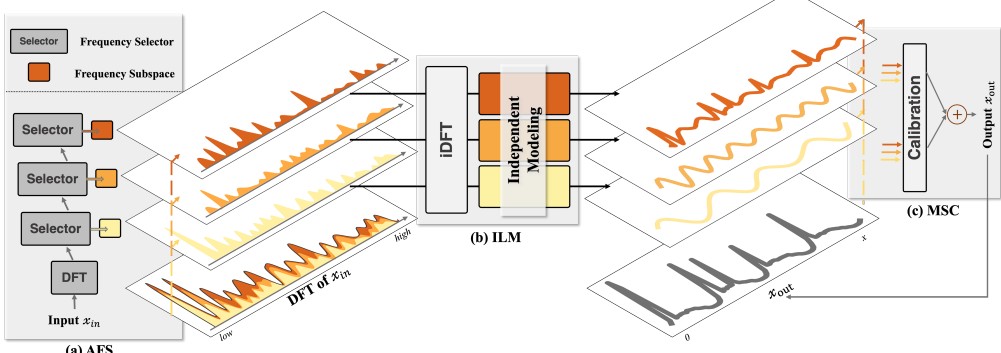

Figure 3: The UniFy pipeline. **(a) Adaptive Frequency Selector (AFS)**: Multi-round mask selection on the DFT spectrum, followed by inverse DFT to obtain scale-specific signals. **(b) Independent Linear Modeler (ILM)**: Independent linear projections for each scale, ensuring frequencies are modeled without mutual interference. **(c) Multi-subspace Calibration (MSC)**: A lightweight calibration module that removes scale misalignment across subspaces.

## 3 METHODOLOGY

The objective of MTSF is to predict the future values of each variable over $T$ time steps based on their respective historical values over the past $H$ time steps. Given the input series $X \in \mathbb{R}^{H \times d}$ with $d$ channels over $H$ history time steps. We aim to learn a function $f : \mathbf{x}_{t-H+1:t} \in \mathbb{R}^{H \times d} \rightarrow \bar{\mathbf{x}}_{t+1:t+T} \in \mathbb{R}^{T \times d}$.

### 3.1 OVERALL ARCHITECTURE

The overall architecture of UniFy is illustrated in Fig. 3, which primarily consists of the AFS module, ILM module, and MSC module. The overall workflow of UniFy is as follows: given an input sequence $X_{in} \in \mathbb{R}^{H \times d}$, we apply RevIn (Reversible Instance Normalization) Kim et al. (2022) to normalize the data and mitigate distributional shifts across different instances. The normalized sequence is then transformed into the frequency domain via the Discrete Fourier Transform (DFT). Next, the AFS module performs multi-round selection over frequency components using a set of learnable frequency masks, progressively constructing multiple frequency subspaces at different scales. Each subspace is then independently modeled by the ILM, which assigns dedicated linear layers to extract periodic features from different frequency bands. Finally, the MSC module fuses the outputs from all frequency subspaces, addressing potential misalignment across scales and generating the final forecasts. For multivariate time series with $d$ channels, UniFy applies the same pipeline to each channel independently: every channel is processed through the AFS, ILM, and MSC modules, ensuring that frequency competition and subspace modeling are handled consistently without cross-channel interference.

### 3.2 ADAPTIVE FREQUENCY SELECTOR

From a frequency-domain perspective, each discrete frequency point corresponds to a specific periodic component in the time domain, and the superposition of all such components reconstructs the original sequence. However, when all frequency components are jointly modeled within a single representation space, frequency competition often arises, where dominant frequencies overshadow secondary yet informative components. This imbalance leads to insufficient modeling of non-stationary periodic variations.

To mitigate this issue, we introduce the Adaptive Frequency Selector (AFS), which progressively decomposes the spectrum into multiple subspaces through a multi-round mask-selection mechanism. This design ensures that dominant and secondary frequencies are separated across rounds rather than forced to compete in a shared space.

Given an input sequence $X_{in} \in \mathbb{R}^{H \times d}$ (per instance), we first transform it into the frequency domain using the Discrete Fourier Transform (DFT). For each channel $j \in \{1, \ldots, d\}$, its temporal values $\{p_{t,j}\}_{t=0}^{H-1}$ yield

$$P_{k,j} = \sum_{t=0}^{H-1} p_{t,j}\, e^{-i2\pi kt/H}, \qquad k = 0, \ldots, H-1. \tag{1}$$

For notational convenience, let $\mathbf{P}_k = [P_{k,1}, \ldots, P_{k,d}]^\top \in \mathbb{C}^d$ denote the channel-stacked spectrum at frequency bin $k$.

AFS performs decomposition in $R$ rounds over the residual spectrum. Let $\mathbf{R}_k^{(1)} = \mathbf{P}_k$. In each round $r \in \{1, \ldots, R\}$, a learnable one-dimensional mask $\mathbf{m}^{(r)} \in [0, 1]^H$ (shared across channels and broadcast along $d$) assigns an importance weight to every frequency bin. The selected spectrum and the residual update are:

$$\widetilde{\mathbf{P}}_k^{(r)} = m_k^{(r)}\, \mathbf{R}_k^{(r)}, \qquad \mathbf{R}_k^{(r+1)} = (1 - m_k^{(r)})\, \mathbf{R}_k^{(r)}, \quad k = 0, \ldots, H-1. \tag{2}$$

After each round, only the unselected portion is passed to the next round, ensuring complementarity among subspaces.

Each selected subspace is projected back to the time domain via inverse DFT:

$$\widetilde{x}_{t,j}^{(r)} = \frac{1}{H} \sum_{k=0}^{H-1} \widetilde{P}_{k,j}^{(r)}\, e^{i2\pi kt/H}, \qquad t = 0, \ldots, H-1. \tag{3}$$

To ensure real-valued reconstruction, we tie mask weights at conjugate frequency pairs and (in implementation) take the real part of the inverse transform. This yields scale-specific temporal components $\{\widetilde{X}^{(r)}\}_{r=1}^R$, each corresponding to one frequency subspace with shape $\mathbb{R}^{H \times d}$.

In this way, AFS adaptively separates the spectrum into multiple interpretable frequency scales, preventing dominant frequencies from overwhelming weaker yet informative ones and laying the foundation for dedicated modeling in subsequent modules.

### 3.3 INDEPENDENT LINEAR MODELER

Given $\{\widetilde{X}^{(r)}\}_{r=1}^R$, the Independent Linear Modeler (ILM) performs scale-specific modeling by allocating an independent linear projection to each frequency scale. For the $r$-th component $\widetilde{X}^{(r)} \in \mathbb{R}^{H \times d}$, ILM applies

$$Y^{(r)} = \left( (\widetilde{X}^{(r)})^\top \mathbf{W}^{(r)} \right)^\top + \mathbf{b}^{(r)}, \qquad r = 1, \ldots, R, \tag{4}$$

where $\mathbf{W}^{(r)} \in \mathbb{R}^{H \times T}$ maps the history length $H$ to the forecasting horizon $T$ along the temporal dimension and is shared across channels, and $\mathbf{b}^{(r)} \in \mathbb{R}^T$ is broadcast over the $d$ channels. The output $Y^{(r)} \in \mathbb{R}^{T \times d}$. By assigning a dedicated linear layer per scale, ILM avoids interference among frequency subspaces while keeping computation low; each projection focuses on the dominant periodic components within its own scale.

By assigning a dedicated linear layer to each scale, ILM avoids interference among different frequency subspaces and allows each projection to focus on the dominant periodic components present within its own scale. This simple yet effective design enhances sensitivity to key frequency patterns while maintaining low computation.

### 3.4 MULTI-SCALE CALIBRATION

After ILM, the per-scale predictions $\{Y^{(r)}\}_{r=1}^R$ are first aggregated by element-wise summation:

$$S = \sum_{r=1}^R Y^{(r)} \in \mathbb{R}^{T \times d}. \tag{5}$$

Since $S$ combines multiple subspaces, it may suffer from cross-scale bias and temporal misalignment. We therefore apply a lightweight Multi-Scale Calibration (MSC) head—implemented as a two-layer MLP along the temporal dimension and shared across channels:

$$\widehat{Y} \;=\; \mathbf{W}^{(2)}\, \sigma\big(\mathbf{W}^{(1)}S\big), \tag{6}$$

where $\mathbf{W}^{(1)} \in \mathbb{R}^{d_{\text{model}} \times T}$, $\mathbf{W}^{(2)} \in \mathbb{R}^{T \times d_{\text{model}}}$, and $\sigma(\cdot)$ is a pointwise nonlinearity. This calibration re-aligns the aggregated temporal envelope and suppresses cross-scale bias, yielding the final prediction $\widehat{Y} \in \mathbb{R}^{T \times d}$.

## 4 EXPERIMENTS

We conduct extensive experiments to evaluate the performance and efficiency of UniFy, covering long-term forecasting, short-term forecasting, zero-shot forecasting and few-shot forecasting, including 12 real-world benchmarks and 9 baselines. As shown in Fig. 4, UniFy achieves state-of-the-art accuracy while maintaining very competitive efficiency (less time & Gpu Mem).

**Benchmarks** For long-term forecasting, we experiment on 8 well-established benchmarks: ETT datasets (including 4 subsets: ETTh1, ETTh2, ETTm1, ETTm2), Weather, Electricity, Exchange, and Traffic following Wu et al. (2022). For short-term forecasting, we adopt the PeMS Chen et al. (2001) which contains four public traffic network datasets (PEMS03, PEMS04, PEMS07, PEMS08). Furthermore, we test the few-shot forecasting and zero-shot forecasting on the ETT datasets.

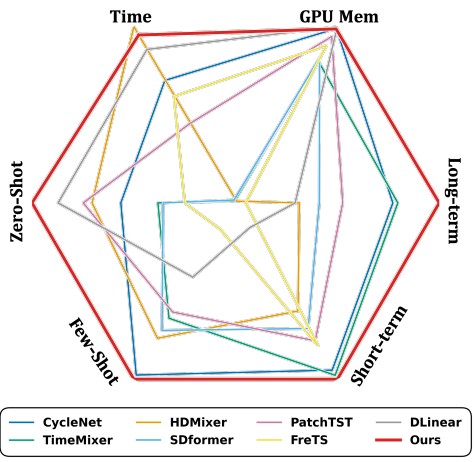

Figure 4: Overall experimental results. All experiments are conducted with an input length of 96 in this work.

**Baselines** We compare UniFy with eight baselines, which comprise the state-of-the-art long-term forecasting model CycleNet Lin et al. (2024) and advanced short-term forecasting models TimesNet Wu et al. (2022). As well as other competitive models including Transformer-based models: SDformer Zhou et al. (2024), PatchTST Nie et al. (2023), Autoformer Wu et al. (2021); Linear-based model: TimeMixer Wang et al. (2024), HDMixer Huang et al. (2024), FreTS Yi et al. (2023) and DLinear Zeng et al. (2023).

**Long-term and Short-term Forecasting Results** As shown in Table 1, UniFy consistently achieves state-of-the-art performance across all eight datasets, with an overall improvement of about 21%. We acknowledge that on the ECL dataset our method performs on par with CycleNet, and on the Traffic dataset the improvements are relatively minor. As reported in the Section A, these two datasets exhibit the weakest frequency competition and highly stable periodic patterns. Such properties particularly favor CycleNet, whose mechanism directly extrapolates future periods from predefined historical cycles. Consequently, CycleNet remains highly competitive on these datasets. Nevertheless, UniFy delivers substantially higher average performance on the other benchmarks.

It is also worth noting that CycleNet relies heavily on manually predefined cycle lengths, making its performance highly sensitive to even small variations in this parameter. This dependence exposes a significant limitation of CycleNet in more complex or less strictly periodic scenarios.

For short-term forecasting (Table 2), UniFy achieves a clear breakthrough, surpassing the current best-performing model (TimesNet) with a 28% reduction in MSE. When combined with the efficiency analysis, the results further demonstrate that UniFy strikes an excellent balance between predictive accuracy and computational cost, achieving SOTA overall performance in both effectiveness and efficiency.

Table 1: **Long-Term Forecasting Results**. A lower MSE or MAE indicates a better prediction. All the results are averaged from 4 different prediction lengths, that is {96, 192, 336, 720}. The best results are highlighted in **bold**.

| Models | UniFy (Ours) | | CycleNet (2024) | | TimeMixer (2024) | | HDMixer (2024) | | SDformer (2024) | | PatchTST (2023) | | FreTS (2023) | | TimesNet (2023) | | DLinear (2023) | | Autoformer (2021) | |
|---|---|---|---|---|---|---|---|---|---|---|---|---|---|---|---|---|---|---|---|---|
| Metric | MSE | MAE | MSE | MAE | MSE | MAE | MSE | MAE | MSE | MAE | MSE | MAE | MSE | MAE | MSE | MAE | MSE | MAE | MSE | MAE |
| ETTm1 | **0.380** | **0.394** | 0.380 | 0.397 | 0.387 | 0.400 | 0.396 | 0.400 | 0.409 | 0.412 | 0.387 | 0.400 | 0.405 | 0.413 | 0.400 | 0.406 | 0.414 | 0.408 | 0.588 | 0.517 |
| ETTm2 | **0.272** | **0.319** | 0.278 | 0.322 | 0.278 | 0.324 | 0.285 | 0.330 | 0.289 | 0.332 | 0.281 | 0.326 | 0.337 | 0.373 | 0.291 | 0.333 | 0.286 | 0.327 | 0.327 | 0.371 |
| ETTh1 | **0.432** | **0.429** | 0.454 | 0.439 | 0.464 | 0.447 | 0.461 | 0.444 | 0.467 | 0.457 | 0.469 | 0.455 | 0.481 | 0.466 | 0.458 | 0.450 | 0.459 | 0.456 | 0.496 | 0.487 |
| ETTh2 | **0.363** | **0.394** | 0.386 | 0.409 | 0.374 | 0.402 | 0.380 | 0.403 | 0.387 | 0.405 | 0.387 | 0.407 | 0.509 | 0.486 | 0.414 | 0.427 | 0.374 | 0.399 | 0.450 | 0.459 |
| ECL | 0.171 | 0.262 | **0.170** | **0.260** | 0.184 | 0.274 | 0.215 | 0.302 | 0.207 | 0.300 | 0.205 | 0.290 | 0.196 | 0.284 | 0.193 | 0.295 | 0.219 | 0.298 | 0.227 | 0.338 |
| Traffic | **0.467** | 0.311 | 0.472 | 0.302 | 0.485 | **0.298** | 0.646 | 0.417 | 0.553 | 0.376 | 0.555 | 0.362 | 0.600 | 0.386 | 0.620 | 0.336 | 0.627 | 0.378 | 0.628 | 0.379 |
| Weather | **0.242** | **0.271** | 0.254 | 0.279 | 0.246 | 0.275 | 0.258 | 0.285 | 0.265 | 0.285 | 0.265 | 0.285 | 0.253 | 0.273 | 0.259 | 0.287 | 0.272 | 0.291 | 0.338 | 0.382 |
| Exchange | **0.352** | **0.398** | 0.382 | 0.412 | 0.364 | 0.404 | 0.377 | 0.409 | 0.394 | 0.423 | 0.367 | 0.404 | 0.368 | 0.411 | 0.416 | 0.443 | 0.379 | 0.418 | 0.613 | 0.539 |

Table 2: **Short-Term Forecasting Results** in the PEMS datasets with multiple variates. A lower MSE or MAE indicates a better prediction. All the results are averaged from 4 different prediction lengths, that is {12, 24, 48, 96}. The best results are highlighted in **bold**.

| Models | UniFy (Ours) | | CycleNet (2024) | | TimeMixer (2024) | | HDMixer (2024) | | SDformer (2024) | | PatchTST (2023) | | FreTS (2023) | | TimesNet (2023) | | DLinear (2023) | | Autoformer (2021) | |
|---|---|---|---|---|---|---|---|---|---|---|---|---|---|---|---|---|---|---|---|---|
| Metric | MSE | MAE | MSE | MAE | MSE | MAE | MSE | MAE | MSE | MAE | MSE | MAE | MSE | MAE | MSE | MAE | MSE | MAE | MSE | MAE |
| PEMS03 | **0.131** | **0.237** | 0.162 | 0.264 | 0.151 | 0.272 | 0.252 | 0.354 | 0.222 | 0.317 | 0.180 | 0.291 | 0.168 | 0.272 | 0.147 | 0.248 | 0.495 | 0.472 | 0.667 | 0.601 |
| PEMS04 | **0.106** | **0.216** | 0.125 | 0.239 | 0.107 | 0.223 | 0.354 | 0.399 | 0.265 | 0.348 | 0.195 | 0.307 | 0.187 | 0.291 | 0.129 | 0.241 | 0.526 | 0.491 | 0.610 | 0.590 |
| PEMS07 | **0.101** | **0.206** | 0.116 | 0.219 | 0.109 | 0.224 | 0.254 | 0.335 | 0.209 | 0.306 | 0.211 | 0.303 | 0.180 | 0.279 | 0.125 | 0.226 | 0.505 | 0.478 | 0.367 | 0.451 |
| PEMS08 | **0.126** | **0.238** | 0.156 | 0.250 | 0.138 | 0.245 | 0.320 | 0.390 | 0.300 | 0.352 | 0.280 | 0.321 | 0.281 | 0.309 | 0.193 | 0.271 | 0.529 | 0.487 | 0.814 | 0.659 |

**Model Efficiency** We evaluated the training time and GPU memory consumption of UniFy against various baselines, as shown in Fig. 5. UniFy achieves state-of-the-art predictive accuracy while maintaining lightweight computational requirements. In contrast, time-intensive models such as SDFormer and TimesNet incur substantial overhead, limiting their responsiveness in real-world scenarios. HD-Mixer, despite its linear architecture, introduces high complexity and memory usage due to its patch-based decomposition, hindering deployment on diverse hardware platforms. While linear models like Cy-cleNet and DLinear are efficient, their limited periodic modeling capacity constrains forecasting performance. UniFy strikes a better balance between accuracy and efficiency, making it a practical and scalable solution for time series forecasting applications.

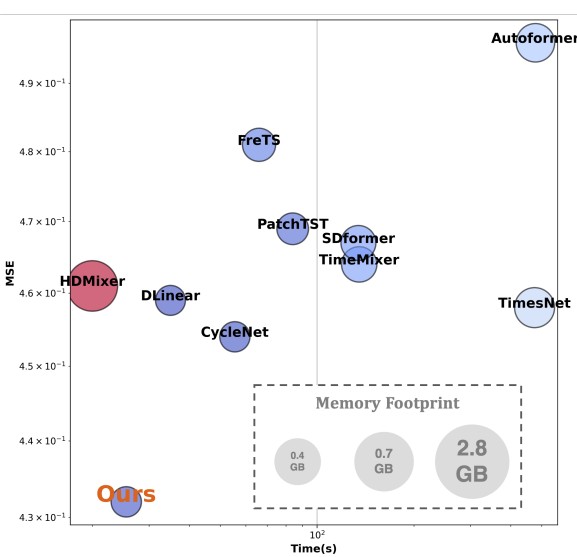

Figure 5: Efficiency comparison on the ETTh1 dataset. Results are averaged over four prediction lengths {96, 192, 336, 720}. All experiments are conducted with a fixed batch size of 8 and hidden dimension $d_{\text{model}} = 128$. Additional details are in Appendix E.

**Zero-Shot and Few-Shot Forecasting Results** To further evaluate the generalization ability of UniFy on time series data, we design few-shot and zero-shot forecasting experiments using the ETT dataset. This dataset, collected by the same sensor over different time periods, exhibits transferable periodic characteristics, making it particularly suitable for assessing a model's capability to capture and generalize periodic patterns. None of the baseline methods, including UniFy, were specifically fine-tuned for these data-scarce scenarios. As shown in Table.3 and Table.4, UniFy consistently outperforms all competing approaches under both settings, delivering substantial gains over state-of-the-art baselines. These results highlight UniFy's robust generalization to diverse, time-varying periodic structures, enabling it to transfer learned periodic knowledge to unseen tasks with minimal supervision. This demonstrates its ability to adapt to non-

Table 3: **Zero-Shot Forecasting Results** in the ETT datasets. A lower MSE or MAE indicates a better prediction. All the results are averaged from 4 different prediction lengths, that is {96, 192, 336, 720}. The best results are highlighted in **bold**.

| Models | UniFy (Ours) | | CycleNet (2024) | | TimeMixer (2024) | | HDMixer (2024) | | SDformer (2024) | | PatchTST (2023) | | FreTS (2023) | | TimesNet (2023) | | DLinear (2023) | | Autoformer (2021) | |
|---|---|---|---|---|---|---|---|---|---|---|---|---|---|---|---|---|---|---|---|---|
| Metric | MSE | MAE | MSE | MAE | MSE | MAE | MSE | MAE | MSE | MAE | MSE | MAE | MSE | MAE | MSE | MAE | MSE | MAE | MSE | MAE |
| ETTm1 ⇒ ETTm2 | **0.293** | 0.333 | 0.295 | 0.329 | 0.302 | 0.339 | 0.295 | **0.328** | 0.299 | 0.335 | 0.296 | 0.331 | 0.338 | 0.392 | 0.377 | 0.392 | 0.362 | 0.405 | 0.502 | 0.513 |
| ETTm2 ⇒ ETTm1 | **0.523** | **0.469** | 0.563 | 0.481 | 0.578 | 0.492 | 0.558 | 0.478 | 0.709 | 0.543 | 0.567 | 0.493 | 0.909 | 0.552 | 1.020 | 0.644 | 0.530 | 0.475 | 0.811 | 0.619 |
| ETTh2 ⇒ ETTh1 | **0.481** | **0.468** | 0.665 | 0.548 | 0.618 | 0.530 | 0.559 | 0.509 | 0.668 | 0.561 | 0.559 | 0.505 | 0.562 | 0.516 | 0.761 | 0.590 | 0.506 | 0.487 | 0.865 | 0.637 |
| ETTm1 ⇒ ETTh1 | **0.585** | **0.515** | 0.705 | 0.563 | 0.834 | 0.614 | 0.730 | 0.583 | 0.767 | 0.590 | 0.645 | 0.548 | 1.110 | 0.689 | 0.693 | 0.548 | 0.618 | 0.537 | 1.016 | 0.694 |
| ETTh1 ⇒ ETTm1 | **0.752** | 0.572 | 0.859 | 0.589 | 0.892 | 0.601 | 0.832 | 0.586 | 0.863 | 0.595 | 0.781 | **0.565** | 0.795 | 0.599 | 1.235 | 0.725 | 0.754 | 0.583 | 0.827 | 0.619 |
| ETTh2 ⇒ ETTm1 | **0.727** | **0.559** | 0.959 | 0.625 | 1.108 | 0.647 | 0.850 | 0.592 | 1.066 | 0.653 | 0.908 | 0.606 | 0.825 | 0.597 | 1.504 | 0.750 | 0.792 | 0.588 | 0.770 | 0.597 |

Table 4: **Few-Shot Forecasting Results** in the ETT datasets with 10% training data. A lower MSE or MAE indicates a better prediction. All the results are averaged from 4 different prediction lengths, that is {96, 192, 336, 720}. The best results are highlighted in **bold**.

| Models | UniFy (Ours) | | CycleNet (2024) | | TimeMixer (2024) | | HDMixer (2024) | | SDformer (2024) | | PatchTST (2023) | | FreTS (2023) | | TimesNet (2023) | | DLinear (2023) | | Autoformer (2021) | |
|---|---|---|---|---|---|---|---|---|---|---|---|---|---|---|---|---|---|---|---|---|
| Metric | MSE | MAE | MSE | MAE | MSE | MAE | MSE | MAE | MSE | MAE | MSE | MAE | MSE | MAE | MSE | MAE | MSE | MAE | MSE | MAE |
| ETTm1 | **0.389** | **0.398** | 0.400 | 0.401 | 0.487 | 0.461 | 0.446 | 0.431 | 0.447 | 0.431 | 0.501 | 0.466 | 0.544 | 0.484 | 0.677 | 0.537 | 0.411 | 0.429 | 0.802 | 0.628 |
| ETTm2 | 0.281 | 0.324 | **0.279** | **0.323** | 0.311 | 0.367 | 0.294 | 0.336 | 0.294 | 0.336 | 0.296 | 0.343 | 0.352 | 0.394 | 0.320 | 0.353 | 0.316 | 0.368 | 1.342 | 0.930 |
| ETTh1 | **0.450** | **0.437** | 0.450 | 0.441 | 0.613 | 0.520 | 0.582 | 0.518 | 0.589 | 0.521 | 0.633 | 0.542 | 0.739 | 0.605 | 0.869 | 0.628 | 0.691 | 0.600 | 0.702 | 0.596 |
| ETTh2 | **0.382** | **0.404** | 0.395 | 0.412 | 0.402 | 0.433 | 0.389 | 0.415 | 0.421 | 0.430 | 0.415 | 0.431 | 0.632 | 0.535 | 0.479 | 0.465 | 0.605 | 0.538 | 0.488 | 0.499 |

stationary cycles and provide a strong periodic inductive bias for reliable forecasting even under distributional shifts.

**Ablation Study**   As shown in Table. 5, we conduct a systematic ablation study on UniFy by progressively removing its core components to assess their individual contributions to model performance.

*w/o AFS*: the model directly learns from the undecomposed original sequence;

*w/o ILM*: only retain the most dominant frequency scale while discarding the others scales of AFS;

*w/o MSC*: outputs from all scales are simply summed to produce the final prediction.

Table 5: **Ablation Experiment Results.** A lower MSE or MAE indicates a better prediction. All the results are averaged from 4 different prediction lengths, that is {96, 192, 336, 720}.

| Models | ETT(Avg) | | ECL | | Traffic | | Weather | |
|---|---|---|---|---|---|---|---|---|
| Metric | MSE | MAE | MSE | MAE | MSE | MAE | MSE | MAE |
| UniFy | **0.371** | **0.389** | **0.177** | **0.247** | **0.239** | **0.284** | **0.244** | **0.302** |
| *w/o AFS* | 0.569 | 0.508 | 0.189 | 0.283 | 0.255 | 0.307 | 0.652 | 0.660 |
| *w/o ILM* | 0.373 | 0.391 | 0.188 | 0.280 | 0.239 | 0.285 | 0.274 | 0.338 |
| *w/o MSC* | 0.421 | 0.425 | 0.233 | 0.319 | 0.251 | 0.295 | 0.438 | 0.489 |

The experimental results demonstrate that removing any of these modules leads to significant performance degradation, thereby validating the necessity of UniFy. Additional ablation studies with more comprehensive settings are presented in Section G, Section H, Section I to further examine the effectiveness of each component.

**Gradient-based Attribution for Frequency Decomposition**   To examine whether the proposed frequency decomposition module allocates distinct spectral components to different scales, we performed a gradient-based attribution experiment. Following Section 3.2, for an input sequence $x \in \mathbb{R}^{B \times T \times C}$, we first apply DFT along the temporal dimension to obtain its spectrum $P \in \mathbb{C}^{T \times C}$. Each scale $i \in \{1, \ldots, K-1\}$ is parameterized by a learnable frequency mask $\sigma(w_i) \in [0,1]^T$, where $w_i$ corresponds to the trainable vectors `freq_weights[i]` in our implementation (More details have shown in Section. C). The masked spectrum at scale $i$ is:

$$\widetilde{P}^{(i)} = \sigma(w_i) \odot P, \tag{7}$$

and the residual spectrum is defined as:

$$P_{\text{residual}} = P - \sum_{i=1}^{K-1} \widetilde{P}^{(i)}, \tag{8}$$

ensuring that all spectral components are fully partitioned across $K$ scales. In practice, the residual attribution is computed as the negative sum of gradients from the other scales, which is equivalent to this constraint.

We define the attribution objective as the mean squared error (MSE) between the model prediction $\hat{y}$ and the ground truth $y$:

$$J = \text{MSE}(\hat{y}, y). \tag{9}$$

The importance of each frequency bin is quantified by computing the gradient of this loss with respect to the mask parameters:

$$g_i = \frac{\partial J}{\partial w_i}, \quad i = 1, \dots, K-1, \qquad g_{\text{residual}} = -\sum_{i=1}^{K-1} g_i. \tag{10}$$

Here, $|g_i|$ is taken as the attribution score, reflecting how strongly the spectral bins selected at scale $i$ influence the predictive objective. Following common practice in gradient-based attribution for spectral analysis, we use normalized gradient magnitudes for visualization, which highlight scale-specific responsibilities and provide contrast across different frequency ranges.

For visualization, we retain only the first half of the DFT bins due to the Hermitian symmetry of real-valued signals, and normalize the gradient curves for comparability across scales. Figure 6 presents the normalized attribution scores obtained for three decomposition scales. We observe that the attribution peaks of the three scales are clearly separated, with minimal overlap across frequency regions. This indicates that the decomposition module assigns complementary responsibilities to different scales, rather than redundantly emphasizing the same spectral components. Such specialization empirically validates the design intuition of AFS: by progressively partitioning the spectrum through learnable masks, it disentangles dominant and secondary frequencies into independent subspaces. This reduces destructive interference i.e., frequency competition in a shared space, thereby improving predictive stability and enhancing interpretability.

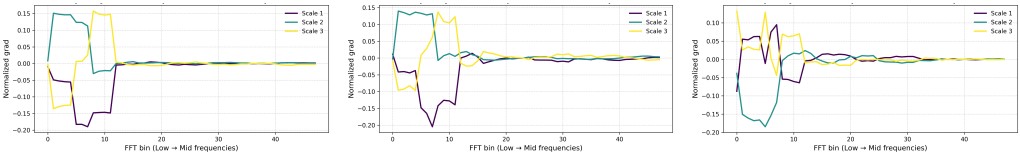

Figure 6: Gradient-based attribution for frequency decomposition results on ETTh1.

## 5 CONCLUSION

This work identifies *frequency competition*, where dominant frequencies suppress secondary yet informative components while secondary ones, in turn, introduce noise to dominant frequencies. We argue that this reciprocal interference constitutes a fundamental bottleneck in non-stationary periodic modeling, leading to inaccurate representations of temporal structures. To address this challenge, we proposed **UniFy**, a purely linear forecasting framework that: (i) employs an **Adaptive Frequency Selector (AFS)** to progressively partition the spectrum into multiple subspaces, thereby separating dominant and secondary frequencies; (ii) applies **Independent Linear Modelers (ILMs)** to perform scale-specific modeling for each subspace without mutual interference; and (iii) leverages a lightweight **Multi-Subspace Calibration (MSC)** module to align, correct, and fuse cross-subspace predictions into a consistent temporal envelope. This design enables fine-grained modeling of non-stationary periodicity while maintaining computational efficiency and scalability. Unlike existing methods that rely on a single periodic template or manually predefined cycles, UniFy adaptively learns frequency responsibilities across scales, which enhances robustness to waveform variations and reduces modeling bias.

Extensive experiments across long-term, short-term, few-shot, and zero-shot forecasting tasks demonstrate that UniFy consistently outperforms nine state-of-the-art baselines, delivering substantial improvements in both accuracy and efficiency. Beyond raw performance gains, the gradient-based attribution analysis further provides interpretability, showing how UniFy disentangles frequency responsibilities and alleviates destructive interference between scales.

## 6 ETHICS STATEMENT

Our study relies exclusively on publicly available time series datasets that have been widely adopted in prior research. No private, sensitive, or personally identifiable information was used. We strongly advocate for the responsible deployment of our model: while UniFy has the potential to advance forecasting in domains such as energy, transportation, and healthcare, practitioners should carefully consider broader social impacts and avoid uses that could lead to harmful outcomes such as surveillance, profiling, or discriminatory decision-making. All experiments were conducted under principles of academic integrity. By openly sharing our methodology, we aim to promote transparency, ethical scrutiny, and constructive dialogue within the research community.

## 7 REPRODUCIBILITY STATEMENT

To ensure full reproducibility of our results, we provide comprehensive implementation details throughout the paper. Section 3 describes the construction of UniFy in detail, and the Appendix includes additional explanations of datasets, hyperparameter choices, and evaluation protocols. In accordance with our commitment to open science, we will release the source code and configuration files of UniFy upon publication, enabling other researchers to replicate, verify, and extend our findings. The code for our experiments is anonymously available at: `https://anonymous.4open.science/r/UniFy-22F2/`.

## 8 USE OF LLMS

We clarify the use of large language models (LLMs) during the preparation of this manuscript. LLMs were employed exclusively for supportive purposes in three aspects: (i) assisting in the translation of early drafts written in the authors' native language, (ii) providing linguistic refinement such as correcting grammar, improving sentence structure, and enhancing readability, and (iii) offering technical assistance in code editing and debugging (e.g., clarifying library usage, resolving implementation issues, or refactoring scripts for clarity).

It is important to emphasize that all core scientific ideas, methodological contributions, analyses, and results presented in this work are entirely the intellectual product of the human authors. The LLMs functioned strictly as auxiliary tools for writing and coding support, and were not involved in the conceptual or analytical aspects of the research.

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

## A   BACKGROUND: DEFINITION OF FREQUENCY COMPETITION

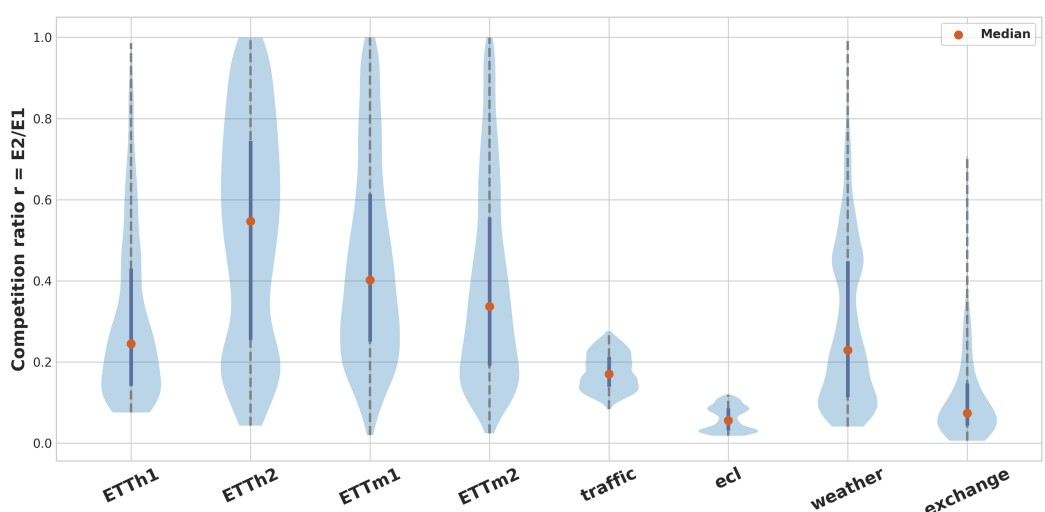

Figure 7: Visualization of frequency competition across all datasets. For each dataset, the multivariate time series is segmented into overlapping windows of length $L = 720$ with stride $S = 16$. Within each window, we compute the rDFT of the channel-wise averaged signal and calculate the competition ratio $r = E_2/E_1$ between the second and first dominant frequency components.

To gain a deeper understanding of the role of frequency competition in time series forecasting, we propose a case-study pipeline that quantifies and visualizes the interaction between the dominant and secondary spectral components. This allows us to assess the degree of frequency competition within a given dataset.

Formally, let $\mathbf{x} = \{x_t\}_{t=0}^{L-1}$ denote the channel-averaged signal within a window of length $L$. We compute its discrete Fourier transform (DFT):

$$X_k = \sum_{t=0}^{L-1} x_t \, e^{-j2\pi kt/L}, \quad k = 0, 1, \ldots, L-1, \tag{11}$$

and the corresponding amplitude spectrum:

$$A_k = |X_k|, \quad E_k = |X_k|^2, \tag{12}$$

where $E_k$ denotes the energy at frequency bin $k$.

We then identify the dominant peak index

$$k_1 = \arg \max_{k \notin \mathcal{N}_0} E_k, \tag{13}$$

where $\mathcal{N}_0$ is a small neighborhood around the DC component that is excluded. The dominant energy is thus $E_1 = E_{k_1}$.

To find the secondary peak, we suppress a small neighborhood $\mathcal{N}_{k_1}$ around $k_1$ to avoid overlap, and define:

$$k_2 = \arg \max_{k \notin \mathcal{N}_{k_1}} E_k, \quad E_2 = E_{k_2}. \tag{14}$$

Finally, we define the competition ratio as

$$r = \frac{E_2}{E_1}, \tag{15}$$

which quantifies the relative contribution of the secondary frequency with respect to the dominant one. A small $r$ indicates that the dominant oscillation prevails, whereas a large $r$ implies significant competition from alternative frequencies.

Based on this definition, we analyze the frequency competition distribution across the entire dataset. Specifically, we use violin plots with box-like annotations to visualize the distribution of $r$. The shape of the violin represents the density of the competition ratio, while scatter markers indicate the median, and vertical bars denote the interquartile range (IQR) and the overall span.

Through Fig 7, we can assess the prevalence of strong frequency competition in each dataset and compare the frequency competition patterns across different benchmarks. We observe that the ETT series datasets exhibit severe frequency competition, while the Traffic and ECL datasets show mild competition, and the Weather and Exchange datasets show moderate competition.

## B    QUANTITATIVE ANALYSIS OF FREQUENCY COMPETITION: HIGH-COMPETITION IMPACT (HCI)

In this section, we build on the analysis from Section A, where we quantified the competition levels in the dataset. Now, we focus on quantifying the impact of the high-competition regions on the model's prediction error.

We define the **High-Competition Impact (HCI)** metric, which measures the relative contribution of high-competition regions to the overall prediction error. The calculation proceeds as follows:

**Step 1: Frequency Competition Ratio ($r$)**    As described in Section A, the competition ratio $r = E_2/E_1$ is computed for each time window, where $E_1$ and $E_2$ denote the dominant and secondary peak energies. Based on two thresholds (`weak_thr` and `strong_thr`), each window is categorized into low, medium, or high competition.

**Step 2: Squared Error (SSE) Calculation**    For each time point, the squared error is computed as

$$\text{SSE}_{\text{total}} = \sum_{i=1}^{N \times L} (\text{gt}_i - \text{pred}_i)^2, \tag{16}$$

where $N$ is the batch size and $L$ is the sequence length.

**Step 3: Error Attribution by Competition Level**    The squared error is then aggregated according to the competition category of each window. This yields $\text{SSE}_{\text{low}}, \text{SSE}_{\text{medium}}, \text{SSE}_{\text{high}}$, corresponding to the contributions of low-, medium-, and high-competition regions.

**Step 4: High-Competition Impact (HCI)**    Finally, we define HCI as the relative error contribution of high-competition regions, normalized by their coverage:

$$\text{HCI} = \frac{\frac{\text{SSE}_{\text{high}}}{\text{SSE}_{\text{total}}}}{\frac{W_{\text{high}}}{B \cdot L}}, \tag{17}$$

where $W_{\text{high}}$ denotes the effective number of time points covered by high-competition windows, and $B \cdot L$ is the total number of time points.

The HCI quantifies the relative impact of the high-competition regions by comparing their contribution to the total error against their coverage. Specifically:

- **HCI $> 1$**: High-competition regions contribute disproportionately to the total error, indicating these regions are particularly challenging.
- **HCI $\approx 1$**: High-competition regions contribute error proportionally to their coverage, suggesting no exceptional difficulty.
- **HCI $< 1$**: High-competition regions contribute less error than expected, suggesting they are easier to predict.

## C    GRADIENT-BASED FREQUENCY ATTRIBUTION EXPERIMENT DETAILS

We provide the detailed procedure of our gradient-based frequency attribution experiment, corresponding to the implementation in `analyze_frequency_attribution()` within UniFy. The

goal is to measure the sensitivity of different frequency bins to the predictive objective, by backpropagating the loss with respect to the learnable frequency masks in the Adaptive Frequency Selector (AFS).

Table 6: Procedure of gradient-based frequency attribution in UniFy.

| Step | Description |
|------|-------------|
| 1 | Normalize input $x \in \mathbb{R}^{B \times T \times C}$ by subtracting the mean and dividing by the standard deviation. |
| 2 | Flatten channel dimension: $x \to x' \in \mathbb{R}^{(B \cdot C) \times T}$. |
| 3 | Apply DFT along temporal axis to obtain spectrum $P$. Initialize residual spectrum $R^{(1)} \leftarrow P$. |
| 4 | For each round $r = 1, \ldots, R - 1$:
(a) Compute mask $\mathbf{m}^{(r)} = \sigma(w^{(r)}) \in [0,1]^T$.
(b) Select spectrum $\widetilde{P}^{(r)} = \mathbf{m}^{(r)} \odot R^{(r)}$.
(c) Update residual $R^{(r+1)} = (1 - \mathbf{m}^{(r)}) \odot R^{(r)}$.
(d) Inverse DFT: $\widetilde{X}^{(r)} = \mathrm{iDFT}(\widetilde{P}^{(r)})$. |
| 5 | Obtain final residual component $\widetilde{X}^{(R)} = \mathrm{iDFT}(R^{(R)})$. |
| 6 | Apply ILM and MSC to $\{\widetilde{X}^{(r)}\}_{r=1}^R$ to generate prediction $\hat{y}$. |
| 7 | Compute attribution objective $J$: by default $J = \mathrm{MSE}(\hat{y}, y)$. |
| 8 | Backpropagate to obtain gradients $g^{(r)} = \partial J / \partial w^{(r)}$ for $r = 1, \ldots, R - 1$. |
| 9 | Compute residual attribution $g^{(R)} = -\sum_{r=1}^{R-1} g^{(r)}$ to cover all scales. |
| 10 | Normalize $\{g^{(r)}\}$ (abs-max or z-score) for visualization. |

# D    Implementation Details

We summarized details of datasets, evaluation metrics and experiments in this section.

**Datasets details** We evaluate the performance of different models for long-term forecasting on 8 well-established datasets including ETT datasets (including 4 subsets: ETTh1, ETTh2, ETTm1, ETTm2), Weather, Electricity, and Traffic. For short-term forecasting, we adopt the PeMS which contains four public traffic network datasets (PEMS03, PEMS04, PEMS07, PEMS08). More information is summarized in Table 7.

Table 7: **Summary of datasets**. Forecastability is one minus the entropy of Fourier domain.

| Tasks | Datasets | Dim | Series Length | Dataset Size | Frequency | Forecastability | Information |
|-------|----------|-----|---------------|--------------|-----------|-----------------|-------------|
| Long-term Forecasting | ETTm1 | 7 | $\{96, 192, 336, 720\}$ | $\{34465, 11521, 11521\}$ | 15min | 0.46 | Temperature |
| | ETTm2 | 7 | $\{96, 192, 336, 720\}$ | $\{34465, 11521, 11521\}$ | 15min | 0.46 | Temperature |
| | ETTh1 | 7 | $\{96, 192, 336, 720\}$ | $\{34465, 11521, 11521\}$ | 15min | 0.46 | Temperature |
| | ETTh2 | 7 | $\{96, 192, 336, 720\}$ | $\{34465, 11521, 11521\}$ | 15min | 0.46 | Temperature |
| | Electricity | 321 | $\{96, 192, 336, 720\}$ | $\{18317, 2633, 5261\}$ | Hourly | 0.77 | Electricity |
| | Traffic | 862 | $\{96, 192, 336, 720\}$ | $\{12185, 1757, 3509\}$ | Hourly | 0.68 | Transportation |
| | Weather | 21 | $\{96, 192, 336, 720\}$ | $\{34465, 11521, 11521\}$ | 15min | 0.46 | Weather |
| | Exchange | 8 | $\{96, 192, 336, 720\}$ | $\{5120, 665, 1422\}$ | Daily | 0.46 | Exchange Rate |
| Shot-term Forecasting | PEMS03 | 358 | $\{12, 24, 48, 96\}$ | $\{15617, 5135, 5135\}$ | 5min | 0.65 | Transportation |
| | PEMS04 | 307 | $\{12, 24, 48, 96\}$ | $\{10172, 3375, 3375\}$ | 5min | 0.45 | Transportation |
| | PEMS07 | 883 | $\{12, 24, 48, 96\}$ | $\{16911, 5622, 5622\}$ | 5min | 0.58 | Transportation |
| | PEMS08 | 170 | $\{12, 24, 48, 96\}$ | $\{10690, 3548, 265\}$ | 5min | 0.52 | Transportation |

Table 8: Experiment configuration of UniFy. All the experiments use the ADAM optimizer with the default hyperparameter configuration for $(\beta_1, \beta_2) = (0.9, 0.999)$.

| Datasets / Configurations | Model Hyper-parameter | | Training Config | | |
|---|---|---|---|---|---|
| | Decomposition Round Num from Equ. 2 | $d_{model}$ | LR | Loss | Batch Size |
| ETTh1 | 2 | 256 | $10^{-2}$ | MSE | 16 |
| ETTh2 | 2 | 256 | $10^{-2}$ | MSE | 16 |
| ETTm1 | 3 | 256 | $10^{-2}$ | MSE | 16 |
| ETTm2 | 3 | 128 | $10^{-2}$ | MSE | 32 |
| Weather | 2 | 128 | $10^{-2}$ | MSE | 16 |
| Electricity | 2 | 128 | $10^{-2}$ | MSE | 8 |
| Traffic | 2 | 128 | $10^{-2}$ | MSE | 8 |
| PEMS | 2 | 128 | $10^{-2}$ | MSE | 8 |

Table 9: The GPU memory (MiB) and speed (running time, s/iter) of each model. All experiments are conducted with an input length of 96, batch size of 8, and hidden dimension $d_{model} = 128$.

| Series Length / Models | 96 | | 192 | | 384 | | 768 | | 1536 | | 3072 | |
|---|---|---|---|---|---|---|---|---|---|---|---|---|
| | Mem | Speed | Mem | Speed | Mem | Speed | Mem | Speed | Mem | Speed | Mem | Speed |
| UniFy(Ours) | 357 | 0.015 | 359 | 0.019 | 379 | 0.019 | 381 | 0.021 | 387 | 0.016 | 433 | 0.017 |
| CycleNet(2024) | 349 | 0.026 | 349 | 0.026 | 351 | 0.027 | 369 | 0.033 | 369 | 0.033 | 391 | 0.038 |
| TimeMixer(2024) | 463 | 0.060 | 575 | 0.057 | 813 | 0.063 | 1247 | 0.056 | 2123 | 0.095 | 4011 | 0.268 |
| HDMixer(2024) | 359 | 0.029 | 375 | 0.029 | 393 | 0.026 | 457 | 0.019 | 545 | 0.023 | 775 | 0.032 |
| SDformer(2024) | 375 | 0.169 | 375 | 0.181 | 375 | 0.169 | 377 | 0.189 | 397 | 0.256 | 413 | 0.475 |
| PatchTST(2023) | 357 | 0.039 | 363 | 0.036 | 379 | 0.041 | 589 | 0.034 | 609 | 0.034 | 1237 | 0.035 |
| FreTS(2023) | 453 | 0.021 | 549 | 0.020 | 771 | 0.019 | 1153 | 0.020 | 1961 | 0.029 | 3619 | 0.049 |
| TimesNet(2023) | 1425 | 0.576 | 1721 | 0.736 | 2455 | 1.045 | 3759 | 1.816 | 6543 | 2.889 | 12173 | 5.413 |
| DLinear(2023) | 347 | 0.049 | 349 | 0.056 | 351 | 0.084 | 355 | 0.062 | 373 | 0.071 | 387 | 0.081 |
| Autoformer(2021) | 735 | 0.166 | 899 | 0.164 | 1253 | 0.167 | 1973 | 0.165 | 3577 | 0.219 | 7093 | 0.288 |

**Metric details** We adopt the mean square error (MSE) and mean absolute error (MAE) for all experiments:

$$\text{MSE} = \frac{1}{n} \sum_{i=1}^{n} (Y_i - \hat{Y}_i)^2, \quad \text{MAE} = \frac{1}{n} \sum_{i=1}^{n} |Y_i - \hat{Y}_i| \tag{18}$$

where $\hat{Y}_i$ is the predicted value of the $i$-th channel, and $Y_i$ is the ground truth value.

**Experienment details** All experiments were run three times, implemented in Pytorch, detailed model configuration information is presented in Table 8.

# E    EFFICIENCY ANALYSIS

In the main text, we have plotted the Model efficiency comparison in Fig. 5. Here, we further report the quantitative results in Table 9. Notably, UniFy demonstrates a remarkable efficiency advantage over Transformer-based models such as PatchTST, SDformer, and Autoformer. Moreover, UniFy consistently outperforms most linear models, including CycleNet, TimeMixer, and HDMixer. Across input sequence lengths ranging from 96 to 3072, UniFy achieves superior efficiency, highlighting its strong potential for applications that require scalable and long-horizon time series forecasting.

# F    SIGNIFICANT EXPERIMENT

In this paper, all experiments are repeated three times. We report the standard deviation of our model and the second-best model, together with the results of statistical significance tests, in Table 10 and Table 11.

Table 10: Standard deviation and statistical tests for our UniFy method and second-best method (Cycle-Net) on ETT, Weather, Exchange, Electricity and Traffic datasets. All experiments are conducted with an input length of 96, output length of 96, batch size of 8, and hidden dimension $d_{\text{model}} = 128$.

| Model | UniFy (Ours) | | CycleNet (2024) | | Confidence Interval |
|---|---|---|---|---|---|
| Dataset | MSE | MAE | MSE | MAE | |
| ETTm1 | $0.317 \pm 0.095$ | $0.356 \pm 0.013$ | $0.326 \pm 0.003$ | $0.364 \pm 0.003$ | 99% |
| ETTm2 | $0.170 \pm 0.064$ | $0.253 \pm 0.000$ | $0.170 \pm 0.004$ | $0.251 \pm 0.002$ | 99% |
| ETTh1 | $0.372 \pm 0.010$ | $0.392 \pm 0.102$ | $0.384 \pm 0.026$ | $0.400 \pm 0.013$ | 99% |
| ETTh2 | $0.286 \pm 0.342$ | $0.338 \pm 0.213$ | $0.298 \pm 0.018$ | $0.341 \pm 0.015$ | 99% |
| ECL | $0.145 \pm 0.032$ | $0.241 \pm 0.051$ | $0.143 \pm 0.003$ | $0.239 \pm 0.005$ | 99% |
| Traffic | $0.447 \pm 0.057$ | $0.308 \pm 0.088$ | $0.470 \pm 0.020$ | $0.304 \pm 0.009$ | 99% |
| Weather | $0.155 \pm 0.032$ | $0.210 \pm 0.089$ | $0.172 \pm 0.003$ | $0.217 \pm 0.004$ | 99% |
| Exchange | $0.081 \pm 0.000$ | $0.199 \pm 0.017$ | $0.087 \pm 0.008$ | $0.205 \pm 0.010$ | 99% |

Table 11: Standard deviation and statistical tests for our UniFy method and second-best method (CycleNet) on PEMS datasets. All experiments are conducted with an input length of 96, output length of 96, batch size of 8, and hidden dimension $d_{\text{model}} = 128$.

| Model | UniFy (Ours) | | CycleNet (2024) | | Confidence Interval |
|---|---|---|---|---|---|
| Dataset | MSE | MAE | MSE | MAE | |
| PEMS03 | $0.070 \pm 0.012$ | $0.176 \pm 0.019$ | $0.091 \pm 0.008$ | $0.210 \pm 0.015$ | 99% |
| PEMS04 | $0.072 \pm 0.005$ | $0.175 \pm 0.006$ | $0.080 \pm 0.003$ | $0.189 \pm 0.004$ | 99% |
| PEMS07 | $0.059 \pm 0.001$ | $0.159 \pm 0.003$ | $0.069 \pm 0.058$ | $0.164 \pm 0.003$ | 99% |
| PEMS08 | $0.071 \pm 0.009$ | $0.178 \pm 0.011$ | $0.084 \pm 0.001$ | $0.189 \pm 0.006$ | 99% |

## G ABLATION WITH STD AND TOPK DECOMPOSITION

In the original design of UNIFY, we employ multi-round adaptive frequency selection (AFS) to decompose the input sequence into multiple frequency subspaces. Similar decomposition strategies can also be found in classical methods such as standard deviation based splitting (STD) and TopK truncation. However, both STD and TopK only divide the sequence into two parts. To enable a fair comparison, we set the number of decomposition rounds in UNIFY to one, so that it also produces a two-part decomposition. We then replace AFS with STD and TopK to conduct ablation studies.

Formally, let $X \in \mathbb{R}^{B \times T \times C}$ denote the input sequence and $\mathcal{F}(X)$ its Fourier transform.

**STD.** STD (Seasonal–Trend Decomposition) separates the original series into two main component. STD first extracts a trend by a centered moving average with window $w = 2k + 1$:

$$T_t = \frac{1}{w} \sum_{i=-k}^{k} x_{t+i} \quad \text{(with boundary handling)}, \tag{19}$$

and then forms a two-way split by taking the remainder as the residual:

$$x_t = T_t + R_t, \qquad R_t = x_t - T_t. \tag{20}$$

**TopK.** TopK first transforms the sequence into the frequency domain, selects the $K$ largest-magnitude frequency bins to reconstruct the main signal, and regards the rest as residual:

$$x_t = x_t^{(\text{TopK-main})} + r_t^{(\text{TopK})}. \tag{}$$

**Results.** As shown in Table 24, the original UniFy consistently achieves the best performance, which demonstrates the effectiveness of the proposed AFS mechanism within the overall framework.

Table 12: Comparison between AFS and other decomposition methods within our UniFy framework on ETT, Weather, Exchange, Electricity, and Traffic datasets. All experiments are conducted with an input length of 96, output length of 96, batch size of 8, and hidden dimension $d_{\mathrm{model}} = 128$. Top-$k = 4$.

| Decomposition Method | UniFy | | STD | | Top-$k$ | |
|:---:|:---:|:---:|:---:|:---:|:---:|:---:|
| Dataset | MSE | MAE | MSE | MAE | MSE | MAE |
| ETTm1 | 0.323 | 0.359 | 0.343 | 0.372 | 0.339 | 0.375 |
| ETTm2 | 0.176 | 0.258 | 0.181 | 0.268 | 0.178 | 0.259 |
| ETTh1 | 0.384 | 0.399 | 0.412 | 0.420 | 0.393 | 0.408 |
| ETTh2 | 0.299 | 0.348 | 0.315 | 0.367 | 0.305 | 0.353 |
| ECL | 0.169 | 0.261 | 0.179 | 0.270 | 0.180 | 0.271 |
| Traffic | 0.563 | 0.369 | 0.588 | 0.394 | 0.573 | 0.380 |
| Weather | 0.171 | 0.218 | 0.172 | 0.218 | 0.189 | 0.230 |
| Exchange | 0.087 | 0.208 | 0.092 | 0.214 | 0.112 | 0.232 |

## H   ABLATION WITH ADAPTIVE SELECTOR

In each decomposition round of UniFy, we adopt an adaptive mask mechanism to select frequency points. The model learns data-dependent masks so that different rounds can focus on different parts of the spectrum. To further validate the effectiveness of this adaptive mechanism, we replace it with a deterministic TopK selection in each round, where the $K$ largest-magnitude frequency bins are retained.

**Original AFS.** The adaptive frequency selection in round $r$ is formulated as

$$\widetilde{P}_k^{(r)} = \mathbf{M}_k^{(r)} \odot P_k, \qquad k = 0, 1, \ldots, H - 1, \qquad r \in \{1, \ldots, R\}, \tag{21}$$

where $P_k$ denotes the input spectrum at frequency bin $k$, $\mathbf{M}_k^{(r)} \in [0, 1]$ is a learnable mask in round $r$, and $\odot$ represents element-wise multiplication across channels. After each round, the residual spectrum is updated as

$$P \leftarrow P - \widetilde{P}^{(r)}. \tag{22}$$

**Round-wise TopK.** In the TopK variant, instead of using a learnable mask, we select the $K$ bins with the largest magnitude in each round:

$$\widetilde{P}_k^{(r)} = \mathbb{I}\big[k \in \mathrm{TopK}\big(|P|, K\big)\big] \odot P_k, \tag{23}$$

where $\mathbb{I}[\cdot]$ is the indicator function. The residual spectrum is updated in the same way as in AFS.

**Results.** As shown in Table 13, the adaptive AFS consistently outperforms the TopK-based selection. To make the comparison convincing, we varied $K$ from 2 to 16. In all cases, AFS achieved the best results, demonstrating that the proposed adaptive masking mechanism is more effective than fixed TopK selection in capturing task-relevant frequency components.

## I   ALTERNATIVE DEPENDENT MODELING METHODS

In the original design of UNIFY, the Independent Linear Mapping (ILM) component models each decomposed frequency subspace with a simple independent linear layer. To further investigate whether more sophisticated architectures could provide benefits, we replaced the linear layer in ILM with alternative modules that play a similar role in sequence modeling, namely: (i) Transformer layers, (ii) convolutional layers (CNNs), and (iii) graph neural network layers (GNNs).

These substitutions were designed to assess whether the performance advantage of ILM stems merely from its lightweight structure or from its suitability for handling frequency-decomposed subspaces.

Table 13: Comparison between AFS and Top-$k$ selection within our UniFy framework on ETT, Weather, Exchange, Electricity and Traffic datasets. All experiments are conducted with an input length of 96, output length of 96, batch size of 8, Decomposition Round Num of 2 (from Equ. 2) and hidden dimension $d_{\text{model}} = 128$.

| Decomposition Scale | UniFy | | Top-$k$=2 | | Top-$k$=4 | | Top-$k$=8 | | Top-$k$=16 | |
|---|---|---|---|---|---|---|---|---|---|---|
| **Dataset** | **MSE** | **MAE** | **MSE** | **MAE** | **MSE** | **MAE** | **MSE** | **MAE** | **MSE** | **MAE** |
| ETTm1 | 0.323 | 0.359 | 0.412 | 0.418 | 0.421 | 0.426 | 0.406 | 0.415 | 0.350 | 0. 373 |
| ETTm2 | 0.176 | 0.258 | 0.181 | 0.263 | 0.183 | 0.266 | 0.204 | 0.282 | 0.183 | 0.266 |
| ETTh1 | 0.384 | 0.399 | 0.391 | 0.402 | 0.382 | 0.397 | 0.381 | 0.398 | 0.379 | 0.395 |
| ETTh2 | 0.299 | 0.348 | 0.311 | 0.358 | 0.439 | 0.442 | 0.305 | 0.351 | 0.302 | 0.353 |
| ECL | 0.169 | 0.261 | 0.183 | 0.278 | 0.183 | 0.277 | 0.155 | 0.250 | 0.156 | 0.250 |
| Traffic | 0.563 | 0.369 | 0.463 | 0.317 | 0.472 | 0.319 | 0.477 | 0.318 | 0.481 | 0.321 |
| Weather | 0.171 | 0.218 | 13.630 | 1.797 | 0.797 | 0.431 | 2.684 | 0.795 | 10.572 | 1.361 |
| Exchange | 0.087 | 0.208 | 0.103 | 0.220 | 0.099 | 0.220 | 0.102 | 0.219 | 0.105 | 0.224 |

Table 14: Comparison between alternative dependent modeling methods within our UniFy framework on ETT, Weather, Exchange, Electricity and Traffic datasets. All experiments are conducted with an input length of 96, output length of 96, batch size of 8, Decomposition Round Num of 2 (from Equ. 2) and hidden dimension $d_{\text{model}} = 128$.

| Model | Linear (Ours) | | Transformer | | CNN | | RNN | |
|---|---|---|---|---|---|---|---|---|
| **Dataset** | **MSE** | **MAE** | **MSE** | **MAE** | **MSE** | **MAE** | **MSE** | **MAE** |
| ETTm1 | 0.471 | 0.447 | 0.756 | 0.573 | 0.384 | 0.402 | 0.612 | 0.511 |
| ETTm2 | 0.177 | 0.260 | 0.238 | 0.312 | 0.323 | 0.380 | 0.196 | 0.283 |
| ETTh1 | 0.383 | 0.396 | 0.702 | 0.562 | 0.762 | 0.573 | 0.401 | 0.416 |
| ETTh2 | 0.307 | 0.355 | 0.399 | 0.416 | 0.310 | 0.353 | 0.386 | 0.409 |
| ECL | 0.147 | 0.244 | 0.886 | 0.771 | 0.187 | 0.278 | 0.188 | 0.273 |
| Traffic | 0.474 | 0.310 | 0.578 | 0.396 | 0.504 | 0.321 | 0.560 | 0.369 |
| Weather | 0.162 | 0.209 | 0.213 | 0.266 | 0.204 | 0.248 | 0.198 | 0.244 |
| Exchange | 0.090 | 0.207 | 0.152 | 0.278 | 0.088 | 0.207 | 0.097 | 0.217 |
| Avg | **0.276** | **0.304** | 0.491 | 0.447 | 0.345 | 0.345 | 0.330 | 0.340 |

As reported in Table 14, the original ILM with independent linear layers consistently achieves the best performance across benchmarks, confirming that the linear formulation is not only efficient but also more effective than Transformer-, CNN-, or GNN-based alternatives in the UNIFY framework.

## J HYPERPARAMETER SENSITIVITY

**Hyperparameter Sensitivity on Training Parameters** We evaluate the hyperparameter sensitivity of UniFy with respect to the following factors: the hidden dimension and the batch size. The results are shown in the Fig.8. We found that UniFy demonstrates strong robustness across various datasets.

**Hyperparameter Sensitivity on Model Parameters** We further evaluate the number of decomposition round from Equ. 2. As shown in Table 15, we can find that in general, increasing the number of layers (L) will bring improvements across different prediction lengths. Therefore, we set to 2 to trade off efficiency and performance.

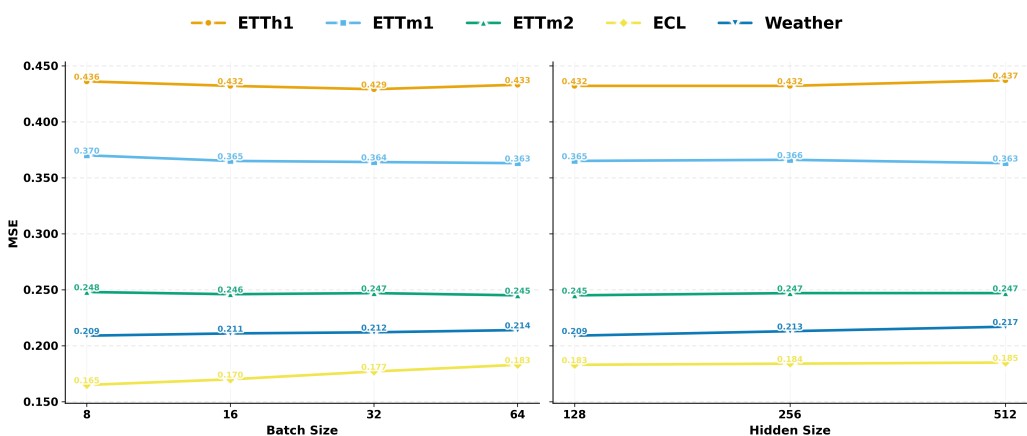

Figure 8: Hyperparameter Sensitivity on Training Parameters.

Table 15: Decomposition Round Num (from Equ. 2) Sensitivity analysis for our UniFy method on ETT, Weather, Exchange, Electricity and Traffic datasets. All experiments are conducted with an input length of 96, output length of 96, batch size of 8 and hidden dimension $d_{\mathrm{model}} = 128$.

| Decomposition Round Num (from Equ. 2) | 1 | | 2 | | 3 | | 4 | | 5 | |
|---|---|---|---|---|---|---|---|---|---|---|
| Dataset | MSE | MAE | MSE | MAE | MSE | MAE | MSE | MAE | MSE | MAE |
| ETTm1 | 0.323 | 0.359 | 0.323 | 0.362 | 0.324 | 0.363 | 0.323 | 0.361 | 0.323 | 0. 360 |
| ETTm2 | 0.176 | 0.258 | 0.182 | 0.261 | 0.175 | 0.256 | 0.195 | 0.280 | 0.176 | 0.260 |
| ETTh1 | 0.384 | 0.399 | 0.387 | 0.401 | 0.383 | 0.396 | 0.396 | 0.407 | 0.391 | 0.403 |
| ETTh2 | 0.299 | 0.348 | 0.299 | 0.348 | 0.302 | 0.352 | 0.296 | 0.347 | 0.302 | 0.353 |
| ECL | 0.169 | 0.261 | 0.172 | 0.263 | 0.178 | 0.267 | 0.184 | 0.271 | 0.180 | 0.268 |
| Traffic | 0.563 | 0.369 | 0.575 | 0.374 | 0.540 | 0.354 | 0.596 | 0.380 | 0.622 | 0.391 |
| Weather | 0.171 | 0.218 | 0.251 | 0.288 | 0.301 | 0.302 | 0.220 | 0.254 | 0.185 | 0.230 |
| Exchange | 0.087 | 0.208 | 0.094 | 0.213 | 0.087 | 0.206 | 0.088 | 0.206 | 0.090 | 0.207 |

## K FORECASTING VISUALIZATION

To further illustrate the limitations of existing periodic modeling approaches, we visualize the predictions of DLinear, CycleNet, and our proposed UniFy on Fig. 9.

The first row depicts a case of underfitting periodic components. Both DLinear and CycleNet are able to capture parts of the first two periodic intervals but largely ignore the third one. This aligns with our earlier analysis: when all frequencies are learned in a shared space, dominant components suppress weaker yet informative frequencies, resulting in insufficient modeling of non-stationary periodic details.

The second row presents an example of overfitting. In this case, the underlying periodic structure is sparse, with a long interval lacking clear cycles. However, both baselines, after learning the scattered periodic segments, incorrectly infer spurious cycles in the middle flat region, producing false oscillations. This phenomenon reflects the other side of frequency competition: when multiple frequencies compete without proper separation, the model may misinterpret noise or weak signals as true cycles, leading to over-responsiveness in smooth regions.

In contrast, UniFy mitigates both issues by decomposing the frequency domain into multiple subspaces and modeling them independently, allowing it to adapt to non-stationary periodic variations while avoiding unnecessary periodic artifacts.

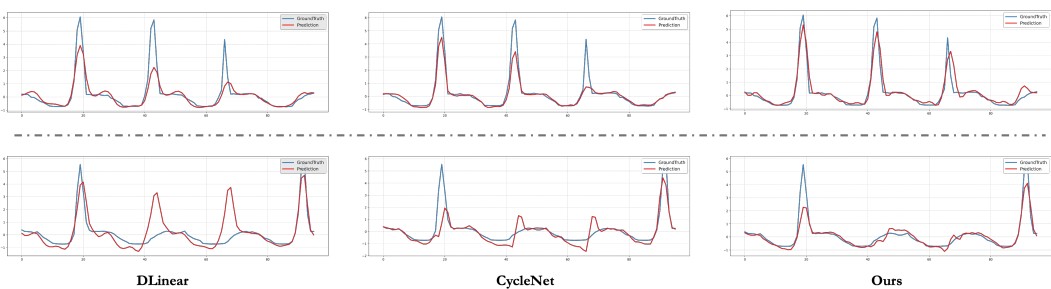

Figure 9: Model forecasting result visualization on traffic dataset with 96 output length.

## L  STATISTICAL-BASED ATTRIBUTION STUDY FOR FREQUENCY DECOMPOSITION

In the main text (Section.4), we have already conducted a gradient-based attribution study for frequency decomposition. To further strengthen the argument, we extend the analysis from a statistical perspective, examining whether our decomposition module indeed allocates frequency components of different energy levels into independent subspaces, thereby alleviating frequency competition.

**Decomposition.**  In the AFS module of UniFy, the original sequence is decomposed through multiple rounds of spectral masking, producing $d$ complementary frequency components. This design explicitly partitions the spectrum with the goal of mitigating cross-band competition. Ideally, each subspace should capture distinct frequency information. If this holds, the contribution distribution across decomposition scales should be consistent with the spectral energy distribution of the input signal.

**Ablation-based attribution.**  During evaluation, for each mini-batch we iteratively set the projected output of one component to zero (zero ablation) while keeping all other components unchanged, and recompute the prediction. Let $\hat{y}$ denote the baseline prediction with all components and $\hat{y}_{\setminus k}$ the prediction with the $k$-th component removed. The degradation in performance is quantified as:

$$\Delta\mathrm{MSE}_k \;=\; \max\Big\{\,\mathrm{MSE}\big(y, \hat{y}_{\setminus k}\big) \;-\; \mathrm{MSE}\big(y, \hat{y}\big),\; 0\,\Big\}. \tag{24}$$

A larger $\Delta\mathrm{MSE}_k$ indicates that the $k$-th component plays a more critical role in prediction.

**Spectral energy of the input.**  In parallel, we construct an energy-based reference from the spectrum of the input sequence. Specifically, we identify the $(d-1)$ frequency intervals with the highest magnitudes[1], while the residual frequencies are grouped into the $d$-th interval. This yields a set of energy terms $\{E_k\}_{k=1}^d$ aligned with the decomposition scale:

$$E_k \;=\; \frac{1}{T} \sum_{f \in \mathcal{I}_k} \big|X(f)\big|^2, \qquad \mathcal{I}_1, \ldots, \mathcal{I}_d \text{ are disjoint and cover the entire spectrum.} \tag{25}$$

The spectral energy thus serves as a reference for the potential importance of each component.

**Batch-wise aggregation and correlation.**  For each batch, we compute the sample averages of $\Delta\mathrm{MSE}_k$ and $E_k$, and accumulate them across the evaluation set, yielding global means $\overline{\Delta\mathrm{MSE}}_k$ and $\overline{E}_k$. We then assess their linear relationship by reporting the Pearson correlation coefficient and statistical significance:

$$r, p \;=\; \texttt{pearsonr}\Big(\{\overline{E}_k\}_{k=1}^d,\; \{\overline{\Delta\mathrm{MSE}}_k\}_{k=1}^d\Big). \tag{26}$$

This analysis quantifies the consistency between the spectral energy distribution and the attribution-based importance distribution.

---

[1]Each interval is defined by a dominant spectral peak and its neighboring bins; the remaining frequencies are aggregated as the residual part.

Figure 10: Results with decomposition scale $d = 2$. The first row reports UniFy's performance on ETTh1 with prediction lengths of 96, 192, 336, and 720. The second row corresponds to ETTh2, the third row to ETTm1, and the fourth row to ETTm2.

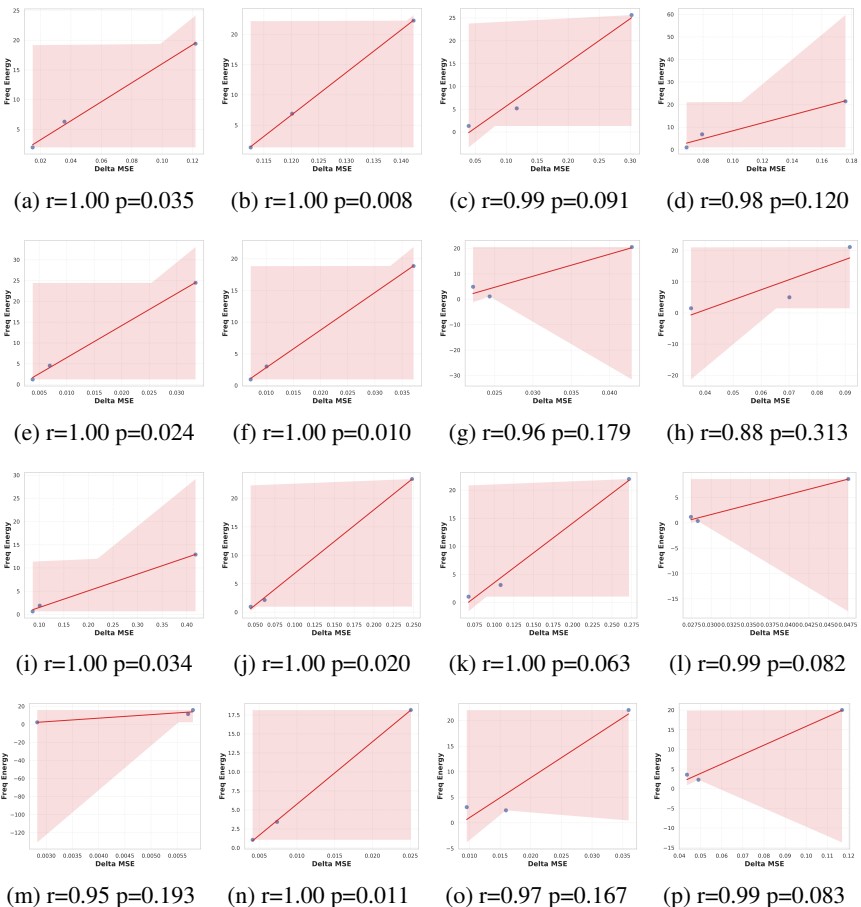

(a) r=1.00 p=0.035    (b) r=1.00 p=0.008    (c) r=0.99 p=0.091    (d) r=0.98 p=0.120

(e) r=1.00 p=0.024    (f) r=1.00 p=0.010    (g) r=0.96 p=0.179    (h) r=0.88 p=0.313

(i) r=1.00 p=0.034    (j) r=1.00 p=0.020    (k) r=1.00 p=0.063    (l) r=0.99 p=0.082

(m) r=0.95 p=0.193    (n) r=1.00 p=0.011    (o) r=0.97 p=0.167    (p) r=0.99 p=0.083

**Interpretation.** If $r > 0$ and statistically significant (small $p$ value), it indicates that higher-energy frequency bands tend to exhibit greater attribution importance, suggesting that the decomposition scales align with energy-dominant spectral components and thus support the hypothesis of frequency competition alleviation. Conversely, a weak correlation would imply that predictive importance does not simply follow energy magnitude, and that the benefits of decomposition may arise from structural factors beyond energy separation. As shown in Fig. 10 and Fig. 11, the proposed analysis yields the expected outcome. In particular, we observe that the removal of high-energy components consistently leads to larger increases in prediction error, while low-energy residual components contribute less critically. Moreover, when comparing decomposition with $d = 2$ scales and $d = 6$ scales, both settings demonstrate stable prediction performance and a clear alignment between spectral energy and attribution importance, further supporting the validity of our frequency decomposition design.

Figure 11: Results with decomposition scale $d = 6$. The first row reports UniFy's performance on ETTh1 with prediction lengths of 96, 192, 336, and 720. The second row corresponds to ETTh2, the third row to ETTm1, and the fourth row to ETTm2.

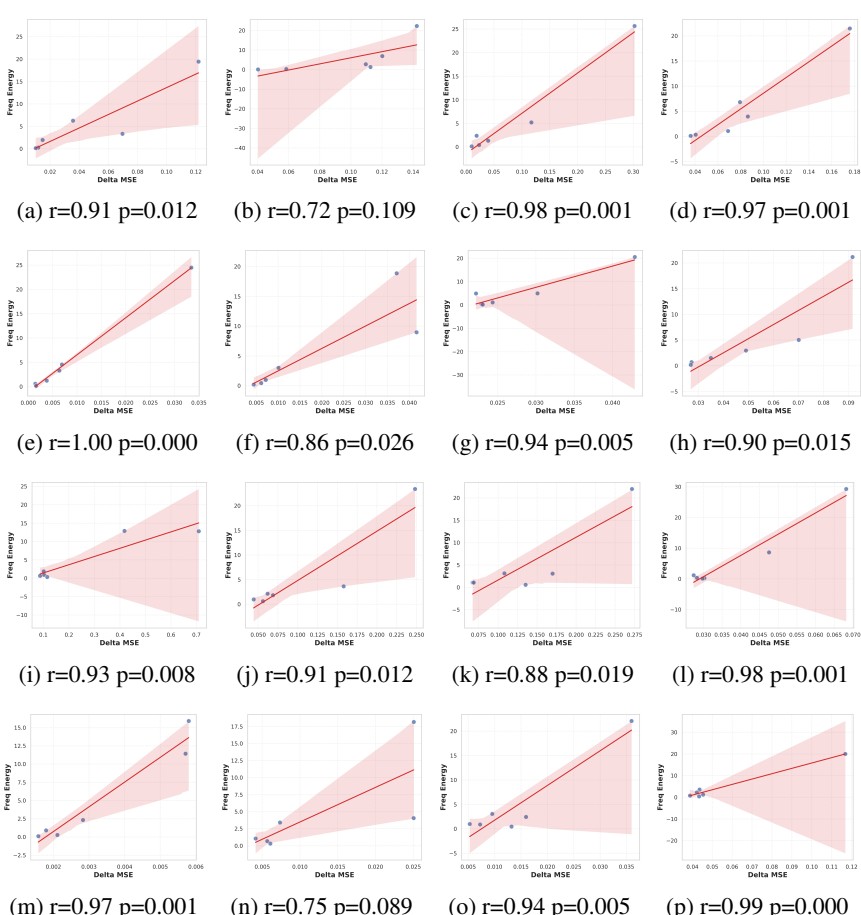

(a) r=0.91 p=0.012  (b) r=0.72 p=0.109  (c) r=0.98 p=0.001  (d) r=0.97 p=0.001

(e) r=1.00 p=0.000  (f) r=0.86 p=0.026  (g) r=0.94 p=0.005  (h) r=0.90 p=0.015

(i) r=0.93 p=0.008  (j) r=0.91 p=0.012  (k) r=0.88 p=0.019  (l) r=0.98 p=0.001

(m) r=0.97 p=0.001  (n) r=0.75 p=0.089  (o) r=0.94 p=0.005  (p) r=0.99 p=0.000

# M    COMPLETE RESULTS OF GRADIENT-BASED ATTRIBUTION

In Section 4, we only presented a subset of the frequency attribution visualizations. In this section, we provide the complete set of frequency attribution visualizations to further substantiate and enhance the credibility of our analysis framework. The full results are shown in Fig.12, Fig.13 and Fig.16.

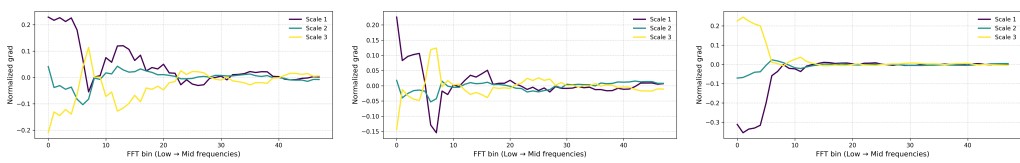

Figure 12: Gradient-based attribution for frequency decomposition results on ETTh2.

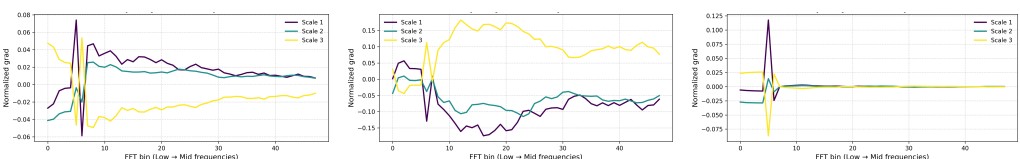

Figure 13: Gradient-based attribution for frequency decomposition results on ETTm1.

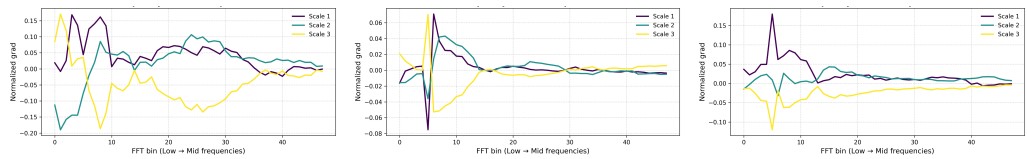

Figure 14: Gradient-based attribution for frequency decomposition results on ETTm2.

# N    FULL RESULTS

While the main text reports only the averaged results for clarity, the complete experimental results are provided in Table 16, Table 17, Table 18, and Table 19, covering long-term forecasting, short-term forecasting, few-shot learning, and zero-shot evaluation, respectively.

# O    FULL RESULTS FOR SUPPLEMENTARY BASELINES

We have introduced additional baselines beyond those presented in the main text to provide a more comprehensive evaluation of the effectiveness of UniFy. The new baselines include models that perform frequency-domain analysis, each approaching time series modeling from different perspectives within the frequency domain.

The complete experimental results are provided in Table 20, Table 21, Table 22, and Table 23, which cover long-term forecasting, short-term forecasting, few-shot learning, and zero-shot evaluation, respectively.

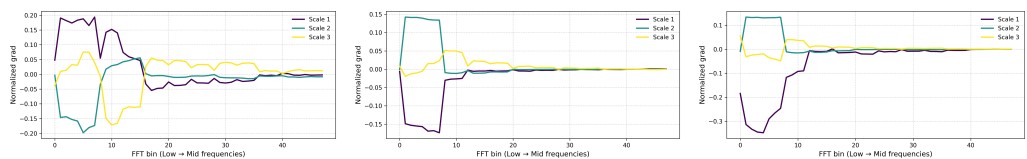

Figure 15: Gradient-based attribution for frequency decomposition results on ECL.

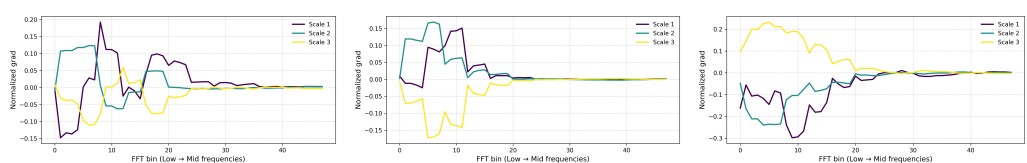

Figure 16: Gradient-based attribution for frequency decomposition results on Traffic.

Table 16: **Long-Term Forecasting Full Results**. A lower MSE or MAE indicates a better prediction. *Avg* is averaged from all four prediction lengths, that is {96, 192, 336, 720}. The best results are highlighted in **bold**.

| Models | | UniFy (Ours) | | CycleNet (2024) | | TimeMixer (2024) | | HDMixer (2024) | | SDformer (2024) | | PatchTST (2023) | | FreTS (2023) | | TimesNet (2023) | | DLinear (2023) | | Autoformer (2021) | |
|---|---|---|---|---|---|---|---|---|---|---|---|---|---|---|---|---|---|---|---|---|---|
| Metric | | MSE | MAE | MSE | MAE | MSE | MAE | MSE | MAE | MSE | MAE | MSE | MAE | MSE | MAE | MSE | MAE | MSE | MAE | MSE | MAE |
| ETTm1 | 96 | **0.315** | **0.355** | 0.318 | 0.360 | 0.326 | 0.362 | 0.337 | 0.367 | 0.349 | 0.380 | 0.329 | 0.367 | 0.338 | 0.375 | 0.338 | 0.375 | 0.355 | 0.376 | 0.505 | 0.475 |
| | 192 | **0.360** | **0.379** | 0.362 | 0.382 | 0.370 | 0.386 | 0.377 | 0.385 | 0.386 | 0.397 | 0.367 | 0.385 | 0.381 | 0.397 | 0.374 | 0.387 | 0.391 | 0.392 | 0.553 | 0.496 |
| | 336 | **0.390** | **0.402** | 0.393 | 0.404 | 0.393 | 0.406 | 0.403 | 0.407 | 0.419 | 0.419 | 0.399 | 0.410 | 0.420 | 0.423 | 0.410 | 0.411 | 0.424 | 0.415 | 0.621 | 0.537 |
| | 720 | 0.456 | **0.438** | **0.448** | 0.441 | 0.459 | 0.446 | 0.466 | 0.443 | 0.483 | 0.453 | 0.454 | 0.439 | 0.480 | 0.458 | 0.478 | 0.450 | 0.487 | 0.450 | 0.671 | 0.561 |
| | Avg | **0.380** | **0.394** | 0.380 | 0.397 | 0.387 | 0.400 | 0.396 | 0.400 | 0.409 | 0.412 | 0.387 | 0.400 | 0.405 | 0.413 | 0.400 | 0.406 | 0.414 | 0.408 | 0.588 | 0.517 |
| ETTm2 | 96 | **0.169** | **0.253** | 0.176 | 0.260 | 0.177 | 0.259 | 0.182 | 0.265 | 0.184 | 0.267 | 0.175 | 0.259 | 0.186 | 0.276 | 0.187 | 0.267 | 0.182 | 0.265 | 0.255 | 0.339 |
| | 192 | **0.234** | 0.295 | 0.235 | **0.293** | 0.239 | 0.300 | 0.245 | 0.306 | 0.250 | 0.309 | 0.241 | 0.302 | 0.259 | 0.324 | 0.249 | 0.309 | 0.246 | 0.304 | 0.281 | 0.340 |
| | 336 | **0.292** | **0.334** | 0.304 | 0.341 | 0.298 | 0.339 | 0.303 | 0.344 | 0.312 | 0.348 | 0.305 | 0.343 | 0.347 | 0.385 | 0.321 | 0.351 | 0.307 | 0.342 | 0.339 | 0.372 |
| | 720 | **0.393** | 0.393 | 0.398 | **0.392** | 0.399 | 0.398 | 0.411 | 0.405 | 0.410 | 0.403 | 0.402 | 0.400 | 0.554 | 0.508 | 0.408 | 0.403 | 0.407 | 0.398 | 0.433 | 0.432 |
| | Avg | **0.272** | **0.319** | 0.278 | 0.322 | 0.278 | 0.324 | 0.285 | 0.330 | 0.289 | 0.332 | 0.281 | 0.326 | 0.337 | 0.373 | 0.291 | 0.333 | 0.286 | 0.327 | 0.327 | 0.371 |
| ETTh1 | 96 | **0.372** | **0.394** | 0.378 | 0.397 | 0.379 | 0.400 | 0.396 | 0.406 | 0.403 | 0.416 | 0.414 | 0.419 | 0.394 | 0.406 | 0.384 | 0.402 | 0.392 | 0.409 | 0.449 | 0.459 |
| | 192 | **0.426** | **0.423** | 0.440 | 0.431 | 0.440 | 0.431 | 0.443 | 0.430 | 0.455 | 0.447 | 0.460 | 0.445 | 0.451 | 0.442 | 0.436 | 0.429 | 0.443 | 0.439 | 0.500 | 0.482 |
| | 336 | **0.463** | **0.435** | 0.495 | 0.453 | 0.486 | 0.452 | 0.489 | 0.452 | 0.498 | 0.469 | 0.501 | 0.466 | 0.510 | 0.479 | 0.491 | 0.469 | 0.480 | 0.459 | 0.521 | 0.496 |
| | 720 | **0.467** | **0.464** | 0.502 | 0.473 | 0.552 | 0.503 | 0.516 | 0.488 | 0.513 | 0.497 | 0.500 | 0.488 | 0.567 | 0.537 | 0.521 | 0.500 | 0.521 | 0.516 | 0.514 | 0.512 |
| | Avg | **0.432** | **0.429** | 0.454 | 0.439 | 0.464 | 0.447 | 0.461 | 0.444 | 0.467 | 0.457 | 0.469 | 0.455 | 0.481 | 0.466 | 0.458 | 0.450 | 0.459 | 0.456 | 0.496 | 0.487 |
| ETTh2 | 96 | **0.281** | **0.335** | 0.299 | 0.344 | 0.298 | 0.348 | 0.293 | 0.343 | 0.303 | 0.349 | 0.302 | 0.348 | 0.334 | 0.388 | 0.340 | 0.374 | 0.288 | 0.338 | 0.346 | 0.388 |
| | 192 | **0.366** | 0.391 | 0.375 | 0.400 | 0.381 | 0.399 | 0.379 | 0.395 | 0.383 | 0.398 | 0.388 | 0.400 | 0.450 | 0.456 | 0.374 | **0.390** | 0.374 | **0.390** | 0.456 | 0.452 |
| | 336 | **0.386** | **0.413** | 0.426 | 0.436 | 0.395 | 0.419 | 0.418 | 0.429 | 0.433 | 0.432 | 0.426 | 0.433 | 0.461 | 0.467 | 0.452 | 0.452 | 0.415 | 0.426 | 0.482 | 0.486 |
| | 720 | **0.418** | **0.438** | 0.442 | 0.454 | 0.424 | 0.441 | 0.429 | 0.444 | 0.430 | 0.441 | 0.431 | 0.446 | 0.790 | 0.634 | 0.462 | 0.468 | 0.420 | 0.440 | 0.515 | 0.511 |
| | Avg | **0.363** | **0.394** | 0.386 | 0.409 | 0.374 | 0.402 | 0.380 | 0.403 | 0.387 | 0.405 | 0.387 | 0.407 | 0.509 | 0.486 | 0.414 | 0.427 | 0.374 | 0.399 | 0.450 | 0.459 |
| ECL | 96 | 0.142 | 0.237 | **0.141** | **0.234** | 0.155 | 0.247 | 0.197 | 0.284 | 0.182 | 0.279 | 0.181 | 0.270 | 0.183 | 0.265 | 0.168 | 0.272 | 0.201 | 0.281 | 0.201 | 0.317 |
| | 192 | **0.155** | **0.247** | 0.155 | 0.247 | 0.169 | 0.259 | 0.200 | 0.286 | 0.194 | 0.288 | 0.188 | 0.274 | 0.180 | 0.267 | 0.184 | 0.289 | 0.201 | 0.283 | 0.222 | 0.334 |
| | 336 | **0.171** | **0.263** | 0.172 | 0.264 | 0.186 | 0.277 | 0.211 | 0.301 | 0.209 | 0.303 | 0.204 | 0.293 | 0.191 | 0.283 | 0.198 | 0.300 | 0.215 | 0.298 | 0.231 | 0.338 |
| | 720 | 0.215 | 0.302 | **0.210** | **0.296** | 0.227 | 0.312 | 0.254 | 0.335 | 0.243 | 0.331 | 0.246 | 0.324 | 0.228 | 0.322 | 0.220 | 0.320 | 0.257 | 0.331 | 0.254 | 0.361 |
| | Avg | 0.171 | 0.262 | **0.170** | **0.260** | 0.184 | 0.274 | 0.215 | 0.302 | 0.207 | 0.300 | 0.205 | 0.290 | 0.196 | 0.284 | 0.193 | 0.295 | 0.219 | 0.298 | 0.227 | 0.338 |
| Traffic | 96 | **0.441** | 0.302 | 0.459 | 0.297 | 0.462 | **0.285** | 0.619 | 0.405 | 0.512 | 0.357 | 0.544 | 0.359 | 0.556 | 0.372 | 0.593 | 0.321 | 0.649 | 0.389 | 0.613 | 0.388 |
| | 192 | **0.452** | 0.301 | 0.457 | **0.295** | 0.473 | 0.296 | 0.656 | 0.421 | 0.535 | 0.368 | 0.540 | 0.354 | 0.574 | 0.374 | 0.617 | 0.336 | 0.601 | 0.366 | 0.616 | 0.382 |
| | 336 | **0.468** | 0.307 | 0.470 | 0.300 | 0.498 | **0.296** | 0.611 | 0.400 | 0.557 | 0.377 | 0.551 | 0.358 | 0.606 | 0.384 | 0.629 | 0.336 | 0.609 | 0.369 | 0.622 | 0.337 |
| | 720 | 0.508 | 0.332 | **0.502** | 0.314 | 0.506 | **0.313** | 0.699 | 0.442 | 0.606 | 0.401 | 0.586 | 0.375 | 0.664 | 0.414 | 0.640 | 0.350 | 0.647 | 0.387 | 0.660 | 0.408 |
| | Avg | **0.467** | 0.311 | 0.472 | 0.302 | 0.485 | **0.298** | 0.646 | 0.417 | 0.553 | 0.376 | 0.555 | 0.362 | 0.600 | 0.386 | 0.620 | 0.336 | 0.627 | 0.378 | 0.628 | 0.379 |
| Weather | 96 | **0.156** | **0.203** | 0.170 | 0.216 | 0.163 | 0.210 | 0.178 | 0.226 | 0.170 | 0.210 | 0.177 | 0.218 | 0.177 | 0.208 | 0.172 | 0.220 | 0.192 | 0.232 | 0.266 | 0.336 |
| | 192 | **0.206** | 0.248 | 0.222 | 0.259 | 0.210 | 0.253 | 0.227 | 0.264 | 0.225 | 0.258 | 0.225 | 0.259 | 0.219 | **0.247** | 0.240 | 0.261 | 0.240 | 0.271 | 0.307 | 0.367 |
| | 336 | **0.262** | **0.290** | 0.275 | 0.296 | 0.264 | 0.292 | 0.276 | 0.300 | 0.281 | 0.299 | 0.278 | 0.297 | 0.277 | 0.295 | 0.280 | 0.306 | 0.292 | 0.307 | 0.359 | 0.395 |
| | 720 | 0.343 | 0.343 | 0.349 | 0.345 | 0.346 | 0.346 | 0.352 | 0.350 | 0.361 | 0.351 | 0.354 | 0.348 | **0.340** | **0.340** | 0.365 | 0.359 | 0.364 | 0.353 | 0.419 | 0.428 |
| | Avg | **0.242** | **0.271** | 0.254 | 0.279 | 0.246 | 0.275 | 0.258 | 0.285 | 0.265 | 0.285 | 0.265 | 0.285 | 0.253 | 0.273 | 0.259 | 0.287 | 0.272 | 0.291 | 0.338 | 0.382 |
| Exchange | 96 | **0.081** | **0.198** | 0.084 | 0.201 | 0.083 | 0.201 | 0.084 | 0.202 | 0.090 | 0.211 | 0.088 | 0.205 | 0.090 | 0.216 | 0.107 | 0.234 | 0.093 | 0.217 | 0.197 | 0.323 |
| | 192 | **0.172** | **0.293** | 0.184 | 0.304 | 0.177 | 0.298 | 0.174 | 0.296 | 0.190 | 0.313 | 0.176 | 0.299 | 0.192 | 0.322 | 0.226 | 0.344 | 0.184 | 0.307 | 0.300 | 0.369 |
| | 336 | 0.329 | 0.414 | 0.363 | 0.438 | 0.334 | 0.418 | 0.335 | 0.417 | 0.353 | 0.433 | **0.301** | **0.397** | 0.399 | 0.461 | 0.367 | 0.448 | 0.351 | 0.432 | 0.509 | 0.524 |
| | 720 | 0.826 | 0.687 | 0.895 | 0.706 | 0.862 | 0.700 | 0.915 | 0.720 | 0.941 | 0.735 | 0.901 | 0.714 | **0.789** | **0.646** | 0.964 | 0.746 | 0.886 | 0.714 | 1.447 | 0.941 |
| | Avg | **0.352** | **0.398** | 0.382 | 0.412 | 0.364 | 0.404 | 0.377 | 0.409 | 0.394 | 0.423 | 0.367 | 0.404 | 0.368 | 0.411 | 0.416 | 0.443 | 0.379 | 0.418 | 0.613 | 0.539 |
| 1st count | | 67 | | 7 | | 7 | | 0 | | 0 | | 4 | | 4 | | 0 | | 1 | | 0 | |

Table 17: **Short-Term Forecasting Full Results** on the PEMS datasets with multiple variates. A lower MSE or MAE indicates a better prediction. *Avg* is averaged from all four prediction lengths, that is {96, 192, 336, 720}. The best results are highlighted in **bold**.

| Models | | UniFy (Ours) | | CycleNet (2024) | | TimeMixer (2024) | | HDMixer (2024) | | SDformer (2024) | | PatchTST (2023) | | FreTS (2023) | | TimesNet (2023) | | DLinear (2023) | | Autoformer (2021) | |
|---|---|---|---|---|---|---|---|---|---|---|---|---|---|---|---|---|---|---|---|---|---|
| Metric | | MSE | MAE | MSE | MAE | MSE | MAE | MSE | MAE | MSE | MAE | MSE | MAE | MSE | MAE | MSE | MAE | MSE | MAE | MSE | MAE |
| PEMS03 | 12 | **0.068** | **0.174** | 0.070 | 0.179 | 0.068 | 0.180 | 0.094 | 0.219 | 0.088 | 0.201 | 0.099 | 0.216 | 0.080 | 0.189 | 0.085 | 0.192 | 0.126 | 0.236 | 0.272 | 0.385 |
| | 24 | **0.093** | **0.203** | 0.099 | 0.213 | 0.104 | 0.227 | 0.161 | 0.293 | 0.134 | 0.250 | 0.142 | 0.259 | 0.125 | 0.236 | 0.118 | 0.223 | 0.246 | 0.334 | 0.334 | 0.440 |
| | 48 | **0.143** | **0.256** | 0.148 | 0.261 | 0.183 | 0.311 | 0.273 | 0.385 | 0.237 | 0.341 | 0.211 | 0.319 | 0.194 | 0.300 | 0.155 | 0.260 | 0.551 | 0.529 | 1.032 | 0.782 |
| | 96 | **0.218** | **0.314** | 0.330 | 0.404 | 0.249 | 0.369 | 0.481 | 0.519 | 0.427 | 0.477 | 0.269 | 0.370 | 0.271 | 0.362 | 0.228 | 0.317 | 1.057 | 0.787 | 1.031 | 0.796 |
| | Avg | **0.131** | **0.237** | 0.162 | 0.264 | 0.151 | 0.272 | 0.252 | 0.354 | 0.222 | 0.317 | 0.180 | 0.291 | 0.168 | 0.272 | 0.147 | 0.248 | 0.495 | 0.472 | 0.667 | 0.601 |
| PEMS04 | 12 | **0.070** | **0.173** | 0.079 | 0.188 | 0.070 | 0.176 | 0.111 | 0.237 | 0.106 | 0.217 | 0.105 | 0.224 | 0.095 | 0.206 | 0.087 | 0.195 | 0.138 | 0.252 | 0.424 | 0.491 |
| | 24 | 0.086 | 0.194 | 0.103 | 0.219 | **0.080** | **0.190** | 0.180 | 0.306 | 0.165 | 0.276 | 0.153 | 0.275 | 0.140 | 0.255 | 0.103 | 0.215 | 0.258 | 0.348 | 0.459 | 0.509 |
| | 48 | **0.110** | **0.223** | 0.139 | 0.258 | 0.129 | 0.252 | 0.345 | 0.422 | 0.286 | 0.378 | 0.229 | 0.339 | 0.214 | 0.321 | 0.136 | 0.250 | 0.572 | 0.544 | 0.646 | 0.610 |
| | 96 | 0.158 | 0.273 | 0.177 | 0.290 | **0.148** | **0.272** | 0.780 | 0.631 | 0.502 | 0.477 | 0.291 | 0.389 | 0.298 | 0.380 | 0.190 | 0.303 | 1.137 | 0.820 | 0.912 | 0.748 |
| | Avg | **0.106** | **0.216** | 0.125 | 0.239 | 0.107 | 0.223 | 0.354 | 0.399 | 0.265 | 0.348 | 0.195 | 0.307 | 0.187 | 0.291 | 0.129 | 0.241 | 0.526 | 0.491 | 0.610 | 0.590 |
| PEMS07 | 12 | **0.059** | **0.158** | 0.063 | 0.166 | 0.104 | 0.224 | 0.086 | 0.204 | 0.082 | 0.191 | 0.095 | 0.207 | 0.075 | 0.182 | 0.082 | 0.181 | 0.118 | 0.235 | 0.199 | 0.336 |
| | 24 | 0.077 | 0.183 | 0.090 | 0.197 | **0.073** | **0.182** | 0.150 | 0.271 | 0.131 | 0.244 | 0.150 | 0.262 | 0.124 | 0.237 | 0.101 | 0.204 | 0.242 | 0.341 | 0.323 | 0.420 |
| | 48 | 0.107 | **0.217** | 0.135 | 0.240 | **0.100** | 0.218 | 0.338 | 0.396 | 0.224 | 0.329 | 0.253 | 0.340 | 0.218 | 0.318 | 0.134 | 0.238 | 0.562 | 0.541 | 0.390 | 0.470 |
| | 96 | 0.159 | **0.267** | 0.178 | 0.274 | **0.157** | 0.271 | 0.440 | 0.467 | 0.399 | 0.459 | 0.346 | 0.404 | 0.302 | 0.379 | 0.181 | 0.279 | 1.096 | 0.795 | 0.554 | 0.578 |
| | Avg | **0.101** | **0.206** | 0.116 | 0.219 | 0.109 | 0.224 | 0.254 | 0.335 | 0.209 | 0.306 | 0.211 | 0.303 | 0.180 | 0.279 | 0.125 | 0.226 | 0.505 | 0.478 | 0.367 | 0.451 |
| PEMS08 | 12 | 0.068 | 0.176 | 0.082 | 0.186 | **0.065** | **0.172** | 0.110 | 0.232 | 0.100 | 0.210 | 0.168 | 0.232 | 0.099 | 0.202 | 0.112 | 0.212 | 0.133 | 0.247 | 0.436 | 0.485 |
| | 24 | 0.091 | 0.202 | 0.115 | 0.222 | **0.083** | **0.197** | 0.165 | 0.290 | 0.159 | 0.267 | 0.224 | 0.281 | 0.172 | 0.254 | 0.141 | 0.238 | 0.249 | 0.343 | 0.467 | 0.502 |
| | 48 | 0.135 | 0.253 | 0.180 | 0.278 | **0.110** | **0.227** | 0.323 | 0.420 | 0.314 | 0.383 | 0.321 | 0.354 | 0.302 | 0.331 | 0.198 | 0.283 | 0.569 | 0.544 | 0.966 | 0.733 |
| | 96 | **0.209** | 0.320 | 0.246 | **0.312** | 0.292 | 0.383 | 0.680 | 0.616 | 0.625 | 0.549 | 0.408 | 0.417 | 0.550 | 0.448 | 0.320 | 0.351 | 1.166 | 0.814 | 1.385 | 0.915 |
| | Avg | **0.126** | **0.238** | 0.156 | 0.250 | 0.138 | 0.245 | 0.320 | 0.390 | 0.300 | 0.352 | 0.280 | 0.321 | 0.281 | 0.309 | 0.193 | 0.271 | 0.529 | 0.487 | 0.814 | 0.659 |
| 1*st* count | | 25 | | 1 | | 14 | | 0 | | 0 | | 0 | | 0 | | 0 | | 0 | | 0 | |

Table 18: **Few shot Forecasting Full Results** on the ETT datasets with multiple variates. A lower MSE or MAE indicates a better prediction. *Avg* is averaged from all four prediction lengths, that is {96, 192, 336, 720}. The best results are highlighted in **bold**.

| Models | | UniFy (Ours) | | CycleNet (2024) | | TimeMixer (2024) | | HDMixer (2024) | | SDformer (2024) | | PatchTST (2023) | | FreTS (2023) | | TimesNet (2023) | | DLinear (2023) | | Autoformer (2021) | |
|---|---|---|---|---|---|---|---|---|---|---|---|---|---|---|---|---|---|---|---|---|---|
| Metric | | MSE | MAE | MSE | MAE | MSE | MAE | MSE | MAE | MSE | MAE | MSE | MAE | MSE | MAE | MSE | MAE | MSE | MAE | MSE | MAE |
| ETTm1 | 96 | **0.325** | **0.362** | 0.341 | 0.370 | 0.338 | 0.372 | 0.399 | 0.405 | 0.374 | 0.391 | 0.346 | 0.368 | 0.429 | 0.433 | 0.343 | 0.380 | 0.399 | 0.410 | 0.573 | 0.512 |
| | 192 | **0.369** | **0.385** | 0.380 | 0.388 | 0.373 | 0.390 | 0.425 | 0.416 | 0.423 | 0.416 | 0.387 | 0.387 | 0.493 | 0.458 | 0.403 | 0.409 | 0.430 | 0.426 | 0.609 | 0.525 |
| | 336 | **0.397** | **0.402** | 0.410 | 0.407 | 0.402 | 0.410 | 0.451 | 0.435 | 0.464 | 0.441 | 0.418 | 0.407 | 0.592 | 0.496 | 0.434 | 0.430 | 0.459 | 0.446 | 0.590 | 0.519 |
| | 720 | **0.465** | 0.441 | 0.468 | 0.439 | 0.475 | 0.452 | 0.509 | 0.466 | 0.528 | 0.477 | 0.474 | **0.438** | 0.663 | 0.549 | 0.505 | 0.470 | 0.508 | 0.477 | 0.636 | 0.540 |
| | Avg | **0.389** | **0.398** | 0.400 | 0.401 | 0.397 | 0.406 | 0.446 | 0.431 | 0.447 | 0.431 | 0.406 | 0.400 | 0.544 | 0.484 | 0.421 | 0.422 | 0.449 | 0.440 | 0.602 | 0.524 |
| ETTm2 | 96 | 0.175 | **0.257** | **0.173** | 0.257 | 0.180 | 0.262 | 0.193 | 0.277 | 0.188 | 0.272 | 0.185 | 0.270 | 0.215 | 0.305 | 0.191 | 0.273 | 0.227 | 0.330 | 0.216 | 0.298 |
| | 192 | 0.238 | **0.297** | **0.237** | 0.298 | 0.247 | 0.307 | 0.255 | 0.314 | 0.255 | 0.314 | 0.248 | 0.309 | 0.328 | 0.367 | 0.255 | 0.311 | 0.311 | 0.388 | 0.274 | 0.333 |
| | 336 | 0.308 | 0.344 | **0.300** | **0.338** | 0.309 | 0.346 | 0.317 | 0.351 | 0.318 | 0.353 | 0.307 | 0.346 | 0.360 | 0.412 | 0.327 | 0.356 | 0.394 | 0.439 | 0.329 | 0.365 |
| | 720 | 0.402 | 0.399 | 0.404 | **0.397** | **0.400** | 0.399 | 0.412 | 0.402 | 0.415 | 0.404 | 0.406 | 0.402 | 0.506 | 0.493 | 0.447 | 0.424 | 0.585 | 0.547 | 0.419 | 0.419 |
| | Avg | 0.281 | 0.324 | **0.279** | **0.323** | 0.284 | 0.329 | 0.294 | 0.336 | 0.294 | 0.336 | 0.287 | 0.332 | 0.352 | 0.394 | 0.305 | 0.341 | 0.379 | 0.426 | 0.310 | 0.354 |
| ETTh1 | 96 | **0.389** | **0.400** | 0.400 | 0.408 | 0.395 | 0.404 | 0.557 | 0.496 | 0.519 | 0.481 | 0.404 | 0.408 | 0.591 | 0.518 | 0.483 | 0.462 | 0.541 | 0.491 | 0.523 | 0.486 |
| | 192 | **0.443** | **0.429** | 0.443 | 0.432 | 0.463 | 0.439 | 0.589 | 0.512 | 0.577 | 0.510 | 0.454 | 0.437 | 0.645 | 0.551 | 0.523 | 0.489 | 0.571 | 0.510 | 0.526 | 0.489 |
| | 336 | 0.480 | **0.446** | **0.478** | 0.452 | 0.502 | 0.457 | 0.579 | 0.519 | 0.621 | 0.534 | 0.494 | 0.458 | 0.678 | 0.575 | 0.574 | 0.518 | 0.599 | 0.533 | 0.534 | 0.506 |
| | 720 | 0.489 | **0.471** | **0.478** | 0.471 | 0.508 | 0.483 | 0.603 | 0.543 | 0.638 | 0.558 | 0.496 | 0.481 | 1.040 | 0.777 | 0.581 | 0.532 | 0.618 | 0.570 | 0.535 | 0.518 |
| | Avg | **0.450** | **0.437** | 0.450 | 0.441 | 0.467 | 0.446 | 0.582 | 0.518 | 0.589 | 0.521 | 0.462 | 0.446 | 0.739 | 0.605 | 0.540 | 0.500 | 0.582 | 0.526 | 0.530 | 0.500 |
| ETTh2 | 96 | **0.300** | **0.348** | 0.308 | 0.354 | 0.305 | 0.358 | 0.322 | 0.369 | 0.337 | 0.376 | 0.308 | 0.351 | 0.399 | 0.441 | 0.390 | 0.407 | 0.394 | 0.439 | 0.348 | 0.388 |
| | 192 | **0.384** | **0.398** | 0.389 | 0.402 | 0.398 | 0.414 | 0.397 | 0.414 | 0.423 | 0.424 | 0.395 | 0.405 | 0.581 | 0.543 | 0.447 | 0.439 | 0.467 | 0.475 | 0.427 | 0.436 |
| | 336 | 0.412 | 0.424 | 0.436 | 0.440 | 0.433 | 0.438 | **0.394** | **0.422** | 0.462 | 0.456 | 0.429 | 0.432 | 0.657 | 0.543 | 0.489 | 0.477 | 0.504 | 0.503 | 0.455 | 0.465 |
| | 720 | **0.431** | **0.445** | 0.445 | 0.453 | 0.454 | 0.458 | 0.441 | 0.453 | 0.460 | 0.464 | 0.463 | 0.464 | 0.892 | 0.673 | 0.508 | 0.492 | 0.516 | 0.521 | 0.461 | 0.475 |
| | Avg | **0.382** | **0.404** | 0.395 | 0.412 | 0.398 | 0.417 | 0.389 | 0.415 | 0.421 | 0.430 | 0.399 | 0.413 | 0.632 | 0.535 | 0.459 | 0.454 | 0.470 | 0.485 | 0.423 | 0.441 |
| 1*st* count | | 29 | | 11 | | 1 | | 2 | | 0 | | 1 | | 0 | | 0 | | 0 | | 0 | |

Table 19: **Zero-Shot Forecasting Full Results** on the ETT datasets with multiple variates. A lower MSE or MAE indicates a better prediction. *Avg* is averaged from all four prediction lengths, that is {96, 192, 336, 720}. The best results are highlighted in **bold**.

| Models | | UniFy (Ours) | | CycleNet (2024) | | TimeMixer (2024) | | HDMixer (2024) | | SDformer (2024) | | PatchTST (2023) | | FreTS (2023) | | TimesNet (2023) | | DLinear (2023) | | Autoformer (2021) | |
|---|---|---|---|---|---|---|---|---|---|---|---|---|---|---|---|---|---|---|---|---|---|
| Metric | | MSE | MAE | MSE | MAE | MSE | MAE | MSE | MAE | MSE | MAE | MSE | MAE | MSE | MAE | MSE | MAE | MSE | MAE | MSE | MAE |
| ETTm1 ⇒ ETTm2 | 96 | **0.192** | 0.281 | 0.192 | **0.264** | 0.194 | 0.273 | 0.194 | 0.267 | 0.196 | 0.273 | 0.195 | 0.271 | 0.203 | 0.304 | 0.251 | 0.320 | 0.221 | 0.314 | 0.345 | 0.425 |
| | 192 | **0.255** | 0.307 | 0.257 | 0.307 | 0.264 | 0.316 | 0.257 | **0.305** | 0.260 | 0.312 | 0.257 | 0.307 | 0.277 | 0.356 | 0.313 | 0.356 | 0.300 | 0.370 | 0.429 | 0.472 |
| | 336 | **0.314** | **0.344** | 0.317 | 0.345 | 0.321 | 0.352 | 0.318 | 0.345 | 0.319 | 0.349 | 0.314 | 0.346 | 0.361 | 0.409 | 0.429 | 0.425 | 0.378 | 0.417 | 0.645 | 0.605 |
| | 720 | 0.414 | 0.400 | 0.415 | 0.400 | 0.428 | 0.416 | **0.411** | **0.396** | 0.421 | 0.405 | 0.418 | 0.399 | 0.511 | 0.468 | 0.514 | 0.468 | 0.547 | 0.517 | 0.588 | 0.548 |
| | Avg | **0.294** | 0.333 | 0.295 | 0.329 | 0.302 | 0.339 | 0.295 | **0.328** | 0.299 | 0.335 | 0.296 | 0.331 | 0.338 | 0.392 | 0.377 | 0.392 | 0.362 | 0.405 | 0.502 | 0.513 |
| ETTm2 ⇒ ETTm1 | 96 | **0.454** | **0.427** | 0.514 | 0.451 | 0.463 | 0.433 | 0.511 | 0.449 | 0.654 | 0.510 | 0.491 | 0.437 | 0.957 | 0.544 | 0.975 | 0.632 | 0.499 | 0.448 | 0.762 | 0.591 |
| | 192 | **0.477** | **0.442** | 0.573 | 0.482 | 0.539 | 0.475 | 0.547 | 0.470 | 0.715 | 0.538 | 0.540 | 0.469 | 0.948 | 0.552 | 0.995 | 0.639 | 0.519 | 0.467 | 0.786 | 0.609 |
| | 336 | **0.524** | **0.461** | 0.554 | 0.480 | 0.616 | 0.504 | 0.564 | 0.484 | 0.701 | 0.542 | 0.578 | 0.512 | 0.927 | 0.561 | 0.998 | 0.633 | 0.548 | 0.483 | 0.866 | 0.648 |
| | 720 | 0.566 | **0.492** | 0.609 | 0.511 | 0.694 | 0.544 | 0.610 | 0.507 | 0.765 | 0.583 | 0.658 | 0.555 | 0.803 | 0.551 | 1.113 | 0.671 | **0.554** | 0.500 | 0.829 | 0.658 |
| | Avg | **0.505** | **0.456** | 0.563 | 0.481 | 0.578 | 0.489 | 0.558 | 0.478 | 0.709 | 0.543 | 0.567 | 0.493 | 0.909 | 0.552 | 1.020 | 0.644 | 0.530 | 0.475 | 0.811 | 0.619 |
| ETTh2 ⇒ ETTh1 | 96 | **0.416** | **0.419** | 0.583 | 0.502 | 0.571 | 0.506 | 0.513 | 0.479 | 0.577 | 0.512 | 0.485 | 0.465 | 0.506 | 0.471 | 0.625 | 0.545 | 0.448 | 0.443 | 0.728 | 0.592 |
| | 192 | **0.463** | **0.445** | 0.597 | 0.514 | 0.533 | 0.486 | 0.546 | 0.496 | 0.632 | 0.540 | 0.553 | 0.491 | 0.547 | 0.501 | 0.735 | 0.584 | 0.481 | 0.467 | 1.027 | 0.654 |
| | 336 | **0.499** | **0.466** | 0.638 | 0.538 | 0.601 | 0.524 | 0.565 | 0.514 | 0.652 | 0.552 | 0.589 | 0.517 | 0.580 | 0.519 | 0.725 | 0.579 | 0.523 | 0.491 | 0.843 | 0.628 |
| | 720 | **0.547** | **0.542** | 0.843 | 0.636 | 0.768 | 0.604 | 0.610 | 0.546 | 0.809 | 0.641 | 0.607 | 0.547 | 0.615 | 0.572 | 0.958 | 0.653 | 0.572 | 0.548 | 0.861 | 0.672 |
| | Avg | **0.481** | **0.468** | 0.665 | 0.548 | 0.618 | 0.530 | 0.559 | 0.509 | 0.668 | 0.561 | 0.559 | 0.505 | 0.562 | 0.516 | 0.761 | 0.590 | 0.530 | 0.487 | 0.865 | 0.637 |
| ETTm1 ⇒ ETTh1 | 96 | **0.575** | **0.501** | 0.740 | 0.563 | 0.846 | 0.597 | 0.690 | 0.552 | 0.739 | 0.567 | 0.678 | 0.543 | 1.543 | 0.781 | 0.675 | 0.541 | 0.613 | 0.517 | 0.945 | 0.619 |
| | 192 | **0.579** | **0.508** | 0.698 | 0.553 | 0.748 | 0.586 | 0.704 | 0.566 | 0.720 | 0.564 | 0.632 | 0.541 | 1.049 | 0.663 | 0.669 | 0.534 | 0.607 | 0.522 | 1.058 | 0.724 |
| | 336 | **0.604** | **0.527** | 0.705 | 0.563 | 0.793 | 0.604 | 0.808 | 0.619 | 0.805 | 0.603 | 0.643 | 0.546 | 1.015 | 0.668 | 0.684 | 0.536 | 0.628 | 0.540 | 1.216 | 0.785 |
| | 720 | **0.583** | **0.524** | 0.678 | 0.571 | 0.949 | 0.669 | 0.716 | 0.594 | 0.805 | 0.624 | 0.625 | 0.562 | 0.834 | 0.645 | 0.744 | 0.582 | 0.623 | 0.567 | 0.843 | 0.648 |
| | Avg | **0.585** | **0.515** | 0.705 | 0.563 | 0.834 | 0.614 | 0.730 | 0.583 | 0.767 | 0.590 | 0.645 | 0.548 | 1.110 | 0.689 | 0.693 | 0.548 | 0.618 | 0.537 | 1.016 | 0.694 |
| ETTh1 ⇒ ETTm1 | 96 | **0.726** | **0.549** | 0.780 | 0.555 | 0.866 | 0.583 | 0.804 | 0.565 | 0.870 | 0.588 | 0.798 | 0.556 | 0.840 | 0.601 | 1.030 | 0.649 | 0.730 | 0.555 | 0.777 | 0.587 |
| | 192 | **0.750** | 0.567 | 0.874 | 0.592 | 1.031 | 0.637 | 0.860 | 0.589 | 0.866 | 0.591 | 0.788 | **0.566** | 0.791 | 0.596 | 1.174 | 0.709 | 0.754 | 0.577 | 0.777 | 0.585 |
| | 336 | **0.749** | 0.579 | 0.856 | 0.590 | 0.796 | 0.581 | 0.821 | 0.585 | 0.851 | 0.596 | 0.764 | **0.567** | 0.768 | 0.586 | 1.281 | 0.733 | 0.759 | 0.582 | 0.830 | 0.618 |
| | 720 | 0.783 | 0.594 | 0.924 | 0.620 | 0.873 | 0.602 | 0.843 | 0.606 | 0.865 | 0.606 | **0.772** | **0.572** | 0.779 | 0.614 | 1.454 | 0.809 | 0.784 | 0.618 | 0.925 | 0.684 |
| | Avg | **0.752** | 0.572 | 0.859 | 0.589 | 0.892 | 0.601 | 0.832 | 0.586 | 0.863 | 0.595 | 0.781 | **0.565** | 0.795 | 0.599 | 1.235 | 0.725 | 0.754 | 0.583 | 0.827 | 0.619 |
| ETTh2 ⇒ ETTm1 | 96 | **0.685** | **0.522** | 0.859 | 0.580 | 1.702 | 0.744 | 0.848 | 0.571 | 1.064 | 0.638 | 0.862 | 0.581 | 0.832 | 0.571 | 1.254 | 0.688 | 0.777 | 0.558 | 0.758 | 0.586 |
| | 192 | **0.689** | **0.534** | 0.954 | 0.621 | 0.940 | 0.607 | 0.847 | 0.584 | 1.065 | 0.646 | 0.945 | 0.604 | 0.839 | 0.587 | 1.264 | 0.709 | 0.791 | 0.584 | 0.757 | 0.587 |
| | 336 | **0.741** | **0.560** | 0.971 | 0.629 | 0.761 | 0.575 | 0.846 | 0.596 | 0.983 | 0.627 | 0.933 | 0.605 | 0.788 | 0.591 | 1.196 | 0.686 | 0.759 | 0.589 | 0.769 | 0.597 |
| | 720 | 0.794 | 0.618 | 1.051 | 0.669 | 1.029 | 0.662 | 0.859 | **0.616** | 1.153 | 0.701 | 0.892 | 0.634 | 0.841 | 0.639 | 2.302 | 0.918 | 0.816 | 0.621 | 0.794 | 0.618 |
| | Avg | **0.727** | **0.559** | 0.959 | 0.625 | 1.108 | 0.647 | 0.850 | 0.592 | 1.066 | 0.653 | 0.908 | 0.606 | 0.825 | 0.597 | 1.504 | 0.750 | 0.792 | 0.588 | 0.770 | 0.597 |
| **1st count** | | **48** | | 1 | | 0 | | 5 | | 0 | | 5 | | 0 | | 0 | | 1 | | 0 | |

Table 20: **Long-Term Forecasting Full Results for New Baselines**. A lower MSE or MAE indicates a better prediction. *Avg* is averaged from all four prediction lengths, that is {96, 192, 336, 720}. The best results are highlighted in **bold**.

| Models | | UniFy (Ours) | | FITS (2025) | | TimeKAN (2025) | | FreDF (2025) | | SparseTSF (2025) | | FreqMoE (2025) | | Fredformer (2025) | |
|---|---|---|---|---|---|---|---|---|---|---|---|---|---|---|---|---|---|
| Metric | | MSE | MAE | MSE | MAE | MSE | MAE | MSE | MAE | MSE | MAE | MSE | MAE | MSE | MAE |
| ETTm1 | 96 | **0.315** | **0.355** | 0.354 | 0.375 | 0.322 | 0.361 | 0.324 | 0.367 | 0.356 | 0.375 | 0.449 | 0.474 | 0.347 | 0.377 |
| | 192 | 0.360 | **0.379** | 0.393 | 0.394 | **0.355** | 0.381 | 0.365 | 0.387 | 0.394 | 0.392 | 0.529 | 0.479 | 0.379 | 0.392 |
| | 336 | 0.390 | 0.402 | 0.425 | 0.414 | **0.384** | **0.402** | 0.391 | 0.405 | 0.425 | 0.413 | 0.668 | 0.543 | 0.407 | 0.412 |
| | 720 | 0.456 | 0.438 | 0.486 | 0.448 | **0.450** | **0.438** | 0.459 | 0.436 | 0.487 | 0.448 | 0.681 | 0.556 | 0.467 | 0.444 |
| | Avg | 0.380 | **0.394** | 0.415 | 0.408 | **0.378** | 0.396 | 0.385 | 0.399 | 0.416 | 0.407 | 0.582 | 0.513 | 0.400 | 0.406 |
| ETTm2 | 96 | **0.169** | **0.253** | 0.183 | 0.266 | 0.174 | 0.255 | 0.175 | 0.257 | 0.184 | 0.267 | 0.178 | 0.258 | 0.178 | 0.261 |
| | 192 | **0.234** | **0.295** | 0.247 | 0.305 | 0.239 | 0.299 | 0.241 | 0.299 | 0.248 | 0.305 | 0.243 | 0.304 | 0.243 | 0.302 |
| | 336 | **0.292** | **0.334** | 0.307 | 0.342 | 0.301 | 0.340 | 0.303 | 0.341 | 0.307 | 0.342 | 0.302 | 0.342 | 0.301 | 0.340 |
| | 720 | **0.393** | **0.393** | 0.407 | 0.397 | 0.395 | 0.396 | 0.395 | 0.396 | 0.407 | 0.398 | 0.489 | 0.456 | 0.396 | 0.395 |
| | Avg | **0.272** | **0.319** | 0.286 | 0.328 | 0.277 | 0.323 | 0.281 | 0.323 | 0.287 | 0.328 | 0.303 | 0.340 | 0.280 | 0.325 |
| ETTh1 | 96 | 0.372 | 0.394 | 0.385 | 0.393 | **0.367** | 0.395 | 0.402 | 0.416 | 0.385 | **0.390** | 0.539 | 0.526 | 0.380 | 0.395 |
| | 192 | 0.426 | 0.423 | 0.435 | 0.422 | **0.414** | 0.420 | 0.459 | 0.450 | 0.434 | **0.420** | 0.590 | 0.551 | 0.438 | 0.424 |
| | 336 | 0.463 | 0.435 | 0.475 | 0.444 | **0.452** | **0.435** | 0.510 | 0.482 | 0.476 | 0.439 | 0.632 | 0.572 | 0.483 | 0.444 |
| | 720 | 0.467 | 0.464 | 0.480 | 0.479 | 0.476 | 0.465 | 0.646 | 0.567 | **0.461** | **0.454** | 0.717 | 0.599 | 0.487 | 0.463 |
| | Avg | 0.432 | 0.429 | 0.444 | 0.435 | **0.427** | 0.429 | 0.504 | 0.479 | 0.439 | **0.426** | 0.620 | 0.562 | 0.447 | 0.432 |
| ETTh2 | 96 | **0.281** | **0.335** | 0.298 | 0.346 | 0.290 | 0.340 | 0.341 | 0.373 | 0.302 | 0.346 | 0.359 | 0.348 | 0.294 | 0.340 |
| | 192 | **0.366** | **0.391** | 0.385 | 0.398 | 0.375 | 0.392 | 0.462 | 0.435 | 0.384 | 0.395 | 0.599 | 0.436 | 0.376 | 0.392 |
| | 336 | **0.386** | **0.413** | 0.421 | 0.430 | 0.423 | 0.435 | 0.537 | 0.478 | 0.421 | 0.427 | 0.689 | 0.490 | 0.419 | 0.428 |
| | 720 | **0.418** | **0.438** | 0.423 | 0.440 | 0.443 | 0.449 | 0.468 | 0.467 | 0.420 | 0.438 | 0.891 | 0.577 | 0.418 | 0.438 |
| | Avg | **0.363** | **0.394** | 0.382 | 0.404 | 0.383 | 0.404 | 0.452 | 0.438 | 0.382 | 0.402 | 0.635 | 0.463 | 0.377 | 0.400 |
| ECL | 96 | **0.142** | **0.237** | 0.205 | 0.289 | 0.174 | 0.266 | 0.150 | 0.242 | 0.209 | 0.280 | 0.152 | 0.246 | 0.148 | 0.242 |
| | 192 | **0.155** | **0.247** | 0.208 | 0.297 | 0.182 | 0.273 | 0.161 | 0.253 | 0.205 | 0.281 | 0.165 | 0.255 | 0.166 | 0.259 |
| | 336 | **0.171** | **0.263** | 0.220 | 0.307 | 0.197 | 0.286 | 0.220 | 0.321 | 0.218 | 0.295 | 0.181 | 0.274 | 0.178 | 0.273 |
| | 720 | **0.215** | **0.302** | 0.260 | 0.336 | 0.236 | 0.320 | 0.259 | 0.354 | 0.260 | 0.327 | 0.219 | 0.307 | 0.225 | 0.312 |
| | Avg | **0.171** | **0.262** | 0.223 | 0.307 | 0.197 | 0.286 | 0.198 | 0.293 | 0.223 | 0.296 | 0.179 | 0.271 | 0.179 | 0.272 |
| Traffic | 96 | 0.441 | 0.302 | 0.668 | 0.428 | 0.581 | 0.379 | 0.590 | 0.334 | 0.664 | 0.390 | 0.495 | 0.322 | **0.414** | **0.286** |
| | 192 | 0.452 | 0.301 | 0.645 | 0.439 | 0.556 | 0.368 | 0.600 | 0.341 | 0.611 | 0.367 | 0.498 | 0.322 | **0.435** | **0.296** |
| | 336 | 0.468 | 0.307 | 0.642 | 0.425 | 0.563 | 0.364 | 0.614 | 0.349 | 0.617 | 0.369 | 0.554 | 0.356 | **0.447** | **0.300** |
| | 720 | 0.508 | 0.332 | 0.672 | 0.428 | 0.609 | 0.380 | 0.648 | 0.353 | 0.655 | 0.388 | 2.733 | 1.135 | **0.478** | **0.317** |
| | Avg | 0.467 | 0.311 | 0.657 | 0.430 | 0.577 | 0.373 | 0.613 | 0.344 | 0.637 | 0.379 | 1.070 | 0.534 | **0.444** | **0.300** |
| Weather | 96 | **0.156** | **0.203** | 0.165 | 0.212 | 0.162 | 0.208 | 0.157 | 0.208 | 0.197 | 0.236 | 0.168 | 0.215 | 0.162 | 0.207 |
| | 192 | 0.206 | 0.248 | 0.215 | 0.257 | 0.207 | 0.249 | **0.205** | **0.246** | 0.243 | 0.273 | 0.212 | 0.253 | 0.208 | 0.250 |
| | 336 | 0.262 | 0.290 | 0.269 | 0.295 | 0.263 | 0.290 | **0.259** | **0.287** | 0.292 | 0.308 | 0.268 | 0.291 | 0.265 | 0.291 |
| | 720 | 0.343 | 0.343 | 0.351 | 0.347 | **0.338** | 0.340 | 0.341 | **0.339** | 0.368 | 0.357 | 0.342 | 0.345 | 0.341 | 0.341 |
| | Avg | 0.242 | 0.271 | 0.250 | 0.278 | 0.243 | 0.272 | **0.241** | **0.270** | 0.275 | 0.294 | 0.248 | 0.276 | 0.244 | 0.272 |
| Exchange | 96 | **0.081** | **0.198** | 0.095 | 0.219 | 0.087 | 0.206 | 0.082 | 0.199 | 0.089 | 0.210 | 0.088 | 0.205 | 0.093 | 0.213 |
| | 192 | **0.172** | **0.293** | 0.192 | 0.314 | 0.182 | 0.303 | 0.172 | 0.294 | 0.176 | 0.300 | 0.234 | 0.313 | 0.177 | 0.298 |
| | 336 | 0.329 | 0.414 | 0.342 | 0.426 | 0.344 | 0.423 | **0.316** | **0.405** | 0.319 | 0.409 | 0.433 | 0.445 | 0.343 | 0.421 |
| | 720 | **0.826** | **0.687** | 0.882 | 0.714 | 0.898 | 0.710 | 0.838 | 0.694 | 0.829 | 0.687 | 0.921 | 0.699 | 0.887 | 0.710 |
| | Avg | **0.352** | **0.398** | 0.378 | 0.418 | 0.378 | 0.411 | 0.352 | 0.398 | 0.353 | 0.402 | 0.419 | 0.416 | 0.375 | 0.411 |
| **1st count** | | **48** | | 0 | | 12 | | 9 | | 5 | | 0 | | 10 | |

Table 21: **Short-Term Forecasting Full Results for New Baselines** on the PEMS datasets with multiple variates. A lower MSE or MAE indicates a better prediction. *Avg* is averaged from all four prediction lengths, that is {96, 192, 336, 720}. The best results are highlighted in **bold**.

| Models | | UniFy (Ours) | | FITS (2025) | | TimeKAN (2025) | | FreDF (2025) | | SparseTSF (2025) | | FreqMoE (2025) | | Fredformer (2025) | |
|---|---|---|---|---|---|---|---|---|---|---|---|---|---|---|---|
| Metric | | MSE | MAE | MSE | MAE | MSE | MAE | MSE | MAE | MSE | MAE | MSE | MAE | MSE | MAE |
| PEMS03 | 12 | **0.068** | **0.174** | 0.135 | 0.250 | 963.533 | 19.372 | - | - | 0.185 | 0.297 | 0.368 | 0.465 | 0.088 | 0.202 |
| | 24 | **0.093** | **0.203** | 0.281 | 0.368 | 1721.465 | 26.366 | - | - | 0.324 | 0.398 | 0.812 | 0.803 | 0.131 | 0.245 |
| | 48 | **0.143** | **0.256** | 0.639 | 0.586 | 3471.510 | 37.916 | - | - | 0.650 | 0.588 | 1.072 | 0.818 | 0.245 | 0.348 |
| | 96 | **0.218** | **0.314** | 1.277 | 0.896 | 6134.413 | 55.566 | - | - | 1.186 | 0.849 | 0.914 | 0.749 | 0.375 | 0.444 |
| | Avg | **0.131** | **0.237** | 0.583 | 0.525 | 3072.730 | 34.805 | - | - | 0.586 | 0.533 | 0.792 | 0.709 | 0.210 | 0.310 |
| PEMS04 | 12 | **0.070** | **0.173** | 0.168 | 0.291 | 1595.307 | 25.911 | - | - | 0.199 | 0.314 | 0.373 | 0.469 | 0.107 | 0.221 |
| | 24 | **0.086** | **0.194** | 0.336 | 0.424 | 2708.001 | 34.138 | - | - | 0.337 | 0.413 | 0.169 | 0.280 | 0.169 | 0.280 |
| | 48 | **0.110** | **0.223** | 0.739 | 0.653 | 5226.082 | 48.469 | - | - | 0.677 | 0.605 | 1.079 | 0.826 | 0.305 | 0.386 |
| | 96 | **0.158** | **0.273** | 1.455 | 0.974 | 8921.913 | 66.248 | - | - | 1.263 | 0.881 | 0.464 | 0.503 | 0.472 | 0.503 |
| | Avg | **0.106** | **0.216** | 0.675 | 0.586 | 4612.826 | 43.692 | - | - | 0.619 | 0.553 | 0.521 | 0.520 | 0.263 | 0.348 |
| PEMS07 | 12 | **0.059** | **0.158** | 0.127 | 0.246 | 1898.019 | 28.653 | - | - | 0.179 | 0.300 | 0.088 | 0.197 | 0.071 | 0.178 |
| | 24 | **0.077** | **0.183** | 0.276 | 0.370 | 3344.391 | 38.814 | - | - | 0.328 | 0.413 | 0.832 | 0.709 | 0.107 | 0.216 |
| | 48 | **0.107** | **0.217** | 0.648 | 0.592 | 6551.054 | 55.725 | - | - | 0.676 | 0.606 | 1.127 | 0.833 | 0.180 | 0.287 |
| | 96 | **0.159** | **0.267** | 1.289 | 0.895 | 12043.198 | 78.063 | - | - | 1.252 | 0.882 | 1.501 | 0.989 | 0.283 | 0.374 |
| | Avg | **0.101** | **0.206** | 0.585 | 0.526 | 5959.166 | 50.314 | - | - | 0.609 | 0.549 | 0.887 | 0.682 | 0.160 | 0.264 |
| PEMS08 | 12 | **0.068** | **0.176** | 0.161 | 0.283 | 1012.387 | 20.347 | - | - | 0.191 | 0.306 | 0.419 | 0.496 | 0.101 | 0.214 |
| | 24 | **0.091** | **0.202** | 0.327 | 0.416 | 1728.678 | 26.978 | - | - | 0.325 | 0.404 | 0.163 | 0.275 | 0.167 | 0.275 |
| | 48 | **0.135** | **0.253** | 0.748 | 0.653 | 3453.578 | 39.194 | - | - | 0.681 | 0.607 | 1.116 | 0.844 | 0.308 | 0.382 |
| | 96 | **0.209** | **0.320** | 1.529 | 0.982 | 6994.506 | 57.240 | - | - | 1.346 | 0.897 | 0.849 | 0.719 | 0.538 | 0.513 |
| | Avg | **0.126** | **0.238** | 0.691 | 0.584 | 3297.287 | 35.940 | - | - | 0.636 | 0.554 | 0.637 | 0.584 | 0.279 | 0.346 |
| 1st count | | **40** | | 0 | | 0 | | 0 | | 0 | | 0 | | 0 | |

Table 22: **Few shot Forecasting Full Results for New Baselines** on the ETT datasets with multiple variates. A lower MSE or MAE indicates a better prediction. *Avg* is averaged from all four prediction lengths, that is {96, 192, 336, 720}. The best results are highlighted in **bold**.

| Models | | UniFy (Ours) | | FITS (2025) | | TimeKAN (2025) | | FreDF (2025) | | SparseTSF (2025) | | FreqMoE (2025) | | Fredformer (2025) | |
|---|---|---|---|---|---|---|---|---|---|---|---|---|---|---|---|
| Metric | | MSE | MAE | MSE | MAE | MSE | MAE | MSE | MAE | MSE | MAE | MSE | MAE | MSE | MAE |
| ETTm1 | 96 | **0.325** | **0.362** | 0.543 | 0.482 | 0.341 | 0.373 | 0.357 | 0.392 | 0.397 | 0.401 | 0.331 | 0.365 | 0.568 | 0.495 |
| | 192 | 0.369 | **0.385** | 0.563 | 0.493 | **0.368** | 0.387 | 0.422 | 0.430 | 0.425 | 0.413 | 0.378 | 0.388 | 0.623 | 0.522 |
| | 336 | 0.397 | **0.402** | 0.638 | 0.527 | **0.396** | 0.404 | 0.433 | 0.437 | 0.448 | 0.429 | 0.686 | 0.550 | 0.645 | 0.535 |
| | 720 | 0.465 | 0.441 | 0.766 | 0.582 | **0.449** | **0.436** | 0.545 | 0.495 | 0.630 | 0.526 | 2.934 | 1.081 | 0.571 | 0.501 |
| | Avg | **0.389** | **0.398** | 0.628 | 0.521 | **0.389** | 0.400 | 0.439 | 0.439 | 0.475 | 0.442 | 1.082 | 0.596 | 0.602 | 0.513 |
| ETTm2 | 96 | **0.175** | **0.257** | 0.202 | 0.286 | 0.179 | 0.263 | 0.189 | 0.272 | 0.201 | 0.283 | 0.181 | 0.265 | 0.198 | 0.282 |
| | 192 | **0.238** | **0.297** | 0.266 | 0.324 | 0.245 | 0.305 | 0.258 | 0.316 | 0.260 | 0.316 | 0.250 | 0.310 | 0.265 | 0.323 |
| | 336 | **0.308** | **0.344** | 0.331 | 0.365 | 0.311 | 0.347 | 0.327 | 0.358 | 0.320 | 0.354 | 0.314 | 0.352 | 0.333 | 0.365 |
| | 720 | **0.402** | **0.399** | 0.453 | 0.434 | 0.413 | 0.404 | 0.440 | 0.418 | 0.432 | 0.425 | 1.392 | 0.744 | 0.428 | 0.415 |
| | Avg | **0.281** | **0.324** | 0.313 | 0.352 | 0.287 | 0.330 | 0.304 | 0.341 | 0.303 | 0.345 | 0.534 | 0.418 | 0.306 | 0.346 |
| ETTh1 | 96 | **0.389** | **0.400** | 0.847 | 0.614 | 0.403 | 0.410 | 0.553 | 0.502 | 0.439 | 0.435 | 0.394 | 0.407 | 0.409 | 0.419 |
| | 192 | **0.443** | **0.429** | 1.012 | 0.677 | 0.457 | 0.439 | 0.622 | 0.530 | 0.485 | 0.462 | 0.462 | 0.444 | 0.464 | 0.449 |
| | 336 | **0.480** | **0.446** | 1.179 | 0.744 | 0.486 | 0.450 | 0.595 | 0.528 | 0.507 | 0.468 | 0.580 | 0.516 | 0.502 | 0.468 |
| | 720 | **0.489** | **0.471** | 1.637 | 0.883 | 0.489 | 0.473 | 0.695 | 0.595 | 0.493 | 0.482 | 15.709 | 2.634 | 0.519 | 0.498 |
| | Avg | **0.450** | **0.437** | 1.169 | 0.730 | 0.459 | 0.443 | 0.616 | 0.539 | 0.481 | 0.462 | 4.286 | 1.000 | 0.474 | 0.459 |
| ETTh2 | 96 | **0.300** | **0.348** | 0.389 | 0.412 | 0.303 | 0.350 | 0.354 | 0.389 | 0.326 | 0.366 | 0.309 | 0.356 | 0.306 | 0.352 |
| | 192 | 0.384 | 0.398 | 0.502 | 0.476 | 0.385 | 0.400 | 0.432 | 0.435 | 0.410 | 0.416 | 0.389 | 0.403 | 0.387 | 0.402 |
| | 336 | 0.412 | 0.424 | 0.575 | 0.521 | 0.428 | 0.435 | 0.460 | 0.460 | 0.441 | 0.444 | 0.413 | 0.430 | **0.392** | **0.415** |
| | 720 | 0.431 | 0.445 | 0.699 | 0.584 | 0.441 | 0.450 | 0.474 | 0.471 | 0.435 | 0.449 | 4.937 | 1.377 | **0.426** | **0.443** |
| | Avg | 0.382 | 0.404 | 0.541 | 0.498 | 0.389 | 0.409 | 0.430 | 0.439 | 0.403 | 0.419 | 1.512 | 0.642 | **0.378** | **0.403** |
| 1st count | | **30** | | 0 | | 5 | | 0 | | 0 | | 0 | | 6 | |

Table 23: **Few shot Forecasting Full Results for New Baselines** on the ETT datasets with multiple variates. A lower MSE or MAE indicates a better prediction. *Avg* is averaged from all four prediction lengths, that is {96, 192, 336, 720}. The best results are highlighted in **bold**.

| Models | | UniFy (Ours) | | FITS (2025) | | TimeKAN (2025) | | FreDF (2025) | | SparseTSF (2025) | | FreqMoE (2025) | | Fredformer (2025) | |
|---|---|---|---|---|---|---|---|---|---|---|---|---|---|---|---|
| Metric | | MSE | MAE | MSE | MAE | MSE | MAE | MSE | MAE | MSE | MAE | MSE | MAE | MSE | MAE |
| ETTm1 ⇒ ETTm2 | 96 | **0.192** | 0.281 | 0.192 | **0.264** | 0.203 | 0.280 | 0.212 | 0.289 | 0.195 | 0.266 | 0.195 | 0.269 | 0.192 | 0.266 |
| | 192 | 0.255 | 0.307 | 0.255 | **0.304** | 0.263 | 0.316 | 0.282 | 0.332 | 0.265 | 0.309 | 0.258 | 0.309 | **0.254** | 0.305 |
| | 336 | 0.314 | 0.344 | 0.313 | **0.340** | 0.319 | 0.350 | 0.345 | 0.370 | 0.316 | 0.342 | 0.317 | 0.345 | **0.311** | 0.342 |
| | 720 | 0.414 | 0.400 | 0.411 | **0.396** | 0.420 | 0.405 | 0.444 | 0.422 | 0.413 | 0.396 | 1.076 | 0.662 | **0.408** | 0.398 |
| | Avg | 0.294 | 0.333 | 0.293 | **0.326** | 0.301 | 0.338 | 0.321 | 0.353 | 0.297 | 0.328 | 0.462 | 0.396 | **0.291** | 0.328 |
| ETTm2 ⇒ ETTm1 | 96 | **0.454** | **0.427** | 0.479 | 0.440 | 0.670 | 0.514 | 0.712 | 0.538 | 0.485 | 0.443 | 0.499 | 0.449 | 0.547 | 0.459 |
| | 192 | **0.477** | **0.442** | 0.502 | 0.455 | 0.702 | 0.536 | 0.672 | 0.531 | 0.515 | 0.460 | 0.552 | 0.480 | 0.517 | 0.462 |
| | 336 | 0.524 | **0.461** | **0.522** | 0.469 | 0.830 | 0.588 | 0.724 | 0.552 | 0.541 | 0.477 | 0.608 | 0.510 | 0.608 | 0.497 |
| | 720 | 0.566 | **0.492** | **0.559** | 0.492 | 0.811 | 0.590 | 1.014 | 0.647 | 0.578 | 0.499 | 4.881 | 1.315 | 0.643 | 0.530 |
| | Avg | **0.505** | **0.456** | 0.516 | 0.464 | 0.753 | 0.557 | 0.781 | 0.567 | 0.530 | 0.470 | 1.635 | 0.689 | 0.579 | 0.487 |
| ETTh2 ⇒ ETTh1 | 96 | **0.416** | **0.419** | 0.444 | 0.438 | 0.586 | 0.521 | 0.699 | 0.572 | 0.454 | 0.442 | 0.534 | 0.492 | 0.495 | 0.466 |
| | 192 | **0.463** | **0.445** | 0.496 | 0.468 | 0.640 | 0.550 | 0.710 | 0.581 | 0.484 | 0.457 | 0.591 | 0.521 | 0.609 | 0.529 |
| | 336 | **0.499** | **0.466** | 0.521 | 0.484 | 0.703 | 0.583 | 0.790 | 0.614 | 0.517 | 0.473 | 0.681 | 0.563 | 0.738 | 0.591 |
| | 720 | 0.547 | 0.542 | 0.498 | 0.491 | 0.699 | 0.598 | 0.844 | 0.625 | **0.493** | **0.478** | 12.647 | 2.324 | 0.674 | 0.582 |
| | Avg | **0.481** | 0.468 | 0.490 | 0.470 | 0.657 | 0.563 | 0.761 | 0.598 | 0.487 | **0.463** | 3.613 | 0.975 | 0.629 | 0.542 |
| ETTm1 ⇒ ETTh1 | 96 | **0.575** | **0.501** | 0.586 | 0.503 | 0.707 | 0.555 | 0.794 | 0.594 | 0.617 | 0.521 | 0.760 | 0.598 | 0.720 | 0.564 |
| | 192 | **0.579** | 0.508 | 0.583 | **0.506** | 0.733 | 0.568 | 0.793 | 0.604 | 0.607 | 0.518 | 0.713 | 0.582 | 0.658 | 0.545 |
| | 336 | **0.604** | 0.527 | 0.605 | **0.520** | 0.733 | 0.577 | 0.792 | 0.622 | 0.623 | 0.531 | 0.723 | 0.586 | 0.657 | 0.543 |
| | 720 | 0.583 | **0.524** | **0.577** | 0.528 | 0.745 | 0.602 | 0.797 | 0.630 | 0.598 | 0.541 | 4.826 | 1.374 | 0.645 | 0.562 |
| | Avg | **0.585** | 0.515 | 0.588 | **0.514** | 0.730 | 0.576 | 0.794 | 0.613 | 0.611 | 0.528 | 1.756 | 0.785 | 0.670 | 0.554 |
| ETTh1 ⇒ ETTm1 | 96 | 0.726 | **0.549** | **0.694** | 0.643 | 0.800 | 0.568 | 1.031 | 0.657 | 0.753 | 0.559 | 0.772 | 0.562 | 0.821 | 0.567 |
| | 192 | 0.750 | 0.567 | 0.783 | 0.677 | 0.808 | 0.578 | 0.974 | 0.655 | **0.746** | 0.563 | 0.774 | 0.572 | 0.770 | **0.558** |
| | 336 | 0.749 | 0.579 | 0.900 | 0.713 | 0.813 | 0.587 | 0.900 | 0.634 | **0.737** | 0.567 | 0.782 | 0.585 | 0.788 | **0.564** |
| | 720 | 0.783 | 0.594 | 1.247 | 0.794 | 0.854 | 0.604 | 0.861 | 0.625 | **0.771** | 0.586 | 13.907 | 2.334 | 0.772 | **0.570** |
| | Avg | 0.752 | 0.572 | 0.906 | 0.707 | 0.819 | 0.584 | 0.942 | 0.643 | **0.752** | 0.569 | 4.059 | 1.013 | 0.788 | **0.565** |
| ETTh2 ⇒ ETTm1 | 96 | **0.685** | **0.522** | 0.709 | 0.543 | 1.099 | 0.650 | 1.011 | 0.632 | 0.832 | 0.580 | 0.876 | 0.588 | 1.144 | 0.641 |
| | 192 | **0.689** | **0.534** | 0.746 | 0.564 | 0.963 | 0.621 | 1.133 | 0.658 | 0.820 | 0.583 | 0.910 | 0.601 | 1.209 | 0.679 |
| | 336 | **0.741** | **0.560** | 0.742 | 0.568 | 1.034 | 0.645 | 1.063 | 0.656 | 0.826 | 0.591 | 0.812 | 0.587 | 0.961 | 0.627 |
| | 720 | 0.794 | 0.618 | **0.757** | **0.581** | 0.912 | 0.628 | 1.026 | 0.662 | 0.816 | 0.598 | 11.844 | 2.069 | 0.967 | 0.630 |
| | Avg | **0.727** | **0.559** | 0.739 | 0.564 | 1.002 | 0.636 | 1.058 | 0.652 | 0.824 | 0.588 | 3.611 | 0.961 | 1.070 | 0.644 |
| 1st count | | 31 | | 14 | | 0 | | 0 | | 7 | | 0 | | 8 | |

# P  TOY DATASETS FOR CONTROLLED FREQUENCY COMPETITION AND NON-STATIONARITY

To systematically evaluate UniFy under controlled conditions, we construct a synthetic benchmark consisting of nine scenarios formed by crossing:

- **Three levels of frequency competition**: Weak / Medium / Strong;
- **Three levels of non-stationarity**: None / Medium / Strong.

This yields $3 \times 3$ controlled environments, allowing us to independently vary spectral interference and temporal non-stationarity.

**Base construction.**  Each synthetic time series is generated as the superposition of two periodic components:

$$x(t) \;=\; a_1(t) \cos\big(\omega_1(t)\, t + \phi_1(t)\big) \;+\; a_2(t) \cos\big(\omega_2(t)\, t + \phi_2(t)\big), \qquad t = 1, \ldots, T. \tag{27}$$

The design of $\{a_i(t),\, \omega_i(t),\, \phi_i(t)\}_{i=1}^{2}$ controls the competition strength and non-stationarity.

**Frequency competition.**  Frequency competition is controlled by the initial amplitude ratio follow by Section.A:

$$\alpha \;=\; \frac{a_2(0)}{a_1(0)}, \qquad a_1(0) = 1. \tag{28}$$

We use three levels:

- **Weak**: $\alpha = 0.2$    (secondary frequency negligible),
- **Medium**: $\alpha = 0.7$    (comparable strength),
- **Strong**: $\alpha = 1.0$    (maximum competition).

Thus the initial amplitudes are

$$a_1(0) = 1, \qquad a_2(0) = \alpha. \tag{29}$$

We inject non-stationarity into three aspects: amplitude drift, frequency drift, and phase drift.

**(1) Amplitude drift.**  For each component $i \in \{1, 2\}$:

- **None**:
$$a_i(t) \;=\; a_i(0), \qquad \forall t. \tag{30}$$
- **Medium** (slow sinusoidal modulation):
$$a_i(t) \;=\; a_i(0)\Big(1 + 0.2 \sin\big(2\pi t/T\big)\Big). \tag{31}$$
- **Strong** (nonlinear envelope):
$$a_i(t) \;=\; a_i(0)\Big(1 + 0.8 \sin\big(2\pi t/T\big) + 0.4 \sin\big(4\pi t/T\big)\Big). \tag{32}$$

**(2) Frequency drift.**  Let $\omega_1$ and $\omega_2$ be base angular frequencies. We define $\omega_i(t)$ as:

- **None**:
$$\omega_i(t) \;=\; \omega_i, \qquad \forall t. \tag{33}$$
- **Medium** (single regime switch):
$$\omega_i(t) \;=\; \begin{cases} \omega_i, & t < T/2, \\ 1.2\,\omega_i, & t \geq T/2. \end{cases} \tag{34}$$
- **Strong** (multi-segment switch):
$$\omega_i(t) \;=\; \begin{cases} \omega_i, & t < T/3, \\ 1.3\,\omega_i, & T/3 \leq t < 2T/3, \\ 0.8\,\omega_i, & t \geq 2T/3. \end{cases} \tag{35}$$

**(3) Phase drift.** Similarly, phase non-stationarity is modeled as:

- **None**:
$$\phi_i(t) = \phi_i(0), \qquad \forall t. \tag{36}$$

- **Medium** (linear drift):
$$\phi_i(t) = \phi_i(0) + 0.002\,t. \tag{37}$$

- **Strong** (nonlinear drift):
$$\phi_i(t) = \phi_i(0) + 0.01\,t + 0.002\sin(0.1t). \tag{38}$$

## P.1 EXPERIMENT RESULTS ON TOY DATASETS

Using the proposed toy datasets, we can clearly observe the effect of frequency competition and non-stationarity on different models. For UniFy and its ablation variants, all components exhibit noticeable performance drops in the more challenging scenarios (i.e., the four cells in the lower-right region of the $3\times3$ grid). The degradation is particularly severe when removing AFS, which is responsible for constructing frequency subspaces. Together with the ablation results in Table 5, these findings further validate the necessity of AFS and confirm that explicit frequency decomposition is indeed beneficial for forecasting under spectrally complex conditions.

For the other SOTA baselines, we adopt the same configuration settings for fairness. As discussed in the main paper, representative time-domain models CycleNet, it fail to adapt to high-interference or rapidly varying spectral structures.

TimeKAN, FreqMoE and FITS, representing frequency-domain approaches, also show notable weaknesses. TimeKAN constructs multi-scale representations via average pooling, making its performance highly sensitive to the predefined pooling configurations. FreqMoE models each frequency point independently, so in the non-stationary scenarios such as frequency drift, there is a decrease in generalization. FITS exhibits severe performance degradation on our toy datasets, reflecting its difficulty in maintaining stable and robust frequency selection under non-stationary conditions.

Overall, these controlled experiments demonstrate that UniFy not only maintains strong robustness across all nine toy scenarios but also preserves significant performance advantages in the most challenging cases—highlighting its ability to effectively mitigate frequency competition and handle non-stationary periodicity.

## Q SUPPLY RELATED WORK

We next discuss several recent frequency-domain forecasting models and clarify how UniFy differs from them in design motivation and modeling principles. TimeKAN constructs multi-scale sequences via average pooling and then performs frequency analysis on each pooled subsequence. However, pooling inevitably removes fine-grained spectral details, preserving only dominant frequencies. While effective for building efficient KAN blocks, this design is not intended to explicitly resolve frequency competition; instead, it focuses on representing frequency patterns at different scales. Moreover, its reliance on pooling configurations makes it sensitive to hyperparameter choices. FreTS leverages per-channel Fourier features primarily to model cross-channel dependencies. Although it introduces frequency awareness, its frequency modeling still occurs within a shared representation space, meaning that dominant and secondary frequencies can still interfere with each other. FITS shares a conceptual similarity with CycleNet. By applying a low-pass filter to historical sequences, it extracts the stable dominant frequency components—essentially a smoothed periodic template. This design implicitly avoids frequency competition by discarding higher-frequency components. However, FITS still suffers from two limitations: it struggles to model mid-frequency structures that exhibit fluctuations or regime changes; its behavior strongly depends on the manually defined cutoff of the low-pass filter, analogous to CycleNet's reliance on predefined cycle lengths.

Fredformer identifies a key phenomenon that is closely related to our notion of frequency competition, namely that frequency bias arises when the model disproportionately focuses on high-energy frequency components. To counter this, Fredformer partitions the spectrum into bands and normalizes the top-$k$ frequencies within each band, encouraging the model to attend to weaker components

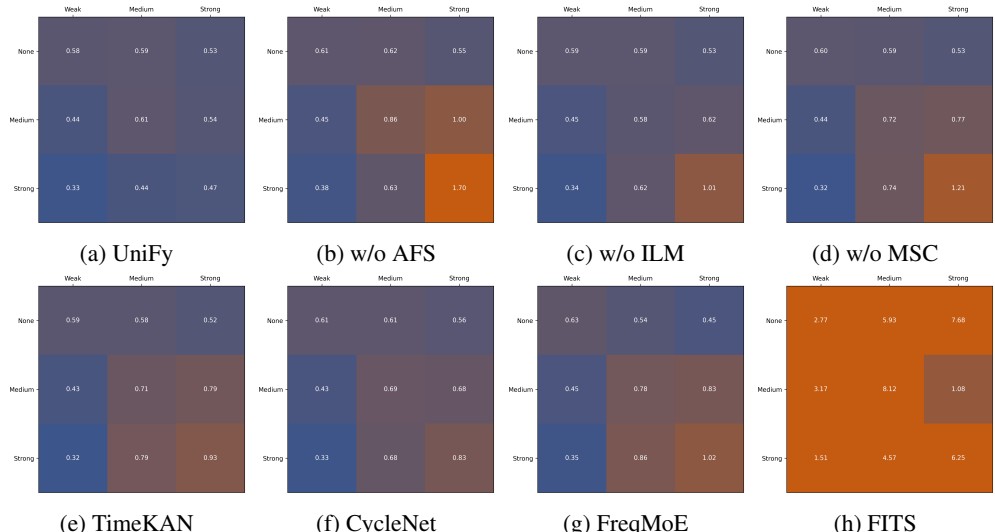

Figure 17: Results on the toy datasets. The first row reports UniFy and its Ablation settings follow with Table. 5. The second row contains 4 SOTA methods. All experiments are conducted with an input length of 96, batch size of 16, learning rate of 0.05 and hidden dimension $d_{model} = 128$. The color gradient from blue (low MSE) to orange (high MSE) visualizes the performance across methods.

instead of over-focusing on peaks. While this band-wise reweighting effectively alleviates energy bias in relatively stable settings, its normalization scheme becomes harder to transfer to scenarios with strong non-stationarity.

FreDF transforms the sequence into the Fourier domain and models each frequency bin independently. FreqMoE further extends this idea by assigning each frequency bin to a mixture-of-experts (MoE) branch. While these approaches provide extremely fine-grained decomposition—one subspace per frequency—they also assume frequency bins to be stable and semantically independent. This is restrictive for real-world non-stationary data, where mid- and high-frequency components often exhibit substantial drift. Modeling each drifting frequency with an isolated expert introduces a brittle inductive bias, leading to poor generalization when frequencies shift over time.

In contrast, UniFy allocates groups of frequencies to adaptive subspaces, allowing each subspace to flexibly track drifting spectral structures while avoiding interference between dominant and secondary frequency components. This distinction enables UniFy to model non-stationary periodicity more robustly than prior frequency-domain methods.

# R ABLATION STUDY ON MSC

To systematically distinguish between the effects of calibration functionality versus mere parameter increase, we conducted controlled ablation studies by varying both the depth and width of the calibration layers. We compared the following six configurations:

1. Our full MSC module (UniFy)
2. Completely removing MSC (Case 1)
3. Using a single linear layer (Case 2)
4. Using 2 linear layers (no activation) (Case 3)
5. Using 3 linear layers (no activation) (Case 4)
6. Using 3 linear layers with activation (Case 5)

The key performance comparison across different hidden dimensions is summarized in Table.24:

Table 24: Ablation study on MSC on ETTh1 datasets. All experiments are conducted with an input length of 96, output length of 96, batch size of 8, Decomposition Round Num of 2 (from Equ. 2) and hidden dimension $d_{\text{model}} = 128$.

| Hidden Size | UniFy | Case.1 | Case.2 | Case.3 | Case.4 | Case.5 |
|---|---|---|---|---|---|---|
| 64 | 0.379 | 0.412 | 0.384 | 0.385 | 0.702 | 0.701 |
| 128 | 0.382 | 0.412 | 0.384 | 0.388 | 0.400 | 0.702 |
| 256 | 0.383 | 0.412 | 0.384 | 0.387 | 0.400 | 0.702 |
| 512 | 0.395 | 0.412 | 0.384 | 0.394 | 0.400 | 0.702 |
| 1024 | 0.398 | 0.412 | 0.384 | 0.385 | 0.469 | 0.704 |

Our analysis reveals two key findings:

1. **The Necessity of Calibration**: Complete removal of MSC (Case 1) consistently degrades performance, confirming its essential role in our framework.

2. Simply increasing the depth or width of the calibration network (Cases 3-5) does not yield monotonic performance improvements. Conversely, excessive parameters (Cases 4 & 5) lead to optimization difficulties and significant performance deterioration. This strongly indicates that MSC's benefits are not derived from the additional parameters or nonlinear capacity alone.

Furthermore, we provide visualization analyses in Figure.18 that illustrate the transformation of feature distributions before and after MSC processing.

In conclusion, both systematic ablation studies and visualization evidence confirm that the core value of MSC lies in its **carefully designed calibration mechanism itself**, not in the incidental parameter count it introduces.

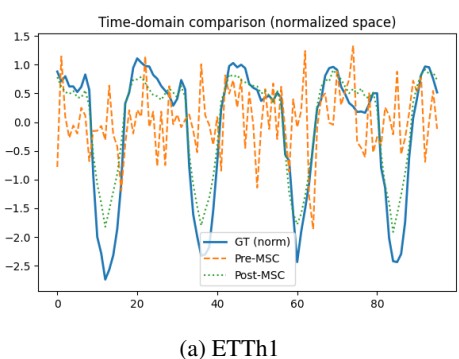 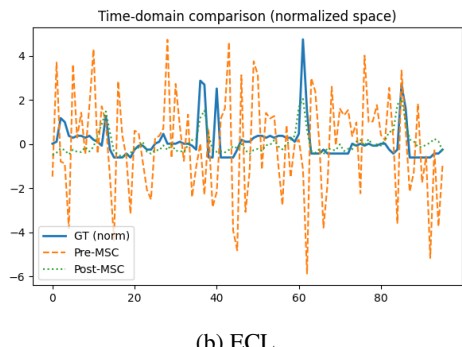

(a) ETTh1                (b) ECL

Figure 18: Visualization for MSC. All experiments are conducted with an input length of 96, output length of 96, batch size of 8, Decomposition Round Num of 2 (from Equ. 2) and hidden dimension $d_{\text{model}} = 128$.

## S   ANALYSIS FOR CHANNEL DEPEDENCY IN UNIFY

Although the Adaptive Frequency Sampling (AFS) module processes data in a channel-wise manner, its frequency decomposition employs **shared parameters across all channels**, making it not strictly channel-independent. To elucidate this behavior more precisely, we conceptualize AFS as a frequency-domain representation module. For each channel $c$, we construct a $K$-dimensional vector characterizing its energy distribution across the learned frequency components.

For each frequency component $k = 1, \ldots, K$, we compute the average energy for channel $c$ across batch and time dimensions:

$$E_c^{(k)} = \mathbb{E}_{b,t}\left[\left(\text{scale}_{b,c}^{(k)}(t)\right)^2\right].$$

The vector $\mathbf{h}_c = [E_c^{(1)}, \dots, E_c^{(K)}]$ serves as the **frequency-space representation** of channel $c$.

We conduct the following analytical procedure on a representative dataset:

1. Compute the Pearson correlation matrix between all channel pairs in the raw data, and identify groups of **strongly correlated** channels using a threshold of $|\rho| \geq 0.7$.

2. Process the data through UniFy, extract $\mathbf{h}_c$ from the AFS module for each channel, and apply **t-SNE** to obtain a 2D embedding.

3. In the resulting t-SNE visualization, channels belonging to the same high-correlation group (based on Pearson correlation) are assigned identical colors.

The resulting visualization are shown in Fig.19 and Fig.20, we randomly choose two cases from the batch, which demonstrates that channels exhibiting strong correlation in the raw data domain tend to form clusters in the t-SNE embedding space, while weakly correlated or isolated channels appear more dispersed. This pattern indicates that the **shared frequency decomposition in AFS induces similar frequency-space representations for statistically correlated variables**, despite the channel-wise architecture of the forecasting head.

Therefore, we conclude that UniFy does not disregard inter-variable dependencies: through its shared frequency-domain parameterization, the model implicitly captures channel relationships rather than treating each variable as completely independent.

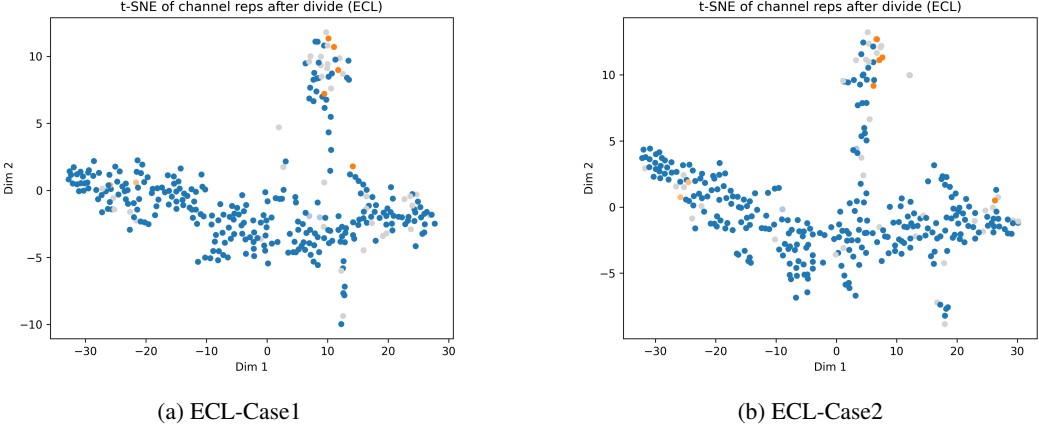

(a) ECL-Case1                                    (b) ECL-Case2

Figure 19: Visualization for MSC on ECL dataset ($Num_{Channel} = 321$). All experiments are conducted with an input length of 96, output length of 96, batch size of 8, Decomposition Round Num of 2 (from Equ. 2) and hidden dimension $d_{\text{model}} = 128$.

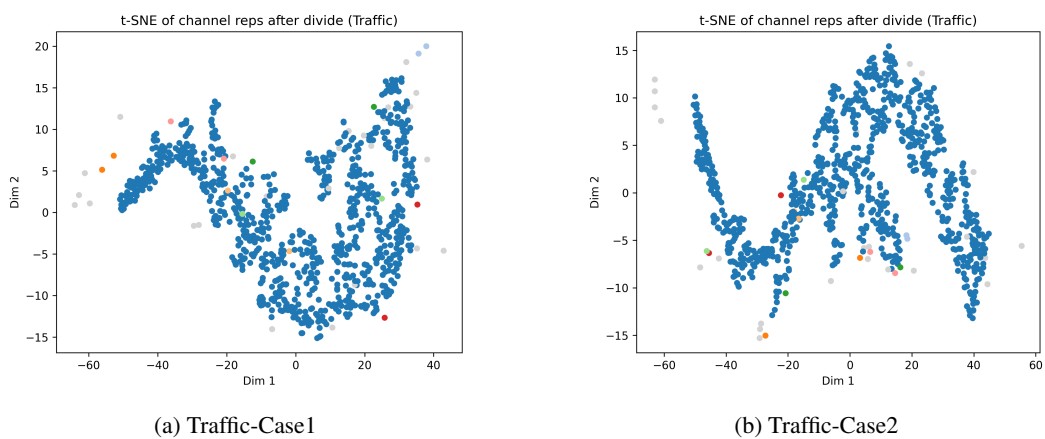

(a) Traffic-Case1        (b) Traffic-Case2

Figure 20: Visualization for MSC on Traffic dataset ($Num_{Channel} = 862$). All experiments are conducted with an input length of 96, output length of 96, batch size of 8, Decomposition Round Num of 2 (from Equ. 2) and hidden dimension $d_{model} = 128$.

