# OpenReview forum: "UniFy: Efficient Modeling of Non-Stationary Periodicity for Time Series Forecasting"
_ICLR.cc/2026/Conference — ICLR 2026 Conference Withdrawn Submission_

### Official Review · Reviewer_dCRg · 2025-10-17

**Soundness:** 3
**Presentation:** 3
**Contribution:** 2
**Rating:** 2
**Confidence:** 4

**Summary:**

This paper proposes UniFy, a linear framework for time series forecasting that aims to model non-stationary periodicity by alleviating what the authors call frequency competition, where the dominant frequency suppresses the secondary frequency. The approach introduces three components for separating and independently modeling different frequency subspaces before recombining them. The paper's contributions include a new perspective on frequency interference in time series and a lightweight, interpretable architecture. However, while the problem of non-stationary periodicity is real, its motivation is less compelling because existing frequency-domain methods already accommodate varying periodic components rather than assuming a fixed period.

**Strengths:**

1. This paper proposes a novel approach for modeling nonstationary periodicity in time series forecasting by applying adaptive frequency selection in the frequency domain.
2. This approach demonstrates reliable empirical performance on multiple real-world datasets, achieving significant improvements in both long-term and short-term forecasting tasks.
3. The paper is well-thought-out and provides a clear explanation of the proposed method.

**Weaknesses:**

1. This article does not explain why selecting specific frequencies or using learned masks improves time series forecasting performance. Given that the frequency domain is commonly used in signal processing, a stronger theoretical justification can be provided to explain why and how frequency selection can benefit forecasting models.
2. Existing frequency-domain models (e.g., those in [1, 2, 3, 4]) do not assume fixed periodicity, but rather accommodate varying periodicity. This makes the motivation for frequency competition somewhat problematic. The paper claims that the problem with existing models is that dominant frequencies suppress minor frequencies, but fails to convincingly demonstrate that this competition is a significant problem in methods that already allow for varying periodicity.
3. The related work section of this paper does not fully discuss the proposed method in relation to recent research in this area, in particular [1, 2, 3, 4]. These methods also deal with periodicity in time series, but in different ways.
4. The discrete Fourier transform (DFT) produces both positive and negative frequencies, while existing frequency-domain methods (such as those in [1, 2, 3, 4]) typically retain only non-negative frequencies. However, this paper retains both positive and negative frequencies in its decomposition. However, the motivation for retaining negative frequencies is not clearly explained.
5. This paper adopts a channel-independent approach, treating each variable as independent of the others. However, time series data often exhibit inter-variable dependencies (for example, in multivariate time series, variables often influence each other). Ignoring these dependencies can limit the effectiveness of the model, and the paper does not provide relevant analysis.

[1] Xu, Zhijian, et al. “FITS: Modeling Time Series with 10k Parameters.” International Conference on Learning Representations (ICLR), 2024.

[2] Zhang, et al. “Not All Frequencies Are Created Equal: Towards a Dynamic Fusion of Frequencies in Time-Series Forecasting.” ACM Multimedia (ACM MM), 2024.

[3] Wang, et al. “FreDF: Learning to Forecast in the Frequency Domain.” International Conference on Learning Representations (ICLR), 2025.

[4] Yi, et al. “Frequency-domain MLPs are More Effective Learners in Time Series Forecasting.” Conference on Neural Information Processing Systems (NeurIPS), 2023.

**Questions:**

1. Can the authors explain, either theoretically or empirically, why frequency selection via learned masks can improve prediction performance?
2. The authors claim that frequency contention is a problem in existing frequency-based methods. However, many recent methods (e.g., [1, 2, 3, 4]) already allow for dynamic, non-stationary periodicity. Can the authors explain why frequency contention remains a problem in these methods?
3. The paper lacks a detailed comparison with existing frequency-domain methods, particularly [1, 2, 3, 4]. How does the proposed method compare to these methods in terms of dynamic frequency modeling and handling nonstationary periodicity? Could the authors discuss the key differences between them and why their method offers an improvement?
4. What is the motivation behind retaining both positive and negative frequencies in the decomposition? Do negative frequencies provide any distinct advantage in forecasting over methods that only retain positive frequencies?
5. The current method treats each variable independently. However, in many real-world time series, there are causal relationships between variables. Does the author's method account for this causal relationship? If not, can the authors provide relevant results along the way on how this omission affects performance, especially when forecasting multivariate data with known dependencies between variables (such as traffic and ECLs datasets)?

[1] Xu, Zhijian, et al. “FITS: Modeling Time Series with 10k Parameters.” International Conference on Learning Representations (ICLR), 2024.

[2] Zhang, et al. “Not All Frequencies Are Created Equal: Towards a Dynamic Fusion of Frequencies in Time-Series Forecasting.” ACM Multimedia (ACM MM), 2024.

[3] Wang, et al. “FreDF: Learning to Forecast in the Frequency Domain.” International Conference on Learning Representations (ICLR), 2025.

[4] Yi, et al. “Frequency-domain MLPs are More Effective Learners in Time Series Forecasting.” Conference on Neural Information Processing Systems (NeurIPS), 2023.

---

> ### Author Response · Authors · 2025-11-25
> **Response to dCRg**
>
> ## W1|Q1: Theoretical Analysis of the Model
>
> **A1:**
>
> We thank the reviewer for raising this crucial concern. We will address it through a theoretical demonstration that explains both the phenomenon of frequency competition and why conventional architectures lack principled mechanisms to balance this competition.
>
> We provide a concise theoretical explanation showing why **any model that forces multiple frequencies to share the same latent space inherently induces frequency competition**, and why there exists no principled mechanism to balance this competition within such shared representations.
>
> Consider a minimal signal composed of two orthogonal frequency components:
>
> $x = a_1\phi_1 + a_2\phi_2,\quad
> \langle \phi_1,\phi_2\rangle = 0,\quad
> \sigma_1^2 > \sigma_2^2,$
>
> where $\sigma_i^2=\mathbb{E}[a_i^2]$ represents the energy of each component.
>
> When a model is constrained to encode both frequencies through a **single shared projection direction** $q$ (as occurs in standard linear layers, shared attention heads, or CNN/MLP channels), the gradient descent update under squared error loss follows:
>
> $\Delta q \propto C q,\qquad
> C = \sigma_1^2\phi_1\phi_1^\top + \sigma_2^2\phi_2\phi_2^\top.$
>
>
> Decomposing $q$ in the frequency basis as $q = \alpha_1\phi_1 + \alpha_2\phi_2$, we obtain the component-wise updates:
>
> $\Delta q_{\phi_1} \propto \sigma_1^2 \alpha_1,\qquad
> \Delta q_{\phi_2} \propto \sigma_2^2 \alpha_2.$
>
>
> Given the energy imbalance $\sigma_1^2 > \sigma_2^2$, the gradient magnitude is consistently larger along the dominant frequency direction:
>
> $\|\Delta q_{\phi_1}\| > \|\Delta q_{\phi_2}\|.$
>
>
> **This demonstrates that gradient dynamics systematically bias the shared representation toward the dominant frequency, inevitably suppressing the weaker component**—this is precisely the phenomenon we identify as *frequency competition*.
>
> Critically, this problematic dynamic persists regardless of architectural depth or nonlinearity, as long as different frequency components are forced to share the same latent dimensions. The fundamental issue is that **there exists no inherent gradient-based mechanism to protect or prioritize weaker frequencies within a shared representation space**.
>
> This theoretical insight directly motivates our AFS framework, which implements a principled solution:
> **Weak frequencies cannot be reliably learned unless the dominant frequency is first separated from the shared representation space.**
>
> By creating multiple adaptive subspaces, AFS ensures that each frequency component can be optimized independently, thereby providing the structural mechanism missing in conventional architectures to balance frequency competition and enable effective learning of both dominant and subtle spectral features.

---

> > ### Author Response · Authors · 2025-11-25
> >
> > ## W2|Q2: Importance of the Problem Addressed by Our Motivation
> >
> > **A2:**
> >
> > We sincerely appreciate your question.
> >
> > We constructed a toy dataset that effectively demonstrates frequency competition and non-stationarity in data, where non-stationarity specifically represents changing periodicity patterns. We tested several SOTA models (including both time-domain and frequency-domain approaches) on this toy dataset, with detailed results available in $\underline{\text{Appendix P (Rows 1566-1652)}}$.
> >
> > The results demonstrate that UniFy indeed achieves significant performance improvements in non-stationary scenarios. Additionally, we have included the performance of these models on public datasets in $\underline{\text{Appendix O (Rows 1283-1291)}}$, where UniFy consistently outperforms all competing methods across all benchmarks.

---

> > > ### Author Response · Authors · 2025-11-25
> > >
> > > ## W3|Q3: Supplementary Related Work
> > >
> > > **A3:**
> > >
> > > We appreciate the reviewer's valuable feedback. Below we clarify how UniFy differs from recent frequency-domain forecasting methods in design motivation and modeling principles:
> > >
> > > **TimeKAN** employs multi-scale pooling for frequency analysis, but pooling removes fine-grained spectral details. While effective for efficient KAN blocks, it doesn't explicitly address frequency competition and remains sensitive to hyperparameters.
> > >
> > > **FreTS** uses per-channel Fourier features primarily for cross-channel dependency modeling. However, its frequency processing still occurs in shared representation spaces where frequency interference persists.
> > >
> > > **FITS** applies low-pass filtering to extract stable periodic patterns, implicitly avoiding competition by discarding higher frequencies. However, it struggles with fluctuating mid-frequency structures and depends heavily on manual filter cutoff settings.
> > >
> > > **FreDF & FreqMoE** model frequency bins independently or with MoE architectures. While providing fine-grained decomposition, they assume frequency stability—a restrictive assumption for non-stationary data where frequency drift is common.
> > >
> > > In contrast, **UniFy** allocates frequency groups to adaptive subspaces, enabling flexible tracking of drifting spectral structures while preventing interference between dominant and secondary components. Rather than simply replicating dominant frequencies, AFS learns complementary frequency subspaces that capture important mid-to-high frequency information often missed by other methods.
> > >
> > > For the newly added baselines evaluated across {96, 192, 336, 720} prediction lengths, UniFy maintains superior overall performance.
> > >
> > > **Long-term forecasting**
> > >
> > > | 　 | Models | UniFy      (ours) |  | FITS |  | TimeKAN |  | FreDF |  | SparseTSF |  | FreqMoE |  |
> > > |:---:|:---:|:---:|:---:|:---:|:---:|:---:|:---:|:---:|:---:|:---:|:---:|:---:|:---:|
> > > | 　 | Metric | MSE | MAE | MSE | MAE | MSE | MAE | MSE | MAE | MSE | MAE | MSE | MAE  |
> > > | ETTm1 | Avg | 0.380  | 0.394  | 0.415  | 0.408  | 0.378  | 0.396  | 0.385  | 0.399  | 0.416  | 0.407  | 0.582  | 0.513  |
> > > | ETTm2 | Avg | 0.272  | 0.319  | 0.286  | 0.328  | 0.277  | 0.323  | 0.281  | 0.323  | 0.287  | 0.328  | 0.303  | 0.340  |
> > > | ETTh1 | Avg | 0.432  | 0.429  | 0.444  | 0.435  | 0.427  | 0.429  | 0.504  | 0.479  | 0.439  | 0.426  | 0.620  | 0.562  |
> > > | ETTh2 | Avg | 0.363  | 0.394  | 0.382  | 0.404  | 0.383  | 0.404  | 0.452  | 0.438  | 0.382  | 0.402  | 0.635  | 0.463  |
> > > | ECL | Avg | 0.171  | 0.262  | 0.223  | 0.307  | 0.197  | 0.286  | 0.198  | 0.293  | 0.223  | 0.296  | 0.179  | 0.271  |
> > > | Traffic | Avg | 0.467  | 0.311  | 0.657  | 0.430  | 0.577  | 0.373  | 0.613  | 0.344  | 0.637  | 0.379  | 1.070  | 0.534  |
> > > | Weather | Avg | 0.242  | 0.271  | 0.250  | 0.278  | 0.243  | 0.272  | 0.241  | 0.270  | 0.275  | 0.294  | 0.248  | 0.276  |
> > > | Exchange | Avg | 0.352  | 0.398  | 0.378  | 0.418  | 0.378  | 0.411  | 0.352  | 0.398  | 0.353  | 0.402  | 0.419  | 0.416  |
> > >
> > > **Short-term Forecasting**
> > >
> > > | 　 | Models | UniFy      (ours) |  | FITS |  | TimeKAN |  | FreDF |  | SparseTSF |  | FreqMoE |  |
> > > |:---:|:---:|:---:|:---:|:---:|:---:|:---:|:---:|:---:|:---:|:---:|:---:|:---:|:---:|
> > > | 　 | Metric | MSE | MAE | MSE | MAE | MSE | MAE | MSE | MAE | MSE | MAE | MSE | MAE  |
> > > | PEMS03 | Avg | 0.131  | 0.237  | 0.583  | 0.525  | 3072.730  | 34.805  | - | - | 0.586  | 0.533  | 0.792  | 0.709  |
> > > | PEMS04 | Avg | 0.106  | 0.216  | 0.675  | 0.586  | 4612.826  | 43.692  | - | - | 0.619  | 0.553  | 0.521  | 0.520  |
> > > | PEMS07 | Avg | 0.101  | 0.206  | 0.585  | 0.526  | 5959.166  | 50.314  | - | - | 0.609  | 0.549  | 0.887  | 0.682  |
> > > | PEMS08 | Avg | 0.126  | 0.238  | 0.691  | 0.584  | 3297.287  | 35.940  | - | - | 0.636  | 0.554  | 0.637  | 0.584  |
> > >
> > > Tips: The official implementation of FreDF is not compatible with the PEMS dataset, and we are unable to obtain a result.

---

> > > > ### Author Response · Authors · 2025-11-25
> > > >
> > > > **Few-Shot Forecasting**
> > > >
> > > > | 　 | Models | UniFy      (ours) |  | FITS |  | TimeKAN |  | FreDF |  | SparseTSF |  | FreqMoE |   |
> > > > |:---:|:---:|:---:|:---:|:---:|:---:|:---:|:---:|:---:|:---:|:---:|:---:|:---:|:---:|
> > > > | 　 | Metric | MSE | MAE | MSE | MAE | MSE | MAE | MSE | MAE | MSE | MAE | MSE | MAE  |
> > > > | ETTm1 | Avg | 0.389  | 0.398  | 0.628  | 0.521  | 0.389  | 0.400  | 0.439  | 0.439  | 0.475  | 0.442  | 1.082  | 0.596  |
> > > > | ETTm2 | Avg | 0.281  | 0.324  | 0.313  | 0.352  | 0.287  | 0.330  | 0.304  | 0.341  | 0.303  | 0.345  | 0.534  | 0.418  |
> > > > | ETTh1 | Avg | 0.450  | 0.437  | 1.169  | 0.730  | 0.459  | 0.443  | 0.616  | 0.539  | 0.481  | 0.462  | 4.286  | 1.000  |
> > > > | ETTh2 | Avg | 0.382  | 0.404  | 0.541  | 0.498  | 0.389  | 0.409  | 0.430  | 0.439  | 0.403  | 0.419  | 1.512  | 0.642  |
> > > >
> > > > **Zero-Shot Forecasting**
> > > >
> > > > | 　 | Models | UniFy      (ours) |  | FITS |  | TimeKAN |  | FreDF |  | SparseTSF |  | FreqMoE | |
> > > > |:---:|:---:|:---:|:---:|:---:|:---:|:---:|:---:|:---:|:---:|:---:|:---:|:---:|:---:|
> > > > | 　 | Metric | MSE | MAE | MSE | MAE | MSE | MAE | MSE | MAE | MSE | MAE | MSE | MAE  |
> > > > | ETTm1 - ETTm2 | Avg | 0.294  | 0.333  | 0.293  | 0.326  | 0.301  | 0.338  | 0.321  | 0.353  | 0.297  | 0.328  | 0.462  | 0.396  |
> > > > | ETTm2 - ETTm1 | Avg | 0.505  | 0.456  | 0.516  | 0.464  | 0.753  | 0.557  | 0.781  | 0.567  | 0.530  | 0.470  | 1.635  | 0.689  |
> > > > | ETTh2 - ETTh1 | Avg | 0.481  | 0.468  | 0.490  | 0.470  | 0.657  | 0.563  | 0.761  | 0.598  | 0.487  | 0.463  | 3.613  | 0.975  |
> > > > | ETTm1 - ETTh1 | Avg | 0.585  | 0.515  | 0.588  | 0.514  | 0.730  | 0.576  | 0.794  | 0.613  | 0.611  | 0.528  | 1.756  | 0.785  |
> > > > | ETTh1 - ETTm1 | Avg | 0.752  | 0.572  | 0.906  | 0.707  | 0.819  | 0.584  | 0.942  | 0.643  | 0.752  | 0.569  | 4.059  | 1.013  |
> > > > | ETTh2 - ETTm1 | Avg | 0.727  | 0.559  | 0.739  | 0.564  | 1.002  | 0.636  | 1.058  | 0.652  | 0.824  | 0.588  | 3.611  | 0.961  |

---

> > > > > ### Author Response · Authors · 2025-11-25
> > > > >
> > > > > ## W4|Q4: negative frequencies from DFT
> > > > >
> > > > > **A4:**
> > > > >
> > > > > Thank you for pointing this out. Our implementation uses the standard complex DFT:
> > > > >
> > > > > ```python
> > > > > freq_seq = torch.fft.fft(x, dim=1)   # shape [B, T], T frequency bins
> > > > > ```
> > > > >
> > > > > and applies a learnable real-valued mask freq_weights ∈ ℝ^T element-wise to all T frequency bins. As a result, we do not explicitly discard the “negative” frequencies; instead, we learn a frequency mask over the full DFT spectrum and then reconstruct the signal via IFFT:
> > > > >
> > > > > ```python
> > > > > selected = freq_seq * mask * residual_mask
> > > > > scale_component = torch.fft.ifft(selected, dim=1).real
> > > > > ```
> > > > > For real-valued time series, the DFT spectrum is conjugate-symmetric, so the negative frequencies do not carry additional independent information compared to the non-negative ones. Existing works that retain only non-negative frequencies mainly do so for parameter/complexity reduction (e.g., by using rFFT). In contrast, our choice to keep the full spectrum is primarily for implementation simplicity and a uniform parameterization over all DFT bins: we directly learn real-valued masks on the standard FFT output without manually splitting or folding positive and negative frequencies.
> > > > >
> > > > > Importantly, our method does not rely on any special modeling assumption about negative frequencies; it can be viewed as learning real-valued frequency filters over the full spectrum, which is equivalent in representational capacity to operating on the unique (non-negative) part of the spectrum for real signals.

---

> ### Author Response · Authors · 2025-11-25
>
> ## W5|Q5: Channel Independence Modelling in UniFy
>
> **A5:**
>
> Thank you for raising this important point about inter-variable dependencies.
>
> Although AFS appears to operate in a channel-wise manner, its frequency decomposition is implemented with **shared parameters across all channels**, so it is not strictly channel-independent. To analyze this more clearly, we treat AFS as a frequency-domain representation module.
>
> For each channel $c$, we construct a $K$-dimensional vector that describes how its energy is distributed over the learned frequency components:
> - For each component $k = 1,\dots,K$, the output of `divide` (`freq_scales[k]`) is reshaped from $[B \cdot C, T]$ to $[B, C, T]$.
> - We compute the average energy for channel $c$ over batch and time:
> $E\_c^{(k)} = \mathbb{E}\_{b,t}\left[ \left(\mathrm{scale}^{(k)}\_{b,c}(t)\right)^2 \right].$
>
> - The vector $\mathbf{h}_c = [E_c^{(1)}, \dots, E_c^{(K)}]$ is used as the **frequency-space representation** of channel $c$.
>
> We then perform the following analysis on a representative dataset:
>
> 1. Compute the Pearson correlation matrix between all channel pairs in the raw data, and form groups of **strongly correlated** channels using $|\rho| \ge 0.7$.
> 2. Feed the data into UniFy, extract $\mathbf{h}_c$ from AFS for each channel, and apply **t-SNE** to obtain a 2D embedding.
> 3. In the t-SNE plot, channels in the same high-correlation group (based on Pearson correlation) share the same color.
>
> The resulting visualization in $\underline{\text{Appendix S (Rows 1772-1819)}}$ shows that channels which are strongly correlated in the raw data tend to cluster together in the t-SNE space, while weakly correlated or isolated channels are more scattered. This indicates that the **shared frequency decomposition in AFS induces similar frequency-space representations for correlated variables**, even though the forecasting head is channel-wise.
>
> Therefore, UniFy does not ignore inter-variable dependencies: through shared frequency-domain parameters, it implicitly captures channel relationships rather than treating each variable as completely independent.
>
> Best regards,
>
> The Authors

---

> ### Comment · Reviewer_dCRg · 2025-11-27
>
> Thank you for the reviewers' replies, most of my concerns have been addressed. But the results of the relevant work I mentioned are still missing[1,2]. I will update my score after that.
>
> [1] Zhang, et al. “Not All Frequencies Are Created Equal: Towards a Dynamic Fusion of Frequencies in Time-Series Forecasting.” ACM Multimedia (ACM MM), 2024.
>
> [2] Yi, et al. “Frequency-domain MLPs are More Effective Learners in Time Series Forecasting.” Conference on Neural Information Processing Systems (NeurIPS), 2023.

---

> > ### Author Response · Authors · 2025-11-27
> >
> > ### Thank you very much for your timely response!
> >
> > We apologize that our previous clarification may not have been sufficiently clear, as the large number of additional baseline results we reported may have made the presentation difficult to follow. We would like to specifically point out that for FreDF[1], we have already included its results in $\underline{\text{Appendix O}}$. As for FreTS, we have provided the corresponding baseline results in the original manuscript $\underline{\text{Appendix N}}$. To clarify, we will present the full comparison results between UniFy and these two baselines again here.
> >
> > ### Long-Term Forecasting
> >
> > | 　 | Models | UniFy |  | FreDF |  | FreTS |  |
> > |---|---|---|---|---|---|---|---|
> > | 　 | Metric | MSE | MAE | MSE | MAE | MSE | MAE |
> > | ETTm1 | 96 | 0.315  | 0.355  | 0.324  | 0.367  | 0.338  | 0.375  |
> > | 　 | 192 | 0.360  | 0.379  | 0.365  | 0.387  | 0.381  | 0.397  |
> > | 　 | 336 | 0.390  | 0.402  | 0.391  | 0.405  | 0.420  | 0.423  |
> > | 　 | 720 | 0.456  | 0.438  | 0.459  | 0.436  | 0.480  | 0.458  |
> > | 　 | Avg | 0.380  | 0.394  | 0.385  | 0.399  | 0.405  | 0.413  |
> > | ETTm2 | 96 | 0.169  | 0.253  | 0.175  | 0.257  | 0.186  | 0.276  |
> > | 　 | 192 | 0.234  | 0.295  | 0.241  | 0.299  | 0.259  | 0.324  |
> > | 　 | 336 | 0.292  | 0.334  | 0.303  | 0.341  | 0.347  | 0.385  |
> > | 　 | 720 | 0.393  | 0.393  | 0.405  | 0.396  | 0.554  | 0.508  |
> > | 　 | Avg | 0.272  | 0.319  | 0.281  | 0.323  | 0.337  | 0.373  |
> > | ETTh1 | 96 | 0.372  | 0.394  | 0.402  | 0.416  | 0.394  | 0.406  |
> > | 　 | 192 | 0.426  | 0.423  | 0.459  | 0.450  | 0.451  | 0.442  |
> > | 　 | 336 | 0.463  | 0.435  | 0.510  | 0.482  | 0.510  | 0.479  |
> > | 　 | 720 | 0.467  | 0.464  | 0.646  | 0.567  | 0.567  | 0.537  |
> > | 　 | Avg | 0.432  | 0.429  | 0.504  | 0.479  | 0.481  | 0.466  |
> > | ETTh2 | 96 | 0.281  | 0.335  | 0.341  | 0.373  | 0.334  | 0.388  |
> > | 　 | 192 | 0.366  | 0.391  | 0.462  | 0.435  | 0.450  | 0.456  |
> > | 　 | 336 | 0.386  | 0.413  | 0.537  | 0.478  | 0.461  | 0.467  |
> > | 　 | 720 | 0.418  | 0.438  | 0.468  | 0.467  | 0.790  | 0.634  |
> > | 　 | Avg | 0.363  | 0.394  | 0.452  | 0.438  | 0.509  | 0.486  |
> > | ECL | 96 | 0.142  | 0.237  | 0.150  | 0.242  | 0.183  | 0.265  |
> > | 　 | 192 | 0.155  | 0.247  | 0.161  | 0.253  | 0.180  | 0.267  |
> > | 　 | 336 | 0.171  | 0.263  | 0.220  | 0.321  | 0.191  | 0.283  |
> > | 　 | 720 | 0.215  | 0.302  | 0.259  | 0.354  | 0.228  | 0.322  |
> > | 　 | Avg | 0.171  | 0.262  | 0.198  | 0.293  | 0.196  | 0.284  |
> > | Traffic | 96 | 0.441  | 0.302  | 0.590  | 0.334  | 0.556  | 0.372  |
> > | 　 | 192 | 0.452  | 0.301  | 0.600  | 0.341  | 0.574  | 0.374  |
> > | 　 | 336 | 0.468  | 0.307  | 0.614  | 0.349  | 0.606  | 0.384  |
> > | 　 | 720 | 0.508  | 0.332  | 0.648  | 0.353  | 0.664  | 0.414  |
> > | 　 | Avg | 0.467  | 0.311  | 0.613  | 0.344  | 0.600  | 0.386  |
> > | Weather | 96 | 0.156  | 0.203  | 0.157  | 0.208  | 0.177  | 0.208  |
> > | 　 | 192 | 0.206  | 0.248  | 0.205  | 0.246  | 0.219  | 0.247  |
> > | 　 | 336 | 0.262  | 0.290  | 0.259  | 0.287  | 0.277  | 0.295  |
> > | 　 | 720 | 0.343  | 0.343  | 0.341  | 0.339  | 0.340  | 0.340  |
> > | 　 | Avg | 0.242  | 0.271  | 0.241  | 0.270  | 0.253  | 0.273  |
> > | Exchange | 96 | 0.081  | 0.198  | 0.082  | 0.199  | 0.090  | 0.216  |
> > | 　 | 192 | 0.172  | 0.293  | 0.172  | 0.294  | 0.192  | 0.322  |
> > | 　 | 336 | 0.329  | 0.414  | 0.316  | 0.405  | 0.399  | 0.461  |
> > | 　 | 720 | 0.826  | 0.687  | 0.838  | 0.694  | 0.789  | 0.646  |
> > | 　 | Avg | 0.352  | 0.398  | 0.352  | 0.398  | 0.368  | 0.411  |
> >
> > ### Short-Term Forecasting
> >
> > | 　 | Models | UniFy |  | FreDF |  | FreTS |  |
> > |---|---|---|---|---|---|---|---|
> > | 　 | Metric | MSE | MAE | MSE | MAE | MSE | MAE |
> > | PEMS03 | 12 | 0.068  | 0.174  | - | - | 0.080  | 0.189  |
> > | 　 | 24 | 0.093  | 0.203  | - | - | 0.125  | 0.236  |
> > | 　 | 48 | 0.143  | 0.256  | - | - | 0.194  | 0.300  |
> > | 　 | 96 | 0.218  | 0.314  | - | - | 0.271  | 0.362  |
> > | 　 | Avg | 0.131  | 0.237  | - | - | 0.168  | 0.272  |
> > | PEMS04 | 12 | 0.070  | 0.173  | - | - | 0.095  | 0.206  |
> > | 　 | 24 | 0.086  | 0.194  | - | - | 0.140  | 0.255  |
> > | 　 | 48 | 0.110  | 0.223  | - | - | 0.214  | 0.321  |
> > | 　 | 96 | 0.158  | 0.273  | - | - | 0.298  | 0.380  |
> > | 　 | Avg | 0.106  | 0.216  | - | - | 0.187  | 0.291  |
> > | PEMS07 | 12 | 0.059  | 0.158  | - | - | 0.075  | 0.182  |
> > | 　 | 24 | 0.077  | 0.183  | - | - | 0.124  | 0.237  |
> > | 　 | 48 | 0.107  | 0.217  | - | - | 0.218  | 0.318  |
> > | 　 | 96 | 0.159  | 0.267  | - | - | 0.302  | 0.379  |
> > | 　 | Avg | 0.101  | 0.206  | - | - | 0.180  | 0.279  |
> > | PEMS08 | 12 | 0.068  | 0.176  | - | - | 0.099  | 0.202  |
> > | 　 | 24 | 0.091  | 0.202  | - | - | 0.172  | 0.254  |
> > | 　 | 48 | 0.135  | 0.253  | - | - | 0.302  | 0.331  |
> > | 　 | 96 | 0.209  | 0.320  | - | - | 0.550  | 0.448  |
> > | 　 | Avg | 0.126  | 0.238  | - | - | 0.281  | 0.309  |
> >
> > Tips: The official FreDF implementation is not compatible with PEMS and obtaining results would require non-trivial code changes that might deviate from the authors’ design, so we do not report FreDF results on PEMS and only evaluate it on officially supported datasets.

---

> ### Author Response · Authors · 2025-11-27
>
> ### Few-Shot Forecasting
>
> | 　 | Models | UniFy |  | FreDF |  | FreTS |  |
> |---|---|---|---|---|---|---|---|
> | 　 | Metric | MSE | MAE | MSE | MAE | MSE | MAE |
> | ETTm1 | 96 | 0.325  | 0.362  | 0.357  | 0.392  | 0.429  | 0.433  |
> | 　 | 192 | 0.369  | 0.385  | 0.422  | 0.430  | 0.493  | 0.458  |
> | 　 | 336 | 0.397  | 0.402  | 0.433  | 0.437  | 0.592  | 0.496  |
> | 　 | 720 | 0.465  | 0.441  | 0.545  | 0.495  | 0.663  | 0.549  |
> | 　 | Avg | 0.389  | 0.398  | 0.439  | 0.439  | 0.544  | 0.484  |
> | ETTm2 | 96 | 0.175  | 0.257  | 0.189  | 0.272  | 0.215  | 0.305  |
> | 　 | 192 | 0.238  | 0.297  | 0.258  | 0.316  | 0.328  | 0.367  |
> | 　 | 336 | 0.308  | 0.344  | 0.327  | 0.358  | 0.360  | 0.412  |
> | 　 | 720 | 0.402  | 0.399  | 0.440  | 0.418  | 0.506  | 0.493  |
> | 　 | Avg | 0.281  | 0.324  | 0.304  | 0.341  | 0.352  | 0.394  |
> | ETTh1 | 96 | 0.389  | 0.400  | 0.553  | 0.502  | 0.591  | 0.518  |
> | 　 | 192 | 0.443  | 0.429  | 0.622  | 0.530  | 0.645  | 0.551  |
> | 　 | 336 | 0.480  | 0.446  | 0.595  | 0.528  | 0.678  | 0.575  |
> | 　 | 720 | 0.489  | 0.471  | 0.695  | 0.595  | 1.040  | 0.777  |
> | 　 | Avg | 0.450  | 0.437  | 0.616  | 0.539  | 0.739  | 0.605  |
> | ETTh2 | 96 | 0.300  | 0.348  | 0.354  | 0.389  | 0.399  | 0.441  |
> | 　 | 192 | 0.384  | 0.398  | 0.432  | 0.435  | 0.581  | 0.484  |
> | 　 | 336 | 0.412  | 0.424  | 0.460  | 0.460  | 0.657  | 0.543  |
> | 　 | 720 | 0.431  | 0.445  | 0.474  | 0.471  | 0.892  | 0.673  |
> | 　 | Avg | 0.382  | 0.404  | 0.430  | 0.439  | 0.632  | 0.535  |
>
> ### Zero-Shot Forecasting
>
> | 　 | Models | UniFy |  | FreDF |  | FreTS |  |
> |---|---|---|---|---|---|---|---|
> | 　 | Metric | MSE | MAE | MSE | MAE | MSE | MAE |
> | ETTm1 - ETTm2 | 96 | 0.192  | 0.281  | 0.212  | 0.289  | 0.203  | 0.304  |
> | 　 | 192 | 0.255  | 0.307  | 0.282  | 0.332  | 0.277  | 0.356  |
> | 　 | 336 | 0.314  | 0.344  | 0.345  | 0.370  | 0.361  | 0.409  |
> | 　 | 720 | 0.414  | 0.400  | 0.444  | 0.422  | 0.511  | 0.498  |
> | 　 | Avg | 0.294  | 0.333  | 0.321  | 0.353  | 0.338  | 0.392  |
> | ETTm2 - ETTm1 | 96 | 0.454  | 0.427  | 0.712  | 0.538  | 0.957  | 0.544  |
> | 　 | 192 | 0.477  | 0.442  | 0.672  | 0.531  | 0.948  | 0.552  |
> | 　 | 336 | 0.524  | 0.461  | 0.724  | 0.552  | 0.927  | 0.561  |
> | 　 | 720 | 0.566  | 0.492  | 1.014  | 0.647  | 0.803  | 0.551  |
> | 　 | Avg | 0.505  | 0.456  | 0.781  | 0.567  | 0.909  | 0.552  |
> | ETTh2 - ETTh1 | 96 | 0.416  | 0.419  | 0.699  | 0.572  | 0.506  | 0.471  |
> | 　 | 192 | 0.463  | 0.445  | 0.710  | 0.581  | 0.547  | 0.501  |
> | 　 | 336 | 0.499  | 0.466  | 0.790  | 0.614  | 0.580  | 0.519  |
> | 　 | 720 | 0.547  | 0.542  | 0.844  | 0.625  | 0.615  | 0.572  |
> | 　 | Avg | 0.481  | 0.468  | 0.761  | 0.598  | 0.562  | 0.516  |
> | ETTm1 - ETTh1 | 96 | 0.575  | 0.501  | 0.794  | 0.594  | 1.543  | 0.781  |
> | 　 | 192 | 0.579  | 0.508  | 0.793  | 0.604  | 1.049  | 0.663  |
> | 　 | 336 | 0.604  | 0.527  | 0.792  | 0.622  | 1.015  | 0.668  |
> | 　 | 720 | 0.583  | 0.524  | 0.797  | 0.630  | 0.834  | 0.645  |
> | 　 | Avg | 0.585  | 0.515  | 0.794  | 0.613  | 1.110  | 0.689  |
> | ETTh1 - ETTm1 | 96 | 0.726  | 0.549  | 1.031  | 0.657  | 0.840  | 0.601  |
> | 　 | 192 | 0.750  | 0.567  | 0.974  | 0.655  | 0.791  | 0.596  |
> | 　 | 336 | 0.749  | 0.579  | 0.900  | 0.634  | 0.768  | 0.586  |
> | 　 | 720 | 0.783  | 0.594  | 0.861  | 0.625  | 0.779  | 0.614  |
> | 　 | Avg | 0.752  | 0.572  | 0.942  | 0.643  | 0.795  | 0.599  |
> | ETTh2 - ETTm1 | 96 | 0.685  | 0.522  | 1.011  | 0.632  | 0.832  | 0.571  |
> | 　 | 192 | 0.689  | 0.534  | 1.133  | 0.658  | 0.839  | 0.587  |
> | 　 | 336 | 0.741  | 0.560  | 1.063  | 0.656  | 0.788  | 0.591  |
> | 　 | 720 | 0.794  | 0.618  | 1.026  | 0.662  | 0.841  | 0.639  |
> | 　 | Avg | 0.727  | 0.559  | 1.058  | 0.652  | 0.825  | 0.597  |
>
> FreDF is an excellent work that offers many insightful contributions, and its design provided important inspiration for the initial development of UniFy. We have added a detailed discussion in $\underline{\text{Appendix Q}}$ to clarify how UniFy differs in its research focus from these strong baselines.
>
> If you have any further questions, we would be very happy to discuss them.
>
> Best regards,
>
> The Authors
>
> [1] Zhang, et al. “Not All Frequencies Are Created Equal: Towards a Dynamic Fusion of Frequencies in Time-Series Forecasting.” ACM Multimedia (ACM MM), 2024.
>
> [2] Yi, et al. “Frequency-domain MLPs are More Effective Learners in Time Series Forecasting.” Conference on Neural Information Processing Systems (NeurIPS), 2023.

---

### Official Review · Reviewer_gRZs · 2025-10-20

**Soundness:** 3
**Presentation:** 2
**Contribution:** 2
**Rating:** 4
**Confidence:** 4

**Summary:**

This paper focuses on the forecasting bottleneck caused by non-stationary periodicity in multivariate time series and identifies frequency competition (where the dominant frequency suppresses secondary yet informative components, and secondary frequencies in turn interfere with the dominant one) as the root cause of underfitting/overfitting. To address this problem, it proposes the nearly purely linear UniFy. This design explicitly mitigates interference between dominant and secondary frequencies, preserving stable primary periodicity while capturing time-varying details; experiments demonstrate the effectiveness of the approach. The paper claims that "there is a key gap in the field of TSF: existing approaches lack a principled mechanism to balance accurate modeling of the dominant component against interference from secondary components".

**Strengths:**

1. This paper addresses frequency competition through an AFS->ILM->MSC pipeline, reducing interference between dominant and secondary frequencies.
2. The proposed method uses an almost purely linear architecture with low inference/training cost and strong scalability; ILM assigns an independent linear projection to each scale to cut computation and avoid subspace interference.
3. This paper achieves good performance on 12 real-world datasets across long/short horizons and zero/few-shot settings.

**Weaknesses:**

1. This article proposes a key gap in the field of TSF: existing approaches lack a principled mechanism to balance accurate modeling of the dominant component against interference from secondary components. But I think this gap is not very reasonable because there are many methods [1-4] that explicitly separate and independently process different frequency components of time series.
2. Many papers[1-4] and the method proposed in this article are similar (both decompose the time series into multiple frequencies and process them separately before fusing the output), but this article does not discuss it. The author needs to supplement the differences between the methods used in this article and these methods.
3. The authors do not explicitly state the input  length in the main text; instead, they place it in a hard-to-find appendix, which reduces readers’ efficiency in accessing key information.

4. Why does the AFS module use a multi round mask selection mechanism to gradually decompose the spectrum into multiple subspaces. Isn't it simpler and more effective to directly decompose a time series into different frequency components through Fourier transform？ The necessity and rationality of the author's use of AFS modules need further elaboration.


[1] Not all frequencies are created equal: Towards a dynamic fusion of frequencies in time-series forecasting

[2] TimeKAN: KAN-based Frequency Decomposition Learning Architecture for Long-term Time Series Forecasting

[3] FreqMoE: Enhancing Time Series Forecasting through Frequency Decomposition Mixture of Experts

[4] Frequency-domain MLPs are More Effective Learners in Time Series Forecasting

**Questions:**

See weakness.

---

> ### Author Response · Authors · 2025-11-25
> **Response to gRZs**
>
> ## W1: Distinction from Works that Explicitly Separate and Independently Process Different Frequency Components of Time Series
>
> **A1:**
>
> We thank the reviewer for raising this crucial concern, and we will address it through a theoretical demonstration.
>
> We provide a concise theoretical explanation showing why **any model that forces multiple frequencies to share the same latent space inherently induces frequency competition**, and why there lacks a principled mechanism to balance this competition.
>
> Consider a simple two-frequency signal:
>
> $x = a_1\phi_1 + a_2\phi_2,\quad
> \langle \phi_1,\phi_2\rangle = 0,\quad
> \sigma_1^2 > \sigma_2^2,$
>
> where $\sigma_i^2=\mathbb{E}[a_i^2]$ denotes the energy of each component.
>
> If a model must encode both frequencies through **one shared projection direction** $q$ (as in linear layers, shared attention heads, CNN/MLP channels), then under squared loss the gradient descent update satisfies:
>
> $\Delta q \propto C q,\qquad
> C = \sigma_1^2\phi_1\phi_1^\top + \sigma_2^2\phi_2\phi_2^\top.$
>
>
> Expanding $q = \alpha_1\phi_1 + \alpha_2\phi_2$, we obtain:
>
> $\Delta q_{\phi_1} \propto \sigma_1^2 \alpha_1,\qquad
> \Delta q_{\phi_2} \propto \sigma_2^2 \alpha_2.$
>
>
> Since $\sigma_1^2 > \sigma_2^2$, the gradient magnitude along the dominant frequency is always larger:
>
> $\|\Delta q_{\phi_1}\| > \|\Delta q_{\phi_2}\|.$
>
>
> **Thus, gradient dynamics systematically bias the shared representation toward the dominant frequency, causing the weaker component to be suppressed**—this is exactly the phenomenon we refer to as *frequency competition*.
> This effect persists regardless of the depth or nonlinearity of the architecture as long as frequencies share the same latent dimensions.
>
> Based on this proof, we wish to clarify that our objective is not to design a solution that independently processes different frequency components of time series, but rather to better model frequency components within each subspace. This fundamentally differs from related works, and specific details will be further discussed in **A2**.

---

> > ### Author Response · Authors · 2025-11-25
> >
> > ## W2: Supplementary Related Work
> >
> > **A2:**
> >
> > We appreciate the reviewer's valuable feedback. Below we clarify how UniFy differs from recent frequency-domain forecasting methods in design motivation and modeling principles:
> >
> > **TimeKAN** employs multi-scale pooling for frequency analysis, but pooling removes fine-grained spectral details. While effective for efficient KAN blocks, it doesn't explicitly address frequency competition and remains sensitive to hyperparameters.
> >
> > **FreTS** uses per-channel Fourier features primarily for cross-channel dependency modeling. However, its frequency processing still occurs in shared representation spaces where frequency interference persists.
> >
> > **FITS** applies low-pass filtering to extract stable periodic patterns, implicitly avoiding competition by discarding higher frequencies. However, it struggles with fluctuating mid-frequency structures and depends heavily on manual filter cutoff settings.
> >
> > **FreDF & FreqMoE** model frequency bins independently or with MoE architectures. While providing fine-grained decomposition, they assume frequency stability—a restrictive assumption for non-stationary data where frequency drift is common.
> >
> > In contrast, **UniFy** allocates frequency groups to adaptive subspaces, enabling flexible tracking of drifting spectral structures while preventing interference between dominant and secondary components. Rather than simply replicating dominant frequencies, AFS learns complementary frequency subspaces that capture important mid-to-high frequency information often missed by other methods.
> >
> > For the newly added baselines evaluated across {96, 192, 336, 720} prediction lengths, UniFy maintains superior overall performance.
> >
> > **Long-term forecasting**
> >
> > | 　 | Models | UniFy      (ours) |  | FITS |  | TimeKAN |  | FreDF |  | SparseTSF |  | FreqMoE |  |
> > |:---:|:---:|:---:|:---:|:---:|:---:|:---:|:---:|:---:|:---:|:---:|:---:|:---:|:---:|
> > | 　 | Metric | MSE | MAE | MSE | MAE | MSE | MAE | MSE | MAE | MSE | MAE | MSE | MAE  |
> > | ETTm1 | Avg | 0.380  | 0.394  | 0.415  | 0.408  | 0.378  | 0.396  | 0.385  | 0.399  | 0.416  | 0.407  | 0.582  | 0.513  |
> > | ETTm2 | Avg | 0.272  | 0.319  | 0.286  | 0.328  | 0.277  | 0.323  | 0.281  | 0.323  | 0.287  | 0.328  | 0.303  | 0.340  |
> > | ETTh1 | Avg | 0.432  | 0.429  | 0.444  | 0.435  | 0.427  | 0.429  | 0.504  | 0.479  | 0.439  | 0.426  | 0.620  | 0.562  |
> > | ETTh2 | Avg | 0.363  | 0.394  | 0.382  | 0.404  | 0.383  | 0.404  | 0.452  | 0.438  | 0.382  | 0.402  | 0.635  | 0.463  |
> > | ECL | Avg | 0.171  | 0.262  | 0.223  | 0.307  | 0.197  | 0.286  | 0.198  | 0.293  | 0.223  | 0.296  | 0.179  | 0.271  |
> > | Traffic | Avg | 0.467  | 0.311  | 0.657  | 0.430  | 0.577  | 0.373  | 0.613  | 0.344  | 0.637  | 0.379  | 1.070  | 0.534  |
> > | Weather | Avg | 0.242  | 0.271  | 0.250  | 0.278  | 0.243  | 0.272  | 0.241  | 0.270  | 0.275  | 0.294  | 0.248  | 0.276  |
> > | Exchange | Avg | 0.352  | 0.398  | 0.378  | 0.418  | 0.378  | 0.411  | 0.352  | 0.398  | 0.353  | 0.402  | 0.419  | 0.416  |
> >
> > **Short-term Forecasting**
> >
> > | 　 | Models | UniFy      (ours) |  | FITS |  | TimeKAN |  | FreDF |  | SparseTSF |  | FreqMoE |  |
> > |:---:|:---:|:---:|:---:|:---:|:---:|:---:|:---:|:---:|:---:|:---:|:---:|:---:|:---:|
> > | 　 | Metric | MSE | MAE | MSE | MAE | MSE | MAE | MSE | MAE | MSE | MAE | MSE | MAE  |
> > | PEMS03 | Avg | 0.131  | 0.237  | 0.583  | 0.525  | 3072.730  | 34.805  | - | - | 0.586  | 0.533  | 0.792  | 0.709  |
> > | PEMS04 | Avg | 0.106  | 0.216  | 0.675  | 0.586  | 4612.826  | 43.692  | - | - | 0.619  | 0.553  | 0.521  | 0.520  |
> > | PEMS07 | Avg | 0.101  | 0.206  | 0.585  | 0.526  | 5959.166  | 50.314  | - | - | 0.609  | 0.549  | 0.887  | 0.682  |
> > | PEMS08 | Avg | 0.126  | 0.238  | 0.691  | 0.584  | 3297.287  | 35.940  | - | - | 0.636  | 0.554  | 0.637  | 0.584  |
> >
> >
> > Tips: The official implementation of FreDF is not compatible with the PEMS dataset, and we are unable to obtain a result.

---

> > > ### Author Response · Authors · 2025-11-25
> > >
> > > **Few-Shot Forecasting**
> > >
> > > | 　 | Models | UniFy      (ours) |  | FITS |  | TimeKAN |  | FreDF |  | SparseTSF |  | FreqMoE |   |
> > > |:---:|:---:|:---:|:---:|:---:|:---:|:---:|:---:|:---:|:---:|:---:|:---:|:---:|:---:|
> > > | 　 | Metric | MSE | MAE | MSE | MAE | MSE | MAE | MSE | MAE | MSE | MAE | MSE | MAE  |
> > > | ETTm1 | Avg | 0.389  | 0.398  | 0.628  | 0.521  | 0.389  | 0.400  | 0.439  | 0.439  | 0.475  | 0.442  | 1.082  | 0.596  |
> > > | ETTm2 | Avg | 0.281  | 0.324  | 0.313  | 0.352  | 0.287  | 0.330  | 0.304  | 0.341  | 0.303  | 0.345  | 0.534  | 0.418  |
> > > | ETTh1 | Avg | 0.450  | 0.437  | 1.169  | 0.730  | 0.459  | 0.443  | 0.616  | 0.539  | 0.481  | 0.462  | 4.286  | 1.000  |
> > > | ETTh2 | Avg | 0.382  | 0.404  | 0.541  | 0.498  | 0.389  | 0.409  | 0.430  | 0.439  | 0.403  | 0.419  | 1.512  | 0.642  |
> > >
> > > **Zero-Shot Forecasting**
> > >
> > > | 　 | Models | UniFy      (ours) |  | FITS |  | TimeKAN |  | FreDF |  | SparseTSF |  | FreqMoE | |
> > > |:---:|:---:|:---:|:---:|:---:|:---:|:---:|:---:|:---:|:---:|:---:|:---:|:---:|:---:|
> > > | 　 | Metric | MSE | MAE | MSE | MAE | MSE | MAE | MSE | MAE | MSE | MAE | MSE | MAE  |
> > > | ETTm1 - ETTm2 | Avg | 0.294  | 0.333  | 0.293  | 0.326  | 0.301  | 0.338  | 0.321  | 0.353  | 0.297  | 0.328  | 0.462  | 0.396  |
> > > | ETTm2 - ETTm1 | Avg | 0.505  | 0.456  | 0.516  | 0.464  | 0.753  | 0.557  | 0.781  | 0.567  | 0.530  | 0.470  | 1.635  | 0.689  |
> > > | ETTh2 - ETTh1 | Avg | 0.481  | 0.468  | 0.490  | 0.470  | 0.657  | 0.563  | 0.761  | 0.598  | 0.487  | 0.463  | 3.613  | 0.975  |
> > > | ETTm1 - ETTh1 | Avg | 0.585  | 0.515  | 0.588  | 0.514  | 0.730  | 0.576  | 0.794  | 0.613  | 0.611  | 0.528  | 1.756  | 0.785  |
> > > | ETTh1 - ETTm1 | Avg | 0.752  | 0.572  | 0.906  | 0.707  | 0.819  | 0.584  | 0.942  | 0.643  | 0.752  | 0.569  | 4.059  | 1.013  |
> > > | ETTh2 - ETTm1 | Avg | 0.727  | 0.559  | 0.739  | 0.564  | 1.002  | 0.636  | 1.058  | 0.652  | 0.824  | 0.588  | 3.611  | 0.961  |

---

> > > > ### Author Response · Authors · 2025-11-25
> > > >
> > > > ## W3: Lack of Input length in the main test.
> > > >
> > > > **A3:**
> > > >
> > > > We thank the reviewer for pointing out this oversight. We apologize for the inconvenience caused by placing the input-length specification only in the appendix. In the revised version, we have moved the input-length information (including the default setting of 96 for all datasets) directly into the main text ($\underline{\text{rows 298-299}}$) to ensure that readers can easily access this key experimental detail.

---

> ### Author Response · Authors · 2025-11-25
>
> ## W4: The necessity and rationality of AFS modules
> **A4**
>
> I sincerely thank the reviewer for their valuable comments.
>
> First, according to our gradient dynamics analysis provided in Q1, high-energy frequency components inherently produce the strongest gradient signals, leading to faster optimization. This means the dominant frequency contributes most significantly to the final error—an objective fact. Our key argument is not that we "directly decompose a time series into different frequency components," but rather that:
>
> **Weak frequencies cannot be reliably learned unless the dominant frequency is first separated from the shared representation space.**
>
> In A3, FreDF and FreqMoE represent exactly the type of models described by the reviewer. We have already elaborated on our distinctions from these approaches, and experimental results further demonstrate UniFy's superiority.
>
> Best regards,
>
> The Authors

---

### Official Review · Reviewer_wFdR · 2025-10-23

**Soundness:** 2
**Presentation:** 3
**Contribution:** 2
**Rating:** 4
**Confidence:** 3

**Summary:**

This paper proposes a lightweight linear framework, UniFy, for time series forecasting with non-stationary periodicity. The authors point out that existing methods often suffer from frequency competition, where dominant frequencies suppress secondary but informative ones, leading to biased periodic modeling. To address this issue, UniFy introduces three modules: AFS (Adaptive Frequency Selector), ILM (Independent Linear Modeler), and MSC (Multi-Subspace Calibration). Experiments on multiple datasets show good performance improvements.

**Strengths:**

1. The concept of frequency competition is clearly presented. The authors analyze the problem from a frequency-domain perspective and propose a quantitative metric, High-Competition Impact (HCI), which improves theoretical interpretability.

2. The framework is logically designed and modular.

3. The experiments cover various datasets and tasks, and the authors provide reproducible code.

**Weaknesses:**

1. The paper does not clearly explain how the model identifies or distinguishes dominant and secondary frequencies, making it hard to understand how frequency competition is alleviated.

2. All decomposition rounds use the same structure and loss without any orthogonality or hierarchy constraints, so it is unclear how the model ensures complementary or non-overlapping frequency components.

3. The meaning and necessity of the residual update and multi-round decomposition (R rounds) are not well explained.

4. The appendix shows that when the number of decomposition rounds $R=1$, the model achieves the best performance. If the first round mainly extracts dominant frequencies, this implies that dominant frequency energy contributes the most, which seems inconsistent with the paper’s claim that multi-round decomposition alleviates frequency competition.

**Questions:**

see weakness

---

> ### Author Response · Authors · 2025-11-25
> **Response to wFdR**
>
> ## W1: How to understand frequency competition
>
> **A1:**
>
> We thank the reviewer for raising this fundamental question regarding frequency competition.
>
> To clarify the necessity of our Adaptive Frequency Separation (AFS) module, we provide a theoretical demonstration showing why **any model architecture that forces multiple frequency components to share the same latent representation space inherently suffers from frequency competition**.
>
> Consider a minimal signal composed of two orthogonal frequency components:
>
> $x = a_1\phi_1 + a_2\phi_2,\quad
> \langle \phi_1,\phi_2\rangle = 0,\quad
> \sigma_1^2 > \sigma_2^2,$
>
> where $\sigma_i^2=\mathbb{E}[a_i^2]$ represents the energy of each frequency component.
>
> When a model is constrained to encode both frequencies through a **single shared projection direction** $q$ (as occurs in standard linear layers, shared attention heads, or CNN/MLP channels), the gradient descent update under squared error loss follows:
>
> $\Delta q \propto C q,\qquad
> C = \sigma_1^2\phi_1\phi_1^\top + \sigma_2^2\phi_2\phi_2^\top.$
>
>
> Decomposing $q$ in the frequency basis as $q = \alpha_1\phi_1 + \alpha_2\phi_2$, the gradient updates along each component direction are:
>
> $\Delta q_{\phi_1} \propto \sigma_1^2 \alpha_1,\qquad
> \Delta q_{\phi_2} \propto \sigma_2^2 \alpha_2.$
>
>
> Given that $\sigma_1^2 > \sigma_2^2$, the gradient magnitude is consistently larger along the dominant frequency direction:
>
> $\|\Delta q_{\phi_1}\| > \|\Delta q_{\phi_2}\|.$
>
>
> **This demonstrates that gradient dynamics systematically bias the shared representation toward the dominant frequency, inevitably causing suppression of the weaker spectral component**—this is precisely the phenomenon we identify as *frequency competition*.
>
> Crucially, this problematic dynamic persists regardless of the depth or nonlinearity of the architecture, as long as different frequency components are forced to share the same latent dimensions.
>
> Our proposed AFS framework directly resolves this structural limitation by creating **multiple adaptive subspaces**, enabling different frequency groups to be optimized independently without destructive interference. This theoretical insight is empirically validated through our gradient-based attribution analysis ($\underline{\text{Rows 419-464}}$), which confirms that AFS effectively separates components according to their spectral energy distribution. Additional supporting evidence and extended discussion are provided in $\underline{\text{Appendix M (Rows 1242-1273)}}$.

---

> > ### Author Response · Authors · 2025-11-25
> >
> > ## W2: How the Model Ensures Complementary and Non-Overlapping Frequency Components
> >
> > **A2:**
> >
> > We sincerely appreciate this insightful question.
> >
> > Based on the derivation in A1, we can understand the fundamental mechanism of AFS, where multi-round selection alleviates frequency competition. In implementation, AFS ensures complementarity not through explicit regularization, but via the **self-avoidance mechanism of residual masking**. This provides an implicit but powerful structural constraint:
> >
> > AFS performs sequential residual masking:
> >
> > $X_{\text{res}}^{(k+1)} = X_{\text{res}}^{(k)} - M^{(k)} \odot X_{\text{res}}^{(k)},$
> >
> > Thus each round receives only the remaining spectrum:
> >
> > $X_{\text{res}}^{(k+1)} = (1 - M^{(k)}) \odot X_{\text{res}}^{(k)}.$
> >
> >
> > If a frequency has been selected in an early round, the residual mask forces later rounds to receive near-zero energy at that frequency. This structurally enforces orthogonal-like constraints, ensuring non-overlapping components in the ILM.

---

> > > ### Author Response · Authors · 2025-11-25
> > >
> > > ## W3: Meaning and necessity of the residual update and multi-round decomposition
> > >
> > > **A3:**
> > >
> > > The proof in A1 has already indicated that the primary significance of AFS is to address the issue that "**Gradient dynamics systematically bias the shared representation toward the dominant frequency, causing the weaker component to be suppressed**."
> > >
> > > Meanwhile, we have conducted extensive ablation studies in the main text, all of which demonstrate its effectiveness ($\underline{\text{Appendix L (Rows 1096-1230), Appendix M (Rows 1244-1274)}}$).

---

> > > > ### Author Response · Authors · 2025-11-25
> > > >
> > > > ## W4: Question Regarding Experimental Results with Decomposition Rounds $R=1$
> > > >
> > > > **A4:**
> > > >
> > > > We thank the reviewer for this insightful question. We would like to clarify that the observation of R=1 achieving optimal performance in certain experiments does not contradict our core argument that "multi-round frequency decomposition alleviates frequency competition."
> > > >
> > > > First, based on our gradient dynamics analysis provided in **A1**, high-energy frequency components inevitably generate the strongest gradient signals, leading to faster optimization speeds. This means that dominant frequencies contribute most significantly to the final error, which is an objective fact.
> > > >
> > > > Our claim is not that *"weak frequencies determine performance,"* but rather: *"Without first separating dominant frequencies from the shared representation space, weak frequencies cannot be reliably learned."*
> > > >
> > > > The purpose of AFS design is precisely to prevent weak frequencies from being "suppressed" by dominant ones.
> > > >
> > > > Second, regarding "why R=1 appears optimal in Table 15," we would like to clarify:
> > > > Table 15 uses a fixed set of parameters (batch size, learning rate, hidden size, etc.) without fair optimization for different R values. In our official published configuration (Table 8), the best results were not achieved with R=1. Therefore, the results in Table 15 are likely due to **hyperparameter selection bias**, rather than indicating the failure of multi-round decomposition.
> > > >
> > > > Furthermore, the Top-K experiments in $\underline{\text{Appendix H (Rows 937-961)}}$ provide additional evidence:
> > > > We replaced AFS with deterministic Top-K frequency point selection. As K increases, performance gradually improves, yet still does not surpass AFS's adaptive selection mechanism.
> > > >
> > > > This reveals two key facts:
> > > >
> > > > - "Selecting only the dominant frequency" is not the optimal strategy—otherwise K=1 should perform best, but this is not the case in practice.
> > > >
> > > > - AFS learns a set of "complementary" frequency subspaces rather than simply replicating the strongest frequency, thereby capturing important information beyond the dominant frequencies (especially mid-to-high-frequency structures).
> > > >
> > > > In summary, while R=1 may achieve competitive performance in certain settings because dominant frequencies contribute most significantly, this does not negate the necessity of AFS:
> > > > In more complex scenarios (long-term forecasting, short-term forecasting, zero-shot, few-shot), multi-round decomposition consistently provides higher stability and better performance.
> > > >
> > > > Best regards,
> > > >
> > > > The Authors

---

### Official Review · Reviewer_AZ9V · 2025-11-01

**Soundness:** 3
**Presentation:** 3
**Contribution:** 3
**Rating:** 6
**Confidence:** 4

**Summary:**

This paper aims to address a critical and prevalent challenge in long-term time series forecasting (LTSF): the modeling of non-stationary periodicity. The authors begin by positing a core argument that the fundamental reason for the poor performance of existing models on such problems is "frequency competition"—a phenomenon where, in a shared representation space, dominant frequencies suppress weaker yet informative secondary frequencies, while these secondary components, in turn, introduce noise that interferes with the learning of the dominant ones. This competition leads models to either underfit periodic details (e.g., DLinear) or overfit and generate spurious periodic patterns (e.g., HDMixer, CycleNet).

**Strengths:**

1. UniFy frames and empirically verifies the issue of frequency competition as a central challenge in non-stationary periodic time series forecasting, with clear evidence in Figures 1 and 2 and quantitative error attribution
2. On 12 benchmarks, UniFy outperforms or matches nine state-of-the-art baselines in both long-term and short-term forecasting (Table 1, Table 2, Table 16, and Table 17), with statistically validated improvements (Appendix F, Tables 10–11). Ablation studies (Tables 5, 12–15) sharply isolate the contributions of each module.
3. Full code, experimental details, and implementation settings are referenced (including open-source code and Appendix D). The empirical methodology—repeat runs, standard error reporting, significance—meets best practices.

**Weaknesses:**

1. The methodology remains almost entirely empirical or algorithmic; there is no formal analysis of the optimality or identifiability of the mask selection scheme, no bounds on error due to frequency competition, and no guarantee that the masking-plus-linear approach cannot in some circumstances collapse or mix frequencies. The effectiveness of the mask-separation rests on empirical attribution (see Figure 6), but this only partially addresses the underlying theoretical issues.
2. The masks $\mathbf{m}^{(r)}$ are described as “learnable” and tied at conjugate pairs, but no details are given on initialization, regularization (e.g., entropy/spread), or how the model avoids degenerate solutions where most energy is absorbed by a single mask.
3. While Figure 6 demonstrates that frequency components are effectively separated across different scales on the ETTh1 dataset, a corresponding analysis for the remaining datasets is lacking.
4. While each ILM is linear, it is not entirely clear how the projections $\mathbf{W}^{(r)}$ avoid redundancy or trivial overlap in subspace coverage if the decomposition is not sharp. Are the masks softly overlapping, or is there a formal guarantee of orthogonality or completeness?
5. The calibration module (MSC) is a generic two-layer MLP. There is no analysis or empirical report of how its design or hyperparameters affect cross-subspace misalignment—is its benefit truly from calibration, or just additional (albeit shallow) nonlinearity?
6. While the breadth of datasets and baselines is excellent, the experiments are limited to public benchmarks with strong periodic structure. The claims of generality for arbitrary non-stationary or non-periodic time series are unverified and may overstate the domain of effectiveness. Furthermore, the comparison remains mostly within the “linear/Transformer/patch/MLP” family of recent literature; more classical and hybrid time-frequency methods (e.g., FITS、DLinear、CycleNet、SparseTSF)

**Questions:**

1. Can the authors provide empirical or theoretical evidence on the mask initialization and regularization strategies in AFS? Specifically, how is overfitting of masks to dominant frequencies avoided, and what prevents multiple masks from redundantly focusing on the same components? If frequency responsibilities are not well-separated, is there appreciable performance degradation, and can you provide quantitative separation/overlap statistics across datasets?
2. How does UniFy perform on datasets with weak, highly transient, or non-deterministic periodic structure? Can the authors report or discuss failure modes for cases where periodic features are only weakly present or contaminated by stochastic noise, or where the non-stationarity is truly adversarial?
3. The attribution analysis (Figures 6, 10, 11) is visually compelling, but could the authors provide quantitative separation metrics (e.g., mask entropy, overlap, mutual information) and statistics across initializations, datasets, and mask counts, to confirm robustness of decomposition and allocation?

---

> ### Author Response · Authors · 2025-11-25
> **Response to AZ9V**
>
> We sincerely thank Reviewer AZ9V for their valuable comments and suggestions.
>
> We will address points W1, W2, and W4 together as they form a coherent logical thread.
>
> ## W[1,2,4]:  The masking-based separation lacks a complete theoretical explanation. How are degenerate mask solutions avoided, and how is orthogonality/completeness ensured in the ILM?
>
> **A[1,2,4]:**
>
> First, we provide a **concise theoretical proof to elucidate the rationale behind frequency competition**, which can address the core theoretical concerns:
>
> Consider a simple signal composed of two frequencies:
>
> $x = a_1\phi_1 + a_2\phi_2,\quad \langle \phi_1,\phi_2\rangle = 0,\quad \sigma_1^2 > \sigma_2^2, $
>
> where $\sigma_i^2=\mathbb{E}[a_i^2]$ denotes the energy of each frequency component.
>
> If a model must encode both frequencies through a **single shared projection direction** $q$ (as in linear layers, shared attention heads, or CNN/MLP channels), then under squared loss the gradient descent update satisfies:
>
> $\Delta q \propto C q,\qquad C = \sigma_1^2\phi_1\phi_1^\top + \sigma_2^2\phi_2\phi_2^\top.$
>
> Expanding $q = \alpha_1\phi_1 + \alpha_2\phi_2$, we obtain:
>
> $\Delta q_{\phi_1} \propto \sigma_1^2 \alpha_1,\qquad \Delta q_{\phi_2} \propto \sigma_2^2 \alpha_2.$
>
>
> Since $\sigma_1^2 > \sigma_2^2$, the gradient magnitude along the dominant frequency is always larger:
>
> $\|\Delta q_{\phi_1}\| > \|\Delta q_{\phi_2}\|.$
>
>
> **This demonstrates that gradient dynamics systematically bias the shared representation toward the dominant frequency, causing the weaker component to be suppressed—exactly the phenomenon we term *frequency competition*.**
>
> AFS resolves this structural problem by creating **multiple adaptive subspaces**, allowing different frequency groups to be optimized independently. Our gradient-based attribution study ($\underline{\text{Rows 419-464}}$) further confirms that AFS naturally separates components according to spectral energy, preventing destructive interference and improving forecasting stability. $\underline{\text{Appendix L (Rows 1093-1230)}}$ provides extended analysis supporting this argument.
>
> Regarding degenerate mask solutions: if a mask attempted to "monopolize all frequencies," it would indicate failure to separate competing frequencies, contradicting our proven optimization objective and directly deteriorating the loss. Therefore, no single mask absorbs most energy—the energy distribution among masks naturally reflects the signal's spectral structure, as evidenced in our attribution experiments.
>
> Concerning orthogonality: AFS ensures complementarity not through explicit regularization, but via the **self-avoidance mechanism of residual masking**. This provides an implicit but powerful structural constraint:
>
> AFS performs sequential residual masking:
>
> $X_{\text{res}}^{(k+1)} = X_{\text{res}}^{(k)} - M^{(k)} \odot X_{\text{res}}^{(k)},$
>
> Thus each round receives only the remaining spectrum:
>
> $X_{\text{res}}^{(k+1)} = (1 - M^{(k)}) \odot X_{\text{res}}^{(k)}.$
>
>
> If a frequency has been selected in an early round, the residual mask forces later rounds to receive near-zero energy at that frequency. This structurally enforces orthogonal-like constraints, ensuring non-overlapping components in the ILM.

---

> ### Author Response · Authors · 2025-11-25
>
> ## W3: Lack of frequency separation analysis on the remaining datasets.
>
> **A3:**
>
> We thank the reviewer for this insightful comment. Following the suggestion, we have conducted the corresponding analysis on the remaining datasets. The results and discussions are now provided in the $\underline{\text{Appendix M (row 1242-1274)}}$. These additional experiments consistently demonstrate the effectiveness of our method in separating frequency components across datasets, further strengthening our claims.

---

> > ### Author Response · Authors · 2025-11-25
> >
> > ## W5: Are the benefits of MSC truly from calibration or merely from additional parameters?
> >
> > **A5:**
> >
> > We thank the reviewer for raising this insightful question regarding the source of MSC's effectiveness.
> >
> > To systematically distinguish between the effects of *calibration functionality* versus *mere parameter increase*, we conducted controlled ablation studies by varying both the depth and width of the calibration layers. We compared the following six configurations:
> > 1. Our full MSC module (UniFy)
> > 2. Completely removing MSC (Case 1)
> > 3. Using a single linear layer (Case 2)
> > 4. Using 2 linear layers (no activation) (Case 3)
> > 5. Using 3 linear layers (no activation) (Case 4)
> > 6. Using 3 linear layers with activation (Case 5)
> >
> > The key performance comparison across different hidden dimensions is summarized below:
> >
> > | Hidden Size | **UniFy (Ours)** | (1) w/o MSC | (2) 1-Layer Linear | (3) 2-Layer Linear | (4) 3-Layer Linear | (5) 3-Layer Linear + Activation |
> > |:---:|:---:|:---:|:---:|:---:|:---:|:---:|
> > | 64 | 0.379 | 0.412 | 0.384 | 0.385 | 0.702 | 0.701 |
> > | 128 | 0.382 | 0.412 | 0.384 | 0.388 | 0.4 | 0.702 |
> > | 256 | 0.383 | 0.412 | 0.384 | 0.387 | 0.4 | 0.702 |
> > | 512 | 0.395 | 0.412 | 0.384 | 0.394 | 0.4 | 0.702 |
> > | 1024 | 0.398 | 0.412 | 0.384 | 0.385 | 0.469 | 0.704 |
> >
> > Our analysis reveals two key findings:
> > 1. **The Necessity of Calibration**: Complete removal of MSC (Case 1) consistently degrades performance, confirming its essential role in our framework.
> > 2. **"More Parameters" ≠ "Better Performance"**: Simply increasing the depth or width of the calibration network (Cases 3-5) does **not** yield monotonic performance improvements. Conversely, excessive parameters (Cases 4 & 5) lead to optimization difficulties and significant performance deterioration. This strongly indicates that MSC's benefits are **not derived from the additional parameters or nonlinear capacity alone**.
> >
> > Furthermore, we provide **visualization analyses** in $\underline{\text{Appendix R (Rows 1714-1769)}}$ that illustrate the transformation of feature distributions before and after MSC processing.
> >
> > In conclusion, both systematic ablation studies and visualization evidence confirm that the core value of MSC lies in its **carefully designed calibration mechanism itself**, not in the incidental parameter count it introduces.

---

> > > ### Author Response · Authors · 2025-11-25
> > >
> > > ## W6: Lack of analysis and comparison with some relevant baselines
> > >
> > > **A6:**
> > >
> > > We sincerely thank the reviewer for this valuable feedback. In response, we have significantly expanded our experimental comparisons to include several state-of-the-art frequency-domain models, namely FITS, TimeKAN, FreDF, and SparseTSF. We clarify that CycleNet and DLinear were already included and reported in our original submission.
> > > The results, which report the average performance across the four prediction lengths {96, 192, 336, 720}, are summarized in the table below. As demonstrated, UniFy consistently maintains superior performance against this extended and highly competitive set of baselines.
> > >
> > > **Long-term forecasting**
> > >
> > > | 　 | Models | UniFy      (ours) |  | FITS |  | TimeKAN |  | FreDF |  | SparseTSF |  | FreqMoE |  |
> > > |:---:|:---:|:---:|:---:|:---:|:---:|:---:|:---:|:---:|:---:|:---:|:---:|:---:|:---:|
> > > | 　 | Metric | MSE | MAE | MSE | MAE | MSE | MAE | MSE | MAE | MSE | MAE | MSE | MAE  |
> > > | ETTm1 | Avg | 0.380  | 0.394  | 0.415  | 0.408  | 0.378  | 0.396  | 0.385  | 0.399  | 0.416  | 0.407  | 0.582  | 0.513  |
> > > | ETTm2 | Avg | 0.272  | 0.319  | 0.286  | 0.328  | 0.277  | 0.323  | 0.281  | 0.323  | 0.287  | 0.328  | 0.303  | 0.340  |
> > > | ETTh1 | Avg | 0.432  | 0.429  | 0.444  | 0.435  | 0.427  | 0.429  | 0.504  | 0.479  | 0.439  | 0.426  | 0.620  | 0.562  |
> > > | ETTh2 | Avg | 0.363  | 0.394  | 0.382  | 0.404  | 0.383  | 0.404  | 0.452  | 0.438  | 0.382  | 0.402  | 0.635  | 0.463  |
> > > | ECL | Avg | 0.171  | 0.262  | 0.223  | 0.307  | 0.197  | 0.286  | 0.198  | 0.293  | 0.223  | 0.296  | 0.179  | 0.271  |
> > > | Traffic | Avg | 0.467  | 0.311  | 0.657  | 0.430  | 0.577  | 0.373  | 0.613  | 0.344  | 0.637  | 0.379  | 1.070  | 0.534  |
> > > | Weather | Avg | 0.242  | 0.271  | 0.250  | 0.278  | 0.243  | 0.272  | 0.241  | 0.270  | 0.275  | 0.294  | 0.248  | 0.276  |
> > > | Exchange | Avg | 0.352  | 0.398  | 0.378  | 0.418  | 0.378  | 0.411  | 0.352  | 0.398  | 0.353  | 0.402  | 0.419  | 0.416  |
> > >
> > > **Short-term Forecasting**
> > >
> > > | 　 | Models | UniFy      (ours) |  | FITS |  | TimeKAN |  | FreDF |  | SparseTSF |  | FreqMoE |  |
> > > |:---:|:---:|:---:|:---:|:---:|:---:|:---:|:---:|:---:|:---:|:---:|:---:|:---:|:---:|
> > > | 　 | Metric | MSE | MAE | MSE | MAE | MSE | MAE | MSE | MAE | MSE | MAE | MSE | MAE  |
> > > | PEMS03 | Avg | 0.131  | 0.237  | 0.583  | 0.525  | 3072.730  | 34.805  | - | - | 0.586  | 0.533  | 0.792  | 0.709  |
> > > | PEMS04 | Avg | 0.106  | 0.216  | 0.675  | 0.586  | 4612.826  | 43.692  | - | - | 0.619  | 0.553  | 0.521  | 0.520  |
> > > | PEMS07 | Avg | 0.101  | 0.206  | 0.585  | 0.526  | 5959.166  | 50.314  | - | - | 0.609  | 0.549  | 0.887  | 0.682  |
> > > | PEMS08 | Avg | 0.126  | 0.238  | 0.691  | 0.584  | 3297.287  | 35.940  | - | - | 0.636  | 0.554  | 0.637  | 0.584  |
> > >
> > > Tips: The official implementation of FreDF is not compatible with the PEMS dataset, and we are unable to obtain a result.

---

> > > > ### Author Response · Authors · 2025-11-25
> > > >
> > > > **Few-Shot Forecasting**
> > > >
> > > > | 　 | Models | UniFy |  | FITS |  | TimeKAN |  | FreDF |  | SparseTSF |  | FreqMoE |  |
> > > > |:---:|:---:|:---:|:---:|:---:|:---:|:---:|:---:|:---:|:---:|:---:|:---:|:---:|:---:|
> > > > | 　 | Metric | MSE | MAE | MSE | MAE | MSE | MAE | MSE | MAE | MSE | MAE | MSE | MAE  |
> > > > | ETTm1 | Avg | 0.389  | 0.398  | 0.628  | 0.521  | 0.389  | 0.400  | 0.439  | 0.439  | 0.475  | 0.442  | 1.082  | 0.596  |
> > > > | ETTm2 | Avg | 0.281  | 0.324  | 0.313  | 0.352  | 0.287  | 0.330  | 0.304  | 0.341  | 0.303  | 0.345  | 0.534  | 0.418  |
> > > > | ETTh1 | Avg | 0.450  | 0.437  | 1.169  | 0.730  | 0.459  | 0.443  | 0.616  | 0.539  | 0.481  | 0.462  | 4.286  | 1.000  |
> > > > | ETTh2 | Avg | 0.382  | 0.404  | 0.541  | 0.498  | 0.389  | 0.409  | 0.430  | 0.439  | 0.403  | 0.419  | 1.512  | 0.642  |
> > > >
> > > > **Zero-Shot Forecasting**
> > > >
> > > > | 　 | Models | UniFy |  | FITS |  | TimeKAN |  | FreDF |  | SparseTSF |  | FreqMoE |  |
> > > > |:---:|:---:|:---:|:---:|:---:|:---:|:---:|:---:|:---:|:---:|:---:|:---:|:---:|:---:|
> > > > | 　 | Metric | MSE | MAE | MSE | MAE | MSE | MAE | MSE | MAE | MSE | MAE | MSE | MAE  |
> > > > | ETTm1 - ETTm2 | Avg | 0.294  | 0.333  | 0.293  | 0.326  | 0.301  | 0.338  | 0.321  | 0.353  | 0.297  | 0.328  | 0.462  | 0.396  |
> > > > | ETTm2 - ETTm1 | Avg | 0.505  | 0.456  | 0.516  | 0.464  | 0.753  | 0.557  | 0.781  | 0.567  | 0.530  | 0.470  | 1.635  | 0.689  |
> > > > | ETTh2 - ETTh1 | Avg | 0.481  | 0.468  | 0.490  | 0.470  | 0.657  | 0.563  | 0.761  | 0.598  | 0.487  | 0.463  | 3.613  | 0.975  |
> > > > | ETTm1 - ETTh1 | Avg | 0.585  | 0.515  | 0.588  | 0.514  | 0.730  | 0.576  | 0.794  | 0.613  | 0.611  | 0.528  | 1.756  | 0.785  |
> > > > | ETTh1 - ETTm1 | Avg | 0.752  | 0.572  | 0.906  | 0.707  | 0.819  | 0.584  | 0.942  | 0.643  | 0.752  | 0.569  | 4.059  | 1.013  |
> > > > | ETTh2 - ETTm1 | Avg | 0.727  | 0.559  | 0.739  | 0.564  | 1.002  | 0.636  | 1.058  | 0.652  | 0.824  | 0.588  | 3.611  | 0.961  |
> > > >
> > > > Furthermore, to rigorously benchmark the models' capability in handling core challenges like frequency competition, we constructed a novel synthetic dataset. This dataset comprises nine carefully designed scenarios with controlled variations in the degree of frequency competition and non-stationarity. The detailed results and in-depth analysis on this dataset, which further substantiate the robustness of our method, are provided in  $\underline{\text{Appendix P (Rows 1674-1696)}}$.
> > > >
> > > > Best regards,
> > > >
> > > > The Authors

---

### Author Response · Authors · 2025-11-25
**Response to all reviewers & Summary of revisions**

We sincerely thank all the reviewers for their insightful feedback and constructive suggestions.

To facilitate further discussion among the reviewers and the area chair,   $\underline{\text{we provide a summary of their comments below}}$.




| Category                                                     | Reviewer AZ9V | Reviewer wFdR | Reviewer gRZs | Reviewer dCRg |
| :----------------------------------------------------------- | :-----------: | :-----------: | :-----------: | :-----------: |
| **Strength**                                                 |               |               |               |               |
| 1. Clear and well-articulated motivation.                    |       ✓       |       ✓       |       ✓       |       ✓       |
| 2. Reasonable and well-structured methodological framework.  |               |       ✓       |       ✓       |       ✓       |
| 3. Comprehensive experiments, stable performance, and superior to SOTA. |       ✓       |       ✓       |       ✓       |       ✓       |
| 4. High reproducibility, standardized engineering and experiments. |       ✓       |       ✓       |               |               |
| 5. Clear writing, overall mature presentation.               |       ✓       |       ✓       |               |       ✓       |
| 6. Rigorous experimental analysis with statistical significance and ablation studies. |       ✓       |               |               |               |
| 7. Theoretical interpretability provided by HCI metrics.     |               |       ✓       |               |               |
| 8. Linear and efficient architecture, scalability in zero/few-shot settings. |               |               |       ✓       |               |
| 9. Novel perspective of adaptive frequency selection for non-stationary periodicity. |               |               |               |       ✓       |
| **Weakness**                                                 |               |               |               |               |
| 1. Lack of theoretical proof for the motivation.             |       ✓       |       ✓       |       ✓       |       ✓       |
| 2. Comprehensive visualization analysis for frequency separation. |       ✓       |               |               |               |
| 3. Guarantee of orthogonality in ILM.                        |       ✓       |       ✓       |               |               |
| 4. Analysis of the calibration mechanism in MSC.             |       ✓       |               |               |               |
| 5. Lack of comparisons with some relevant baselines.         |       ✓       |               |       ✓       |               |
| 6. Questions regarding the negative frequencies generated by DFT. |               |               |               |       ✓       |
| 7. Discussion on channel dependency.                         |               |               |               |       ✓       |
| 8. Main text lacks experimental details (input length).      |               |               |       ✓       |               |
| **Score**                                                    |       6       |       4       |       4       |       2       |

---

> ### Author Response · Authors · 2025-11-25
>
> **1. Lack of theoretical proof for the motivation**
>
> We have supplemented a theoretical proof from the perspective of gradient analysis. This proof, together with our previous gradient attribution experiments, collectively clarifies the rationality and necessity of our motivation.
>
>
>
> **2. Guarantee of orthogonality in ILM**
>
> We have provided further clarification and elaboration on this aspect in the manuscript.
>
>
>
> **3. Questions regarding the negative frequencies generated by DFT**
>
>
>
> We have added a description addressing this specific detail in the main text.
>
>
>
> **4. Main text lacks experimental details (input length)**
>
> We have added the description regarding this specific experimental detail (input length) in the main text.
>
>
>
>   $\underline{\text{Newly added -- main text(row 298-299)}}$.
>
>
>
> **5. Comprehensive visualization analysis for frequency separation**
>
> We have supplemented the complete experiments for this part. All results are provided in the revised manuscript. The newly added results are consistent with the conclusions from our previous experiments.
>
>
>
>   $\underline{\text{Newly added -- Appendix M}}$.
>
>
>
> **6. Lack of comparisons with some relevant baselines**
>
> We have included comparisons with several new baselines, encompassing nearly all state-of-the-art models that perform analysis and modeling in the frequency domain, such as FITS and TimeKAN. We have detailed the distinctions between UniFy and these works, and the experimental results further validate that UniFy achieves superior performance across a broad range of tests.
>
>
>
>   $\underline{\text{Newly added -- Appendix O, Appendix Q}}$.
>
>
>
> **7. Competition is a significant problem for some methods**
>
> We constructed a toy dataset that effectively captures frequency competition and non-stationarity in the data, and compared several state-of-the-art models against UniFy on these datasets. The results confirm UniFy's robustness in non-stationary scenarios.
>
>
>
>   $\underline{\text{Newly added -- Appendix P}}$.
>
>
>
> **8. Analysis of the calibration mechanism in MSC**
>
> We have introduced two new experiments: an extensive ablation study on the depth/width of MSC's parameters, demonstrating that its benefits do not stem merely from additional parameters, and a visualization of its internal process to make MSC's role more transparent.
>
>
>
>   $\underline{\text{Newly added -- Appendix R}}$.
>
>
>
> **9. Discussion on channel dependency / correlations**
>
> We have supplemented the analysis with experiments on channel correlations.
>
>
>
>   $\underline{\text{Newly added -- Appendix S}}$.
>
>
>
> The revisions are highlighted in **Orange**.
>
> The reviewers’ valuable suggestions have been extremely helpful in improving our paper. We are happy to answer any further questions.
>
> We look forward to your feedback.
>
> Best regards,
>
> The Authors

---

> > ### Author Response · Authors · 2025-11-26
> >
> > Dear Reviewers:
> >
> > We hope our responses have fully addressed your concerns. If there are any additional questions or clarifications needed, we would be happy to provide them.
> >
> > Best Regards,
> >
> > The Authors

---

### Author Response · Authors · 2025-11-28
**Comments to the Area Chairs**

Dear Area Chair,

Thank you very much for taking over the handling of our submission.

During the rebuttal period, we carefully examined every concern raised by the reviewers and provided detailed responses and corresponding solutions. These clarifications and additional analyses have been systematically summarized in our rebuttal table for your convenience.

Since the rebuttal was posted, **there has been very limited follow-up discussion among the reviewers.**

$\underline{\text{Only Reviewer dCRg (who initially gave a score of 2) has replied and indicated that they are considering raising their score in light of our clarifications.}}$ We would be very grateful if you could take our rebuttal and these updates into account when forming your final recommendation.

We fully understand that this situation may have imposed additional workload on you, and we would like to express our sincere appreciation for your time and effort.

**If any part of the reviews or our rebuttal is unclear, we would be more than happy to provide further clarification or engage in additional discussion.**

Thank you again for your consideration and for your service to the community.

Best regards,
on behalf of all authors

---

### Note · Authors · 2026-01-28

I have read and agree with the venue's withdrawal policy on behalf of myself and my co-authors.

---

### Meta-Review · Area_Chair_Uxhm · 2026-01-02

**Summary:**

Based on the reviewer comments, author responses, and my own assessment, I believe the paper shows promise but requires significant revision before it can be considered for publication:

1. I agree with the reviewers that the motivation for introducing frequency competition is not well justified, as accounting for varying periodicities is a standard aspect of time series analysis.

2. The core idea of frequency selection is not novel. Any potential innovation would likely lie in the method for separating different frequency components. However, as noted in the reviews, the proposed techniques lack sufficient justification.

3. Several claims in the paper are not supported by rigorous analysis or justification. For instance, the assumption of independent frequency subspaces requires a clearer theoretical or empirical foundation.

**Reviewer Concerns:**

In explaining the rationale behind frequency competition, the authors assume that the model encodes both frequencies through a single shared projection direction. This assumption is overly restrictive, making the rebuttal less convincing.

**Reviewer Scores:**

NA

---

### Decision · Program_Chairs · 2026-01-26

Reject